# Glucose-6-phosphate dehydrogenase maintains redox homeostasis and biosynthesis in *LKB1*-deficient *KRAS*-driven lung cancer

Taijin Lan[1], Sara Arastu[1], Jarrick Lam[1], Hyungsin Kim[1], Wenping Wang[1], Samuel Wang [1], Vrushank Bhatt[1], Eduardo Cararo Lopes [1,2], Zhixian Hu[1], Michael Sun[1], Xuefei Luo[1], Jonathan M. Ghergurovich [3], Xiaoyang Su [1,4], Joshua D. Rabinowitz [1,5,6,7], Eileen White [1,2,6] & Jessie Yanxiang Guo [1,4,8] ✉

Cancer cells depend on nicotinamide adenine dinucleotide phosphate (NADPH) to combat oxidative stress and support reductive biosynthesis. One major NADPH production route is the oxidative pentose phosphate pathway (committed step: glucose-6-phosphate dehydrogenase, G6PD). Alternatives exist and can compensate in some tumors. Here, using genetically-engineered lung cancer mouse models, we show that G6PD ablation significantly suppresses *Kras^{G12D/+};Lkb1^{-/-}* (KL) but not *Kras^{G12D/+};P53^{-/-}* (KP) lung tumorigenesis. In vivo isotope tracing and metabolomics reveal that G6PD ablation significantly impairs NADPH generation, redox balance, and de novo lipogenesis in KL but not KP lung tumors. Mechanistically, in KL tumors, G6PD ablation activates p53, suppressing tumor growth. As tumors progress, G6PD-deficient KL tumors increase an alternative NADPH source from serine-driven one carbon metabolism, rendering associated tumor-derived cell lines sensitive to serine/glycine depletion. Thus, oncogenic driver mutations determine lung cancer dependence on G6PD, whose targeting is a potential therapeutic strategy for tumors harboring *KRAS* and *LKB1* co-mutations.

Tumor cells use nicotinamide adenine dinucleotide phosphate (NADPH) for redox homeostasis and reductive synthesis reactions to sustain their survival and growth[1,2]. Consumption and production of NADPH are compartmentalized in the mitochondria and cytosol[3,4]. Cytosolic NADPH is recycled through reduction of NADP[+] via the oxidative pentose phosphate pathway (oxPPP) enzymes glucose 6-phosphate dehydrogenase (G6PD) and 6-phosphogluconate dehydrogenase (6PGD), malic enzyme 1 (ME1), isocitrate dehydrogenase 1 (IDH1), and the one-carbon metabolism (folate) enzymes methylenetetrahydrofolate dehydrogenase 1 (MTHFD1) and aldehyde dehydrogenase 1 family member L1 (ALDH1L1)[3,5]. The functional importance of different metabolic enzymes involved in cytosolic NADPH homeostasis are not fully understood in cancer in vivo. Better understanding may open therapeutic opportunities.

[1]Rutgers Cancer Institute, New Brunswick, NJ 08901, USA. [2]Department of Molecular Biology and Biochemistry, Rutgers University, Piscataway, NJ 08854, USA. [3]Department of Molecular Biology, Princeton University, Princeton, NJ 08544, USA. [4]Department of Medicine, Rutgers Robert Wood Johnson Medical School, New Brunswick, NJ 08901, USA. [5]Department of Chemistry, Princeton University, Princeton, NJ 08544, USA. [6]Ludwig Princeton Branch, Ludwig Institute for Cancer Research, Princeton University, Princeton, NJ 08544, USA. [7]Lewis-Sigler Institute of Integrative Genomics, Princeton University, Princeton, NJ 08544, USA. [8]Department of Chemical Biology, Rutgers Ernest Mario School of Pharmacy, Piscataway, NJ 08854, USA. ✉e-mail: yanxiang@cinj.rutgers.edu

Pentose phosphate pathway (PPP) flux is the major alternative glucose catabolic pathway to glycolysis. Dysregulation of proteins in this pathway is associated with cancer development, with the master antioxidant transcription factor NRF2 frequently upregulated in human cancers and driving oxPPP gene expression[6,7]. The oxPPP pathway is essential for mammals, with knockout of the committed enzyme G6PD embryonic lethal. G6PD deficiency is the most common human enzyme defect because it protects against malaria[8]. G6PD is upregulated in many cancers, and G6PD deficiency is associated with lower cancer risk and mortality for some cancers[9–12], suggesting that cancer cells may depend on G6PD for survival or proliferation. Loss of p53 upregulates G6PD activity and promotes NADPH-driven biosynthetic processes including de novo lipogenesis[13]. In mouse models, G6PD deficiency significantly reduces melanoma metastasis[14]. Recently, we employed modern genetic tools to evaluate the role of G6PD in lung, breast, and colon cancer driven by oncogenic *KRAS*. We found that, in the studied *KRAS* mutant tumor models, G6PD, at most modestly promotes disease progression and is not strictly essential for solid tumorigenesis or metastatic spread[15]. In particular, G6PD is not required for *Kras*$^{G12D/+}$;*P53*$^{-/-}$ (KP) lung tumorigenesis[15]. However, KP tumors further lacking KEAP1, a tumor suppressor whose loss elevates NRF2, show greater dependence on G6PD[7]. Thus, G6PD is likely to be particularly important in the context of specific tumor types or driver mutations.

Oncogenic *KRAS* mutation in non-small cell lung cancer (NSCLC) patients confer a poor prognosis and a high risk of cancer recurrence. LKB1 signaling negatively regulates tumor growth through direct phosphorylation and activation of the central metabolic sensor, AMP-activated protein kinase (AMPK), which governs glucose and lipid metabolism in response to alterations in nutrients and intracellular energy levels[16–19]. Loss of LKB1 reprograms cancer cell metabolism to efficiently generate energy and biomass components for uncontrolled proliferation and dissemination. Meanwhile, such alterations in turn cause tumor cells to have less plasticity in response to metabolic stress, creating a metabolic vulnerability[20,21]. p53 and LKB1 co-mutations represent two different subgroups of *KRAS*-driven NSCLC, with distinct biological properties, metabolic vulnerabilities, and responses to standard therapies[22–25]. We have revealed that G6PD is not essential for KP lung tumorigenesis[15]. Given the distinct features of KL and KP NSCLC, we here further investigated the dispensability of G6PD in *Kras*$^{G12D/+}$;*Lkb1*$^{-/-}$ (KL) lung tumorigenesis. In contrast to KP lung cancers, G6PD showed greater functional importance in KL lung cancers. G6PD deficiency impaired KL lung tumorigenesis, showing increased oxidative stress, and p53 activation-mediated apoptosis and cell cycle arrest. G6PD loss also impaired de novo lipogenesis, whereas fat supplementation rescued the growth of G6PD-deficient KL lung tumors. Moreover, G6PD loss in KL tumors reprogrammed the NADPH generating metabolic pathway by increasing serine uptake to sustain one carbon metabolism-mediated cytosolic NADPH generation. G6PD-deficient KL lung tumor-derived cell lines (TDCLs) were sensitive to serine/glycine depletion, which was associated with increased reactive oxygen species (ROS). Thus, the dependence of G6PD-mediated oxPPP on *KRAS*-driven lung tumorigenesis is determined by specific oncogenic driver mutations. This also underscores the need for personalized therapies tailored to different subgroups of *KRAS*-driven lung cancers, especially when considering the application of G6PD inhibitors in cancer treatment.

## Results

### *G6PD* expression level correlates with the survival of lung cancer patients carrying *KRAS* and *LKB1* co-mutations

Tumors exhibit an enormous demand for NADPH due to uncontrolled proliferation[2]. The generation of cytosolic NADPH is primarily by metabolic enzymes such as G6PD, IDH1, ME1, and MTHFD1. Review of the cBioPortal datasets gave us further insight to conduct an overall survival comparison between low and high mRNA expression levels of cytosolic NADPH generating enzymes in lung cancer patients with *KRAS* wild type (WT) and *KRAS* mutation. In lung cancer patients with *KRAS* WT, except for *MTHFD1*, the expression levels of *G6PD*, *IDH1*, and *ME1* were not correlated with the patient survival time (Supplementary Fig. 1a). In patients with *KRAS* mutation, high expression levels of *G6PD* and *MTHFD1* were associated with poorer overall survival (Supplementary Fig. 1b). We further analyzed these correlations in patients carrying *KRAS/TP53* co-mutations and *KRAS/LKB1* co-mutations. We found that high expression level of *G6PD* was associated with poorer overall survival in patients with *KRAS/LKB1* co-mutations (Supplementary Fig. 1c), but not in patients with *KRAS/TP53* co-mutations (Supplementary Fig. 1d). Regarding *MTHFD1*, besides its connection with survival outcomes in lung cancer patients with WT *KRAS* (Fig. 1a), its high expression was also associated to poorer survival in patients with *KRAS/LKB1* co-mutations (Supplementary Fig. 1c). KEAP1/ NRF2 signaling pathway plays an essential role in regulating oxidative stress response and PPP genes[26]. Co-mutations of *KEAP1* and *LKB1* occur in 10% of lung tumors and linked to abnormal NRF2 signaling and unfavorable clinical outcomes[27,28]. We observed that high expression level of *NRF2* was associated with poorer survival in patients with *KRAS/LKB1* co-mutations (Supplementary Fig. 1c), but not in patients with *KRAS/TP53* co-mutations (Supplementary Fig. 1d). In addition, *G6PD* expression was significantly higher in lung cancer harboring *KRAS/LKB1* co-mutations compared to *KRAS/TP53* co-mutations (Supplementary Fig. 1e). Taken together, these analyses suggest that G6PD expression is specifically correlated with survival in a subset of lung cancer patients (*KRAS/LKB1* co-mutations and not *KRAS/TP53* co-mutations).

### G6PD promotes KL lung tumorigenesis but is dispensable for KP lung tumorigenesis

Co-mutations of p53 and LKB1 represent two different subgroups of *KRAS*-driven lung cancer with distinct features[22–25]. We therefore explored the role of G6PD in KP and KL lung tumorigenesis by generating a *G6pd*$^{flox/flox}$;*Kras*$^{LSL-G12D/+}$;*P53*$^{flox/flox}$ (*G6pd*$^{flox/flox}$;KP) genetically engineered mouse model (GEMM) for KP NSCLC and a *G6pd*$^{flox/flox}$;*Kras*$^{LSL-G12D/+}$;*Lkb1*$^{flox/flox}$ (*G6pd*$^{flox/flox}$;KL) GEMM for KL NSCLC, and concurrently deleting *G6pd* and inducing lung tumor via intranasal delivery of Lenti-Cre (Fig. 1a). G6PD expression was completely deleted in KP and KL lung tumors, which was validated by immunohistology (IHC) (Fig. 1b) and mRNA expression from KL bulk-tumor mRNA-seq (Supplementary Fig. 2a, b). In line with our prior findings involving the conditional deletion of G6PD in KP lung tumors through CRISPR/Cas9-mediated gene editing in vivo[15], we have further confirmed that G6PD is dispensable for KP lung tumorigenesis. This was substantiated by comparing various parameters including gross lung pathology (Fig. 1c), wet lung weight (Fig. 1d), quantification of tumor number (Fig. 1f) and tumor burden (Fig. 1g) based on scanned lung hematoxylin & eosin (H&E) staining (Fig. 1e), tumor cell proliferation (Ki67) (Fig. 1h), and mouse survival (Fig. 1i) between mice harboring *G6pd*$^{WT}$;KP lung tumors and those with *G6pd*$^{KO}$;KP lung tumors.

In contrast, G6PD is required for KL lung tumorigenesis. Our investigation conducted at 12 weeks post-tumor induction, revealed that G6PD loss significantly impaired KL lung tumorigenesis, as evidenced by lower wet lung weight, reduced tumor number and tumor burden (Fig. 1j–n). Consistent with these phenotypes, reduced cell proliferation (Ki67) and reduced RAS downstream signaling, additional markers for tumor growth (pERK and pS6) (Fig. 1o, Supplementary Fig. 2c), were observed in *G6pd*$^{KO}$;KL lung tumors compared to *G6pd*$^{WT}$;KL lung tumors. Moreover, mice bearing *G6pd*$^{KO}$;KL lung tumors had a significantly longer life span compared to mice bearing *G6pd*$^{WT}$;KL lung tumors (Fig. 1p). Taken together, we demonstrated that, unlike in KP lung tumors, G6PD supports KL lung tumor initiation and growth.

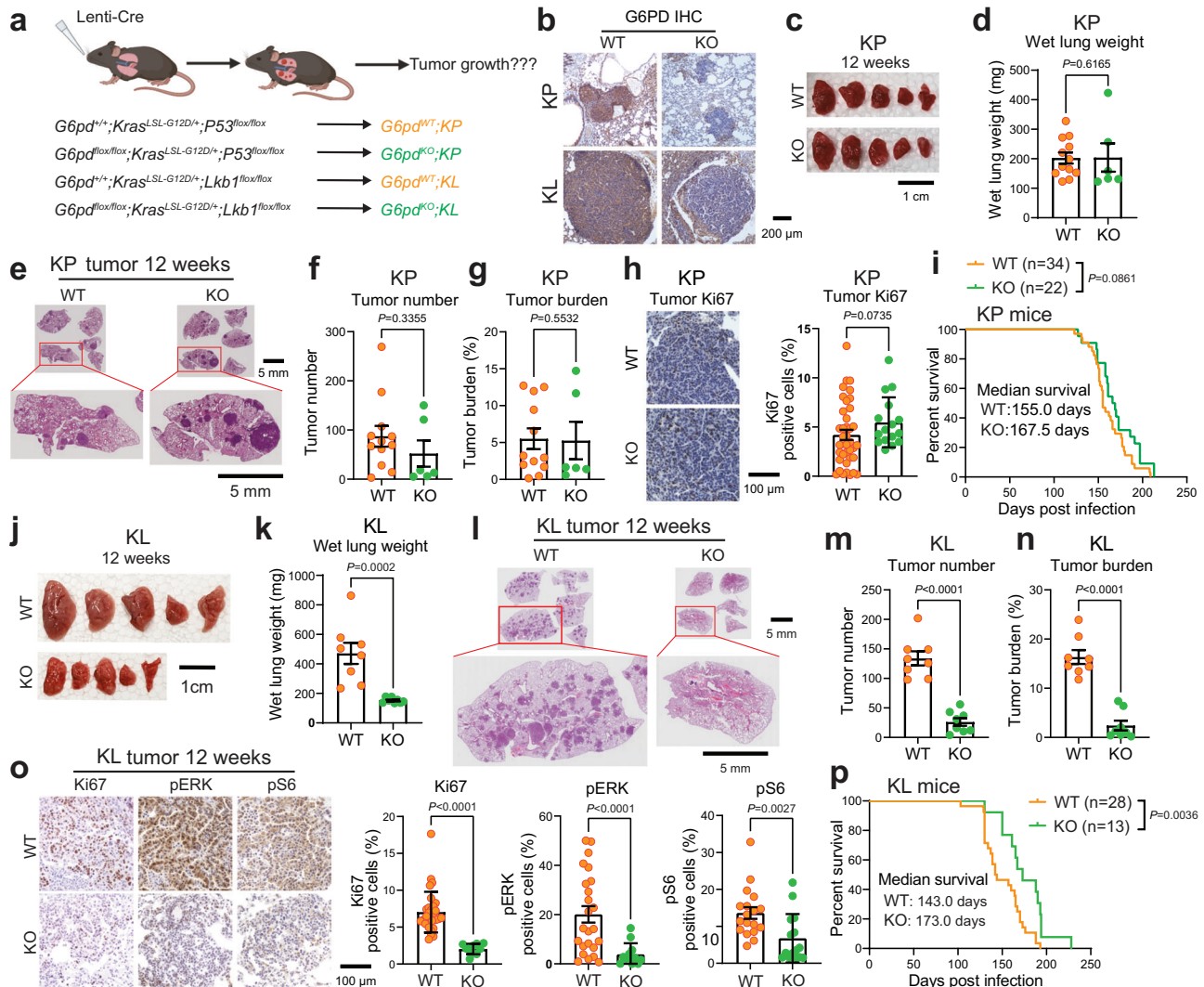

**Fig. 1 | G6PD promotes KL lung tumorigenesis but is dispensable for KP lung tumorigenesis. a** Scheme illustrating the induction of conditional tumoral *G6pd* knockout to investigate the role of G6PD in KP and KL lung tumorigenesis (Created with BioRender.com released under a Creative Commons Attribution-NonCommercial-NoDerivs 4.0 International license). **b** Representative immuno-histochemistry (IHC) images of G6PD in KP and KL lung tumors at 12 weeks post-induction. *n* = 10 images for each genotype, scale bar = 200 μm. **c.** Representative gross lung pathology from mice bearing *G6pd^WT;KP* (*n* = 12 mice) and *G6pd^KO;KP* (*n* = 6 mice) lung tumors at 12 weeks post-tumor induction. Scale bar =1 cm. **d** Graph of wet lung weight from (**c**). **e** Representative H&E staining of scanned lung sections from (**c**). **f, g** Quantification of tumor number (**f**) and tumor burden (**g**) from (**e**). *n* is same with (**c**). **h** Representative IHC images and quantification of Ki67 in *G6pd^WT;KP* (*n* = 39 images) and *G6pd^KO;KP* (*n* = 16 images) lung tumors. Scale bar = 100 μm. **i** Kaplan-Meier survival curve of mice bearing *G6pd^WT;KP* (*n* = 34 mice) or *G6pd^KO;KP*

(*n* = 22 mice) lung tumors. **j** Representative gross lung pathology from mice bearing *G6pd^WT;KL* (*n* = 8 mice) and *G6pd^KO;KL* (*n* = 8 mice) lung tumors at 12 weeks post-tumor induction. Scale bar = 1 cm. **k** Graph of wet lung weight from (**j**). **l** Representative H&E staining of scanned lung sections from (**j**). **m, n** Quantification of tumor number (**m**) and tumor burden (**n**) from (**l**). *n* is same with (**j**). **o** Representative IHC images and quantification of Ki67 (*n* = 31 images for *G6pd^WT;KL, n* = 10 images for *G6pd^KO;KL*), pERK (*n* = 24 images for *G6pd^WT;KL, n* = 12 images for *G6pd^KO;KL*), and pS6 (*n* = 18 images for *G6pd^WT;KL, n* = 15 images for *G6pd^KO;KL*) in KL lung tumors at 12 weeks post-tumor induction. Scale bar = 100 μm. **p** Kaplan-Meier survival curve of mice bearing *G6pd^WT;KL* (*n* = 28 mice) and *G6pd^KO;KL* (*n* = 13 mice) lung tumors. Data are presented as mean ± SEM, significance was calculated by Mann Whitney test (**d, f, g, h, k**, pS6 in **o**), two-tailed unpaired *t*-test (**m**), two-tailed unpaired *t*-test with Welch's correction (**n**, Ki67 and pERK in **o**) or log-rank test (**i, p**). Source data are provided as a Source Data file.

## G6PD maintains cellular redox homeostasis in KL lung tumors

G6PD-mediated oxPPP is one of the cytosolic NADPH generating pathways, essential for maintaining redox balance. We hypothesized that G6PD depletion would suppress NADPH production in KL lung tumors, disrupting redox homeostasis and leading to cell death in the stressed tumor microenvironment. As expected, the pool size level of NADPH in *G6pd^KO;KL* lung tumors was significantly lower than *G6pd^WT;KL* lung tumors, leading to an impaired redox balance in *G6pd^KO;KL* lung tumors, evidenced by a lower ratio of NADPH/NADP+ and GSH/GSSG in *G6pd^KO;KL* lung tumors than *G6pd^WT;KL* tumors (Fig. 2a, b). Moreover, Gene Set Enrichment Analysis (GSEA) of bulk-tumor mRNA-seq revealed that oxidative stress signaling was significantly upregulated in *G6pd^KO;KL*

lung tumors in contrast to *G6pd^WT;KL* lung tumors (Fig. 2c). This was corroborated through IHC analysis of NRF2 and its target NQO1, markers indicating oxidative stress, as well as 8-oxo-dG and γ-H2AX, markers associated with oxidative stress and DNA damage. Expression levels of NRF2, NQO1, 8-oxo-dG and γ-H2AX were significantly increased in *G6pd^KO;KL* lung tumors compared to *G6pd^WT;KL* lung tumors (Fig. 2d, e). In contrast, G6PD loss in KP lung tumors had no impact on the pool size levels of NADPH, NADP+, and the ratio of NADPH/NADP+ (Supplementary Fig. 3a), nor on the pool size levels of glutathione (GSH), glutathione disulfide (GSSG), and the ratio of GSH/GSSG (Supplementary Fig. 3b). Thus, G6PD-mediated NADPH production is critical for maintaining redox homeostasis in KL lung tumors.

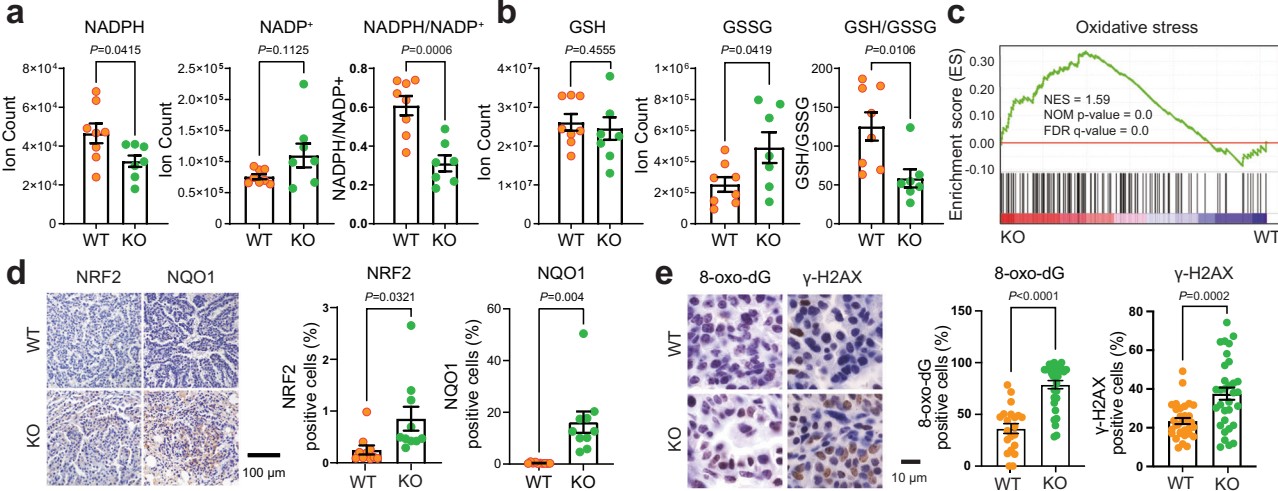

**Fig. 2 | G6PD is required to maintain cellular redox homeostasis in KL lung tumors. a** Pool size of NADPH and NADP⁺, and NADPH/NADP⁺ ratio in *G6pd^WT;KL* (*n* = 8 mice) and *G6pd^KO;KL* (*n* = 7 mice) lung tumors at 12 weeks post-tumor induction. **b** Pool size of GSH and GSSG, and GSH/GSSG ratio in *G6pd^WT;KL* (*n* = 8 mice) and *G6pd^KO;KL* (*n* = 7 mice) lung tumors at 12 weeks post-tumor induction. **c** Gene Set Enrichment Analysis (GSEA) of oxidative stress signaling for *G6pd^KO;KL* (*n* = 7 mice) and *G6pd^WT;KL* (*n* = 8 mice) lung tumors at 12 weeks post-tumor induction based on bulk-tumor mRNA-seq data. Gene set for "Oxidative stress" was downloaded from GeneCards (https://www.genecards.org/, accessed on April 09, 2023). **d** Representative IHC images and quantification of NRF2 and NQO1 in

*G6pd^WT;KL* and *G6pd^KO;KL* lung tumors at 12 weeks post-tumor induction. *n* = 10 images for each quantification, scale bar = 100 μm. **e** Representative IHC images and quantification of 8-oxo-dG (*n* = 21 images for *G6pd^WT;KL*, *n* = 31 images for *G6pd^KO;KL*) and γ-H2AX (*n* = 32 images for both *G6pd^WT;KL* and *G6pd^KO;KL*) in *G6pd^WT;KL* and *G6pd^KO;KL* lung tumors at 12 weeks post-tumor induction. Scale bar = 10 μm. Data are presented as mean ± SEM, significance was calculated by two-tailed unpaired *t*-test (NADPH and NADPH/NADP⁺ in **a**, **b**), Mann-Whitney test (8-oxo-dG in **e**), two-tailed unpaired *t*-test with Welch's correction (NADP⁺ in **a**, **d**, γ-H2AX in **e**). Source data are provided as a Source Data file.

## G6PD maintains cellular redox homeostasis to prevent oxidative stress-induced cell death for KL tumor growth

To further deduce the consequences of damaged redox homeostasis by G6PD loss on KL lung tumor growth, we generated *G6pd^WT;KL* and *G6pd^KO;KL* TDCLs from mouse lung tumors (Fig. 3a, b). Under nutrient-rich conditions, *G6pd^KO;KL* TDCLs displayed lower NADPH, lower ratio of NADPH/NADP⁺ and GSH/GSSG, and higher basal ROS levels than *G6pd^WT;KL* TDCLs (Fig. 3c–e). Additionally, the proliferation rate of *G6pd^KO;KL* TDCLs was significantly slower than *G6pd^WT;KL* TDCLs, as assessed by IncuCyte live-cell analysis system and MTS assay (Fig. 3f, Supplementary Fig. 4a). This decrease in proliferation was not linked to apoptosis and necrosis (Supplementary Fig. 4b, c). Moreover, *G6pd^WT;KL* TDCLs demonstrated sensitivity to G6PD inhibitor (G6PDi-1) (Fig. 3g). Furthermore, compared with *G6pd^WT;KL* TDCLs, *G6pd^KO;KL* TDCLs were more sensitive to H₂O₂-induced oxidative stress (Fig. 3h), which was correlated with a significant increase in ROS levels (Fig. 3i).

We next performed in vivo TDCLs-induced allograft tumor growth to determine whether *G6pd^KO;KL* allograft tumors are more susceptible to oxidative stress than *G6pd^WT;KL* allograft tumors, as observed in in vitro cell culture (Fig. 3h). High-dose Vitamin C (Vit C) has been reported to induce oxidative stress in preclinical mouse models and has been proposed in clinical studies combined with standard therapies[29]. Therefore, we treated mice bearing *G6pd^WT;KL* or *G6pd^KO;KL* allograft tumors with vehicle control or high-dose Vit C (4 g/kg, daily). As expected, G6PD deficiency significantly impeded allograft tumor growth in the vehicle control group. Importantly, a high-dose of Vit C further suppressed the growth of KL allograft tumors only if they also lacked G6PD (Fig. 3j–l). IHC analysis of NRF2, NQO1 and 8-oxo-dG confirmed increased oxidative stress in KL allografts with high-dose Vit C treatment, with *G6pd^KO;KL* allograft tumors exhibiting higher oxidative stress than *G6pd^WT;KL* allografts, further intensified by high-dose Vit C (Fig. 3m). In summary, we demonstrated that G6PD maintains KL lung tumor redox homeostasis to prevent oxidative stress-induced cell death, which is critical for tumor growth.

## G6PD loss activates p53 to suppress KL lung tumorigenesis

DNA damage and oxidative stress activate p53, leading to cell cycle arrest and apoptosis[30]. GSEA of bulk-tumor mRNA-seq showed that the p53-mediated apoptotic pathway was upregulated in *G6pd^KO;KL* lung tumors compared to *G6pd^WT;KL* lung tumors (Fig. 4a). This was further validated by IHC of p53 and its downstream targets, p21 and cleaved caspase-3, which were significantly upregulated in *G6pd^KO;KL* lung tumors compared to *G6pd^WT;KL* lung tumors (Fig. 4b). We subsequently proposed that loss of G6PD in KL lung tumors induces p53 activation, leading to growth arrest and apoptosis. To test this hypothesis, we generated *G6pd^flox/flox;Kras^LSL-G12D/+;P53^flox/flox;Lkb1^flox/flox* (*G6pd^flox/flox;KPL*) and *G6pd^+/+;KPL* mice, and examined the effect of G6PD ablation on KPL lung tumorigenesis (Fig. 4c). G6PD deficiency in KPL lung tumors was confirmed by IHC (Fig. 4d). Notably, p53 loss eliminated the sensitivity of KL lung tumors to G6PD knockout, as evidenced by gross lung pathology, wet lung weight, quantification of tumor number and tumor burden from scanned lung H&E sections at 6 weeks post KPL lung tumor induction (Fig. 4e–i). Moreover, tumor cell proliferation (Ki67) was comparable between *G6pd^WT;KPL* and *G6pd^KO;KPL* lung tumors (Fig. 4j). As a result, the life span of mice bearing *G6pd^WT;KPL* and *G6pd^KO;KPL* lung tumors was similar (Fig. 4k). Despite this, increased oxidative stress was still observed in *KPL* lung tumors with the loss of G6PD, as examined through IHC staining for NRF2 and NQO1 (Fig. 4l). This suggests that oxidative stress alone, without p53, is not sufficient to impede KL lung tumor growth. Thus, the slow growth of G6PD knockout KL lung tumors is due to p53 activation inhibiting tumor progression.

## G6PD depletion impairs KL lung tumor lipid metabolism

Mutations or loss of LKB1 are associated with increased de novo fatty acid synthesis and altered lipid metabolism, contributing to tumor development, progression, and aggressiveness[18]. LKB1 phosphorylates and activates AMPK. AMPK, in turn, regulates various cellular processes to maintain energy balance[19]. AMPK can also be activated by Calcium/calmodulin-dependent protein kinase kinase-beta (CaMKKβ)[31].

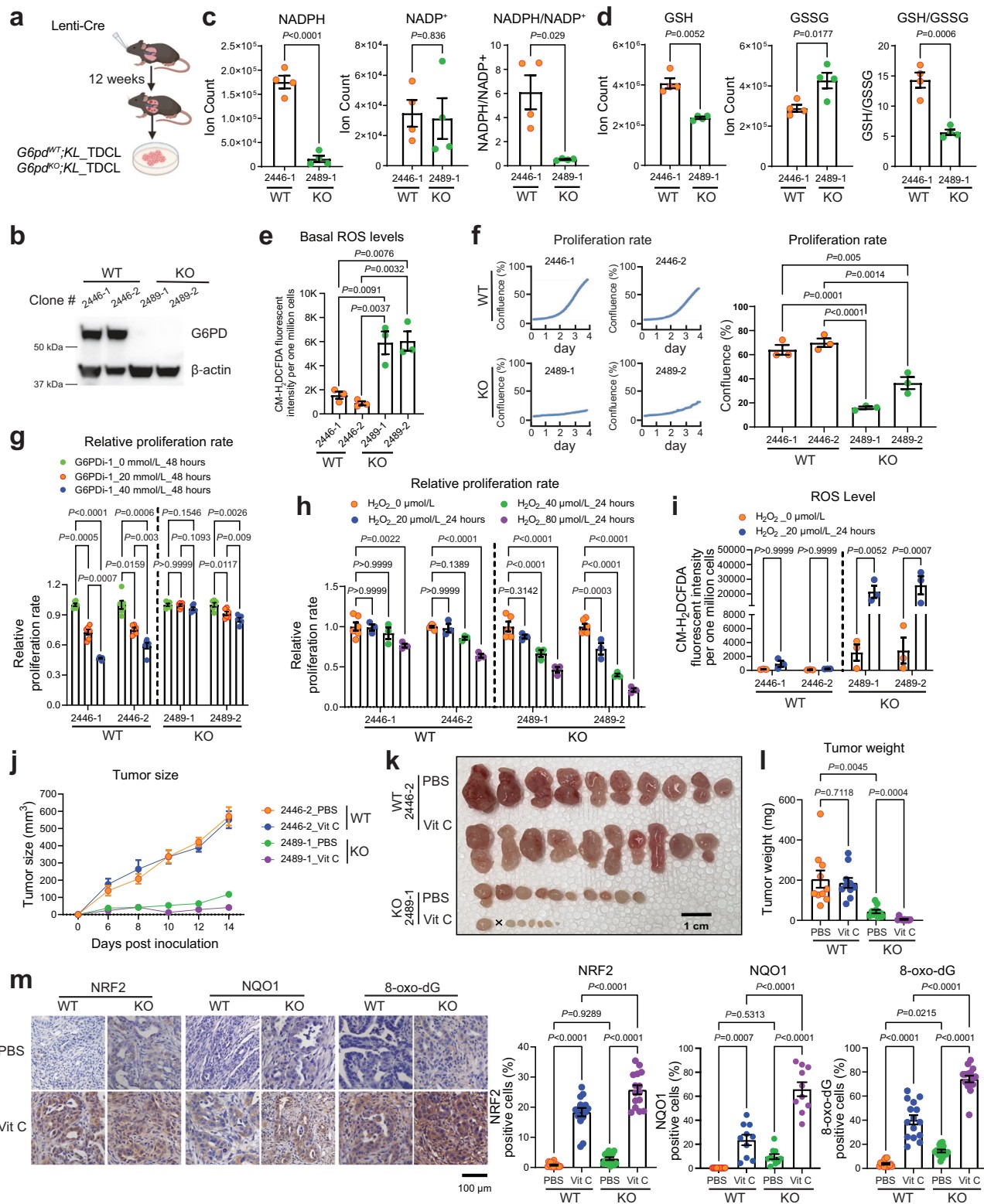

Phosphorylation of Acetyl-CoA Carboxylase (pACC) by AMPK inhibits its activity, leading to a reduction in fatty acid synthesis[32]. Cytosolic NADPH provides a hydride source for de novo fatty acid synthesis. Pathway analysis of bulk-tumor mRNA-seq of KL lung tumors revealed that G6PD ablation significantly altered lipid metabolism by down-regulating the pathways related to lipid and fatty acid biosynthesis (Fig. 5a, b). Lower AMPK activity was observed in *G6pd^WT;KL* lung tumors compared to *G6pd^WT;K (Kras^G12D/+)* lung tumors (Fig. 5c), which supports the known aggressiveness of KL lung tumors relative to K lung

tumors. Conversely, *G6pd^KO;KL* lung tumors exhibited higher AMPK activity than *G6pd^WT;KL* lung tumors (Fig. 5c). These differences in AMPK activity were further confirmed by the pACC (Fig. 5c), suggesting reduced de novo fatty acid synthesis due to G6PD loss.

D_2O has long been used as a tracer for assessing in vivo lipogenesis[33–35], because ²H can be incorporated onto fatty acids primarily via deuterated NADPH exchanged with ambient D_2O[36]. We subsequently examined the de novo fatty acid synthesis of KL lung tumors by intravenously infusing D_2O into mice bearing *G6pd^WT;KL* or

**Fig. 3 | The maintenance of redox homeostasis by G6PD supports the survival of KL tumor cells under oxidative stress conditions. a** Scheme illustrating KL tumor-derived cell lines (TDCLs) generation (Created with BioRender.com released under a Creative Commons Attribution-NonCommercial-NoDerivs 4.0 International license). **b** Western blot of *G6pd^WT* and *G6pd^KO* KL TDCLs. Uncropped Western blot image is shown in Source Data file. **c** Pool size of NADPH and NADP⁺, and NADPH/NADP⁺ ratio of KL TDCLs in nutrient rich conditions (complete RPMI medium). *n* = 4 replicates for each clone. **d** Pool size of GSH and GSSG, and GSH/GSSG ratio of KL TDCLs in nutrient rich conditions. *n* = 4 replicates for each clone. **e** Basal ROS level of KL TDCLs in nutrient rich conditions. *n* = 3 replicates for each clone. **f** Proliferation rate of KL TDCLs in nutrient rich conditions measured by Incucyte for 4 days (left) with statistical analysis at day 4 (right). *n* = 3 replicates for each clone. **g** Relative proliferation rate of KL TDCLs treated with G6PDi-1 for 48 hours. *n* = 6 replicates for each clone at different G6PDi-1 concentrations, except 2489-2 at 40 μmol/L G6PDi-1 with *n* = 5 replicates. **h** Relative proliferation rate of KL

TDCLs treated with H₂O₂ for 24 hours. For each clone, *n* = 6 replicates at 0 μmol/L H₂O₂, *n* = 3 replicates at 20, 40, 80 μmol/L H₂O₂. **i** ROS levels of KL TDCLs treated with 20 μmol/L H₂O₂ for 24 hours. For each clone, *n* = 3 replicates at 0 and 20 μmol/L H₂O₂. **j** Growth curve of KL allograft tumors from mice treated with or without high-dose Vitamin C (Vit C). *n* = 10 allograft tumors for each group. **k** Gross pathology of allograft tumors from (**j**). Scale bar = 1 cm. **l** Graph of allograft tumor weight from (**k**). **m** Representative IHC images and quantification of NRF2 (*n* = 15 images for each quantification), NQO1 (*n* = 10 images for each quantification), and 8-oxo-dG (*n* = 15 images for each quantification) of allograft tumors from (**k**). Scale bar = 100 μm. Data are presented as mean ± SEM, significance was calculated by two-tailed unpaired *t*-test (NADPH and NADP⁺ in **c**, GSSG and GSH/GSSG in **d**), two-tailed unpaired *t*-test with Welch's correction (NADPH/NADP⁺ in **c**, GSH in **d**), one-way ANOVA followed by Bonferroni's multiple comparisons test (**e, f, g, h, i, m**), or one-way ANOVA followed by *t*-test (**l**). Source data are provided as a Source Data file.

*G6pd^KO;KL* lung tumors via jugular vein for 12 hours (8:00 PM - 8:00 AM, fed state) (Fig. 5d) and assessed the fatty acid labeling from D₂O. Lower ²H labeling in C16:0 in *G6pd^KO;KL* lung tumors than in *G6pd^WT;KL* lung tumors was observed (Fig. 5e). As expected, no ²H labeling was detected in essential fatty acid C18:2 (Fig. 5f). Moreover, the pool size level of C16:0 was significantly lower in *G6pd^KO;KL* lung tumors than in *G6pd^WT;KL* lung tumors (Fig. 5g). Glucose provides carbon building blocks for de novo fatty acid synthesis. Using [U-¹³C₆]-glucose as a tracer, we observed significantly lower de novo fatty acid synthesis in *G6pd^KO;KL* TDCLs than in *G6pd^WT;KL* TDCLs (Supplementary Fig. 5a, b), which is consistent with ²H-labeled de novo lipogenesis in vivo (Fig. 5e). Therefore, the loss of G6PD impairs de novo lipogenesis in KL lung tumors.

We next performed lipidomics of lung tumors and serum from KL lung tumor-bearing mice. To investigate the fatty acyl composition in the lipids, we extracted and saponified the lipids to release the fatty acyl chains for the LC-MS analysis. We found that G6PD loss altered the fatty acyl composition in KL lung tumors when in fasted state (Fig. 5h, i), but not in fed state (Supplementary Fig. 5c, d). Compared with *G6pd^WT;KL* lung tumors, *G6pd^KO;KL* lung tumors had significantly lower pool size levels of long-chain fatty acyl groups, whereas very long-chain fatty acyl groups accumulated in *G6pd^KO;KL* lung tumors (Fig. 5i). However, this effect was not observed in the fed state or in serum (Fig. 5h, i, Supplementary Fig. 5c, d). Conversely, G6PD loss in KP lung tumors had no impact on fatty acyl composition in either state or in serum (Supplementary Fig. 6a–d).

Subsequently, we tested the hypothesis that lower de novo lipogenesis may lead to less availability of fatty acids for *G6pd^KO;KL* lung tumor growth by supplementing mice with high-fat diet (HFD) and examining tumor burden at 7- and 11- weeks post-tumor induction (Fig. 5j). HFD successfully rescued the KL lung tumor growth and cell proliferation (Ki67) caused by G6PD depletion (Fig. 5k–p). However, HFD did not rescue tumor number in mice bearing *G6pd^KO;KL* lung tumors (Fig. 5n), indicating that lower fat availability due to reduced de novo lipogenesis may not affect KL lung tumor initiation, but suppress tumor growth. Additionally, we conducted lipidomics of lung tumors and serum from tumor-bearing mice at 7-week post tumor induction in fasted state (Supplementary Fig. 7a). HFD significantly increased the levels of most fatty acyl groups in serum of mice bearing either *G6pd^WT;KL* or *G6pd^KO;KL* lung tumors (Supplementary Fig. 7b, c). However, HFD had no significant impact on fatty acid levels of *G6pd^WT;KL* lung tumors compared to normal diet (ND) (Supplementary Fig. 7d, e). Due to the minimal tumor burden of *G6pd^KO;KL* lung tumors at 7 weeks post-tumor induction, we were unable to collect *G6pd^KO;KL* lung tumors for lipidomics. Therefore, we compared the levels of fatty acids between *G6pd^WT;KL* and *G6pd^KO;KL* lung tumors under HFD condition. The levels of C16:0 is comparable between *G6pd^KO;KL* lung tumors and *G6pd^WT;KL* lung tumors under HFD condition (Supplementary Fig. 7e, f). However, the levels of many very long-chain fatty

acyl groups in *G6pd^KO;KL* lung tumors were lower than those in *G6pd^WT;KL* lung tumors under HFD condition (Supplementary Fig. 7e). This suggests that HFD partially rescue the alterations in levels of fatty acyl groups caused by G6PD loss. Thus, during KL lung tumorigenesis, G6PD-mediated oxPPP is essential for maintaining lipid metabolism for KL lung tumor growth.

## G6PD ablation does not affect TCA cycle metabolism but reprograms serine metabolism in KL lung tumors

Glucose contributes carbon to TCA cycle metabolites, PPP, and non-essential amino acids[6]. G6PD oxidizes the glycolytic intermediate glucose 6-phosphate (G6P) to 6-phosphogluconolactone. The loss of G6PD could potentially influence glucose metabolism. Therefore, we performed in vivo [U-¹³C₆]-glucose tracing and metabolic flux analysis in mice bearing *G6pd^WT;KL* or *G6pd^KO;KL* lung tumors to assess whether G6PD ablation in KL lung tumors has any impact on glucose metabolism beyond the oxPPP (Fig. 6a). We found that glucose carbon flux to KL lung tumor glycolytic intermediates, and TCA cycle metabolites and derivatives was not affected by G6PD ablation (Fig. 6b, c, Supplementary Fig. 8a, b). The pool size levels of glucose, G6P and 3-phosphoglycerate (3-PG) were significantly higher in *G6pd^KO;KL* lung tumors than *G6pd^WT;KL* lung tumors. However, the pool size levels of pyruvate, lactate and TCA cycle metabolites were comparable between *G6pd^WT;KL* and *G6pd^KO;KL* lung tumors (Fig. 6d, Supplementary Fig. 8c). In addition to the aforementioned observations, G6PD deficiency in KL lung tumors did not impact the pool size levels of other major metabolic intermediates, such as PPP intermediate ribose-5-phosphate, nucleotides, and others (Supplementary Fig. 8d). In contrast, in KP lung tumors, the pool size levels of TCA cycle intermediates and derivatives were significantly higher in *G6pd^KO;KP* lung tumors compared to *G6pd^WT;KP* lung tumors (Supplementary Fig. 9a, b). However, the pool size levels of other core metabolic metabolites were similar between *G6pd^WT;KP* and *G6pd^KO;KP* lung tumors (Supplementary Fig. 9a). Hence, the impact of G6PD deficiency on TCA cycle metabolism might differ between KL and KP lung tumors.

Glucose also contributes carbons for biosynthesis by incorporating glycolytic intermediates into different metabolic pathways (Fig. 6a). The synthesis of serine from glucose is a key metabolic pathway supporting cellular proliferation in healthy and malignant cells. We found that the ¹³C labeling of serine and glycine from glucose was significantly lower in *G6pd^KO;KL* lung tumors than in *G6pd^WT;KL* lung tumors, suggesting that G6PD ablation impairs serine biosynthesis (Fig. 7a). The reduction of glucose carbon flux to serine may contribute to the elevated levels of the 3-PG pool size level in *G6pd^KO;KL* lung tumors compared to *G6pd^WT;KL* lung tumors (Fig. 6d). However, the overall pool size levels of serine and glycine in KL lung tumors were not altered by G6PD ablation (Fig. 7b). This could be compensated by the upregulation of uptake or the reduction of catabolism. To distinguish these two possibilities, we first examined the serine consumption of KL TDCLs by measuring the

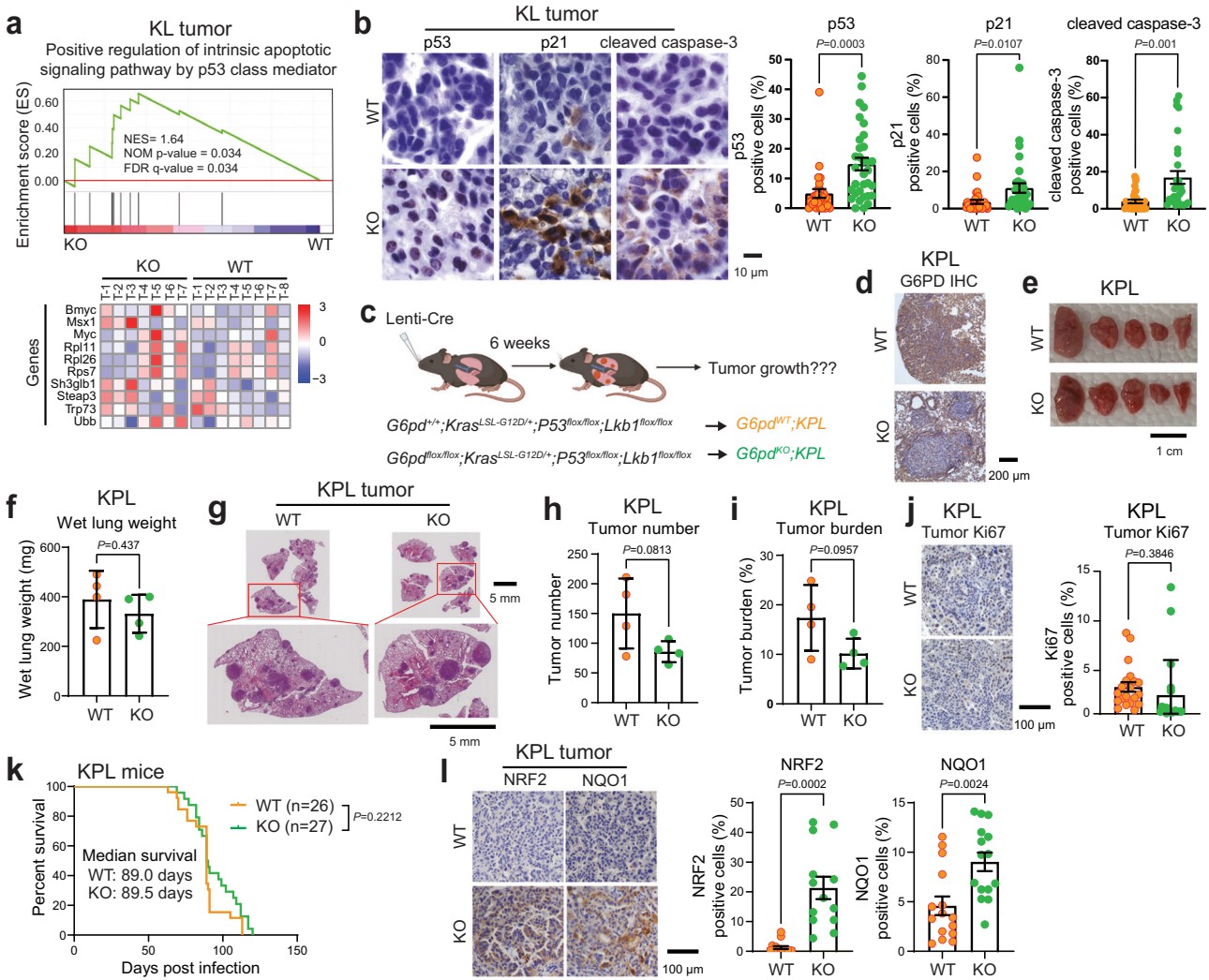

**Fig. 4 | G6PD suppresses p53 activation for KL lung tumorigenesis. a** GSEA (top) and heatmap of relative expression of genes (bottom) contributing to predicting positive regulation of intrinsic apoptotic signaling pathway by p53 class mediator for *G6pd^KO;KL* (*n* = 7 mice) and *G6pd^WT;KL* (*n* = 8 mice) lung tumors at 12 weeks post-tumor induction based on bulk-tumor mRNA-seq data. **b** Representative IHC images and quantification of p53 (*n* = 27 images for *G6pd^WT;KL*, *n* = 33 images for *G6pd^KO;KL*), p21 (*n* = 30 images for *G6pd^WT;KL*, *n* = 33 images for *G6pd^KO;KL*) and cleaved caspase-3 (*n* = 30 images for both *G6pd^WT;KL* and *G6pd^KO;KL*) of *G6pd^WT;KL* and *G6pd^KO;KL* lung tumors at 12 weeks post-tumor induction. Scale bar = 10 μm. **c** Scheme to induce conditional tumoral *G6pd* knockout to study the role of G6PD in KPL lung tumorigenesis (Created with BioRender.com released under a Creative Commons Attribution-NonCommercial-NoDerivs 4.0 International license). **d** Representative IHC images of G6PD in *G6pd^WT;KPL* and *G6pd^KO;KPL* lung tumors at 6 weeks post-tumor induction. *n* = 10 images for each genotype, scale bar = 200 μm. **e** Representative gross lung pathology from mice bearing *G6pd^WT;KPL* (*n* = 4

mice) and *G6pd^KO;KPL* (*n* = 4 mice) lung tumors at 6 weeks post-tumor induction. Scale bar = 1 cm. **f** Graph of wet lung weight from (**e**). **g** Representative H&E staining of scanned lung sections from (**e**). **h, i** Quantification of tumor number (**h**) and tumor burden (**i**) from (**g**). *n* is same with (**e**). **j** Representative IHC images and quantification of Ki67 in *G6pd^WT;KPL* and *G6pd^KO;KPL* lung tumors. *n* = 20 images for each quantification, scale bar = 100 μm. **k** Kaplan-Meier survival curve of mice bearing *G6pd^WT;KPL* (*n* = 26 mice) and *G6pd^KO;KPL* (*n* = 27 mice) lung tumors. **l** Representative IHC images and quantification of NRF2 (*n* = 15 images for *G6pd^WT;KPL*, *n* = 13 images for *G6pd^KO;KPL*) and NQO1 (*n* = 15 images for both *G6pd^WT;KPL* and *G6pd^KO;KPL*) in *G6pd^WT;KPL* and *G6pd^KO;KPL* lung tumors. Scale bar = 100 μm. Data are presented as mean ± SEM, significance was calculated by two-tailed unpaired *t*-test (**f**, **h**, **i**, NQO1 in **l**), two-tailed unpaired *t*-test with Welch's correction (**b**, **j**, NRF2 in **l**), or log-rank test (**k**). Source data are provided as a Source Data file.

reduction of serine in culture medium. We found that serine consumption in *G6pd^KO;KL* TDCLs was significantly higher than *G6pd^WT;KL* TDCLs (Fig. 7c). Subsequently, we assessed serine utilization via in vitro isotope tracing using [2,2,3-$^2$H]-serine and observed a higher serine $^2$H labeling in *G6pd^KO;KL* TDCLs than *G6pd^WT;KL* TDCLs (Fig. 7d), further demonstrating that G6PD deficiency reprograms KL lung tumor cell metabolism by upregulating serine uptake.

Next, we performed [2,2,3-$^2$H]-serine in vivo tracing in KL lung tumor-bearing mice (Fig. 7e, f) and observed significantly higher serine enrichment in *G6pd^KO;KL* lung tumors compared to *G6pd^WT;KL* lung tumors (Fig. 7g), confirming an increased serine uptake in vivo due to G6PD loss. Moreover, compared to *G6pd^WT;KL* lung tumors, a

significantly increased $^2$H labeling onto NADPH in *G6pd^KO;KL* lung tumors was observed (Fig. 7h).

Finally, we cultured *G6pd^WT;KL* and *G6pd^KO;K*L TDCLs in serine/glycine-free RPMI medium to determine the consequence of reprogrammed serine metabolism on cell proliferation. After 24 hours in serine/glycine-free medium, KL TDCLs exhibited significantly reduced pool size levels of serine and glycine (Supplementary Fig. 10 a–c). Moreover, serine/glycine depletion significantly decreased the NADPH pool size level and the ratio of NADPH/NADP$^+$ in *G6pd^KO;KL* TDCLs, while no such effect was observed in *G6pd^WT;KL* TDCLs (Fig. 7i, Supplementary Fig. 10d). *G6pd^WT;KL* TDCLs exhibited a trend of increased GSH/GSSG ratio, indicating an adaptive response to acute serine/glycine depletion

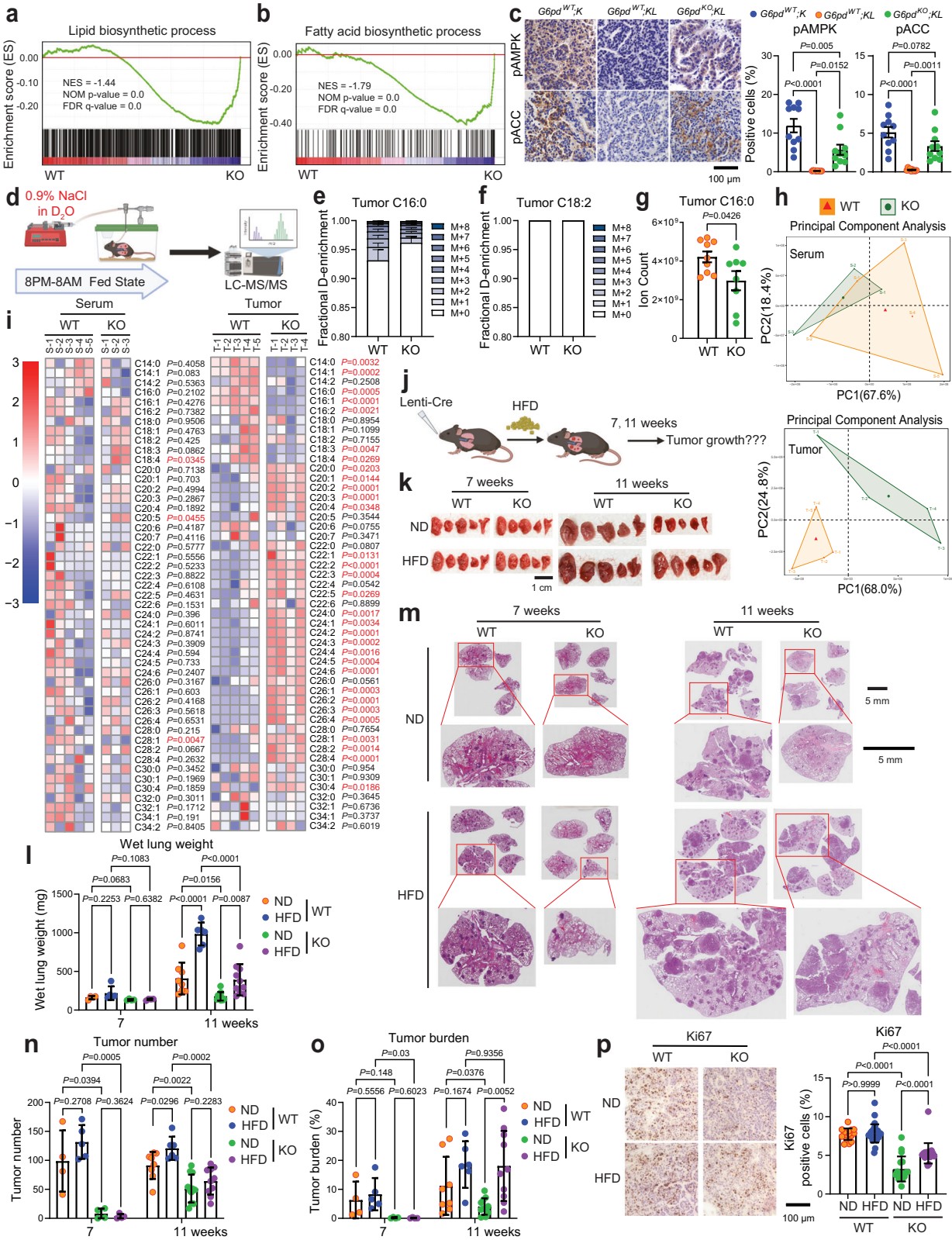

to maintain redox balance. However, the GSH/GSSG ratio in *G6pd^KO;KL* TDCLs remained unchanged, suggesting reduced adaptability in the absence of G6PD (Fig. 7j). As a result, increased ROS levels in *G6pd^KO;KL* TDCLs, but not in *G6pd^WT;KL* TDCLs, were observed by serine/glycine depletion (Fig. 7k). Furthermore, compared to *G6pd^WT;KL* TDCLs, *G6pd^KO;KL* TDCLs displayed higher sensitivity to serine/glycine depletion-induced cell death (Fig. 7l). Taken together, our findings

suggest that the reprogrammed serine metabolism resulting from G6PD loss could be used for NADPH generation, thereby maintaining redox homeostasis for cell proliferation.

## Discussion

Cellular pools of NADP(H) are compartmentalized[2]. In vitro studies have demonstrated that cytosolic and mitochondrial NADPH fluxes are

**Fig. 5 | G6PD depletion impairs KL lung tumor lipid metabolism. a**, **b** GSEA of lipid (**a**) and fatty acids (**b**) biosynthetic process for *G6pd^KO;KL* (*n* = 7 mice) and *G6pd^WT;KL* (*n* = 8 mice) lung tumors at 12 weeks post-tumor induction based on bulk-tumor mRNA-seq data. **c** Representative IHC images and quantification of pAMPK and pACC in *G6pd^WT;K* (*Kras^G12D/+*), *G6pd^WT;KL* and *G6pd^KO;KL* lung tumors at 12 weeks post-tumor induction. *n* = 10 images for each quantification. Scale bar = 100 μm. **d** Scheme of in vivo D₂O infusion to examine tumor de novo fatty acid synthesis (Created with BioRender.com released under a Creative Commons Attribution-NonCommercial-NoDerivs 4.0 International license). **e**, **f** C16:0 (**e**) and C18:2 (**f**) deuterium (²H) labeling fraction in *G6pd^WT;KL* (*n* = 9 mice) and *G6pd^KO;KL* (*n* = 8 mice) lung tumors at 12 weeks post-tumor induction. **g** C16:0 pool size of *G6pd^WT;KL* (*n* = 9 mice) and *G6pd^KO;KL* (*n* = 8 mice) lung tumors at 12 weeks post-tumor induction. **h**, **i** Principal Component Analysis (PCA) (**h**) and Heatmap (**i**) of saponified fatty acids pool size of *G6pd^WT;KL* and *G6pd^KO;KL* lung tumors (*n* = 5 mice for *G6pd^WT;KL*, *n* = 4 mice for *G6pd^KO;KL*) and serum (*n* = 5 mice for *G6pd^WT;KL*, *n* = 3 mice for *G6pd^KO;KL*) in fasted state at 12 weeks post-tumor induction. **j** Scheme to

examine the impact of high-fat diet (HFD) on KL lung tumorigenesis (Created with BioRender.com released under a Creative Commons Attribution-NonCommercial-NoDerivs 4.0 International license). **k** Representative gross lung pathology from mice bearing *G6pd^WT;KL* (*n* = 4 mice for 7 weeks in normal diet (ND), *n* = 5 mice for 7 weeks in HFD, *n* = 8 mice for 11 weeks in ND, *n* = 6 mice for 11 weeks in HFD) and *G6pd^KO;KL* (*n* = 4 mice for 7 weeks in ND, *n* = 4 mice for 7 weeks in HFD, *n* = 11 mice for 11 weeks in ND, *n* = 10 mice for 11 weeks in HFD) lung tumors fed with ND or HFD. Scale bar = 1 cm. **l** Graph of wet lung weight from (**k**). **m** Representative H&E staining of scanned lung sections from (**k**). **n**, **o** Quantification of tumor number (**n**) and tumor burden (**o**) from (**m**). *n* is same with (**k**). **p** Representative IHC images and quantification of Ki67 in *G6pd^WT;KL* (*n* = 17 images for ND, *n* = 29 images for HFD) and *G6pd^KO;KL* (*n* = 17 images for ND, *n* = 21 images for HFD) lung tumors at 11 weeks post-tumor induction. Scale bar = 100 μm. Data are presented as mean ± SEM, significance was calculated by two-tailed unpaired *t*-test (**g**, **i**), one-way ANOVA followed by Bonferroni's multiple comparisons test (**c**, **p**), two-way ANOVA followed by *t*-test (**l**, **n**, **o**). Source data are provided as a Source Data file.

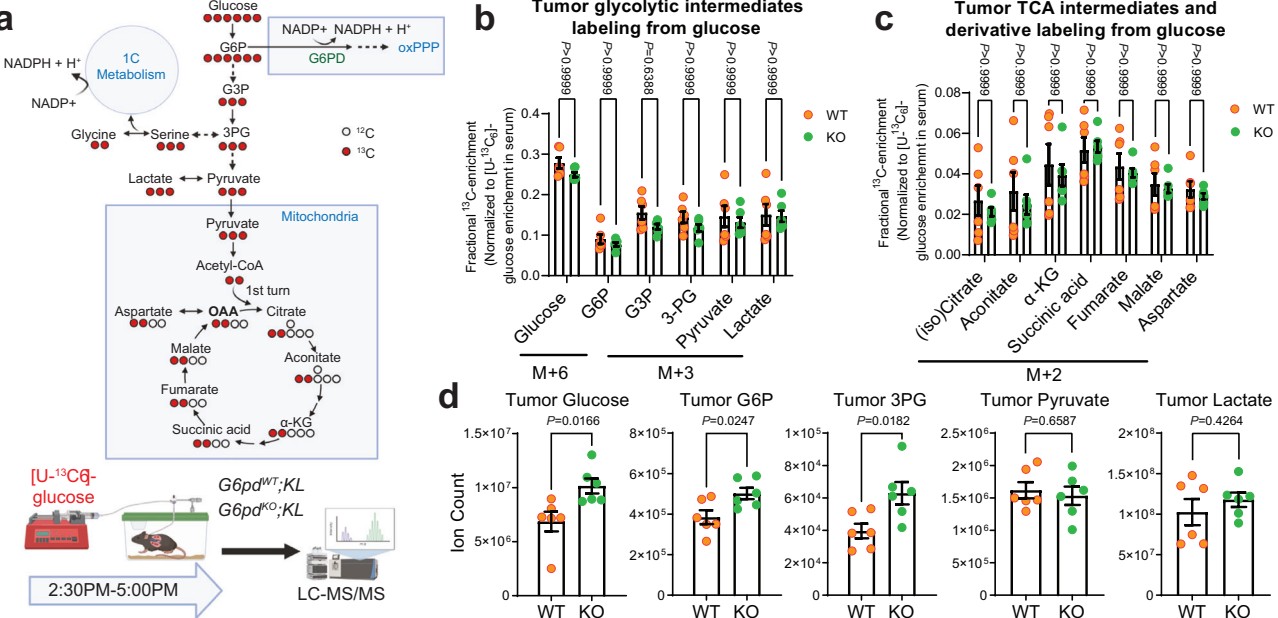

**Fig. 6 | G6PD ablation has no impact on KL lung tumor TCA cycle metabolism. a** Scheme of carbon contribution from glucose to glycolytic intermediates, TCA cycle intermediates, and serine (top) and in vivo [U-¹³C₆]-glucose tracing in the mice bearing *G6pd^WT;KL* and *G6pd^KO;KL* lung tumors (bottom). Schematic images are created with BioRender.com released under a Creative Commons Attribution-NonCommercial-NoDerivs 4.0 International license. **b** Normalized ¹³C labeling fraction of glycolytic intermediates from glucose of *G6pd^WT;KL* (*n* = 6 mice) and *G6pd^KO;KL* (*n* = 6 mice) lung tumors in fasted state (food was removed from the

mice at approximately 9:00 AM, and mice were euthanized with samples collected at 5:00 PM) at 12 weeks post-tumor induction. Glucose 6-phosphate (G6P), gly-ceraldehyde 3-phosphate (G3P), 3-phosphoglycerate (3-PG). **c** Normalized ¹³C labeling fraction of TCA cycle intermediates from glucose of tumors same with (**b**). α-ketoglutarate (α-KG). **d** Pool size of glycolytic intermediates of tumors same with (**b**). Data are presented as mean ± SEM, significance was calculated by two-tailed unpaired *t*-test (**d**), or two-way ANOVA followed by Bonferroni's multiple comparisons test (**b**, **c**). Source data are provided as a Source Data file.

independently and precisely regulated by multiple metabolic pathways[5,37]. G6PD-mediated oxPPP is one of the metabolic pathways involved in cytosolic NADPH generation. In this study, by using GEMMs of *KRAS*-driven NSCLC, we found that the dependence on G6PD is distinct in different subtypes of lung cancer. G6PD promotes KL but not KP lung tumorigenesis. Specifically, in KL lung tumors, G6PD-mediated oxPPP sustains the NADPH pool, crucial for maintaining redox balance and supporting lipid metabolism, and prevents p53 activation-induced cell death. Loss of G6PD in KL lung tumors triggers a shift in serine metabolism, increasing serine uptake to maintain one-carbon metabolism-driven NADPH production as an alternative. This, in turn, maintains redox homeostasis, facilitating the eventual pro-gression of G6PD-deficient KL lung tumors (Fig. 7m).

The differing dependency on G6PD in KL and KP lung tumor-igenesis can be attributed to the following factors. LKB1 serves as a

central modifier of cellular response to different metabolic stress. Loss of LKB1-AMPK signaling results in heightened sensitivity to energy depletion and to disturbances in redox homeostasis[38]. It is possible that KL lung tumors, which lack proper AMPK activity, exhibit a greater metabolic vulnerability and less plasticity in response to G6PD loss when compared to KP lung tumors that retain intact LKB1 function. In contrast, KP lung tumors can swiftly adapt to G6PD loss due to their functional LKB1-AMPK signaling, ensuring tumor survival. Specifically, the absence of G6PD has minimal impact on the NADPH/NADP⁺ and GSH/GSSG ratios in KP lung tumors, whereas these ratios are significantly altered in G6PD-deficient KL lung tumors. This suggests that KL lung tumors may inherently possess higher basal redox stress than KP lung tumors, rendering them more sensitive to disturbances in redox home-ostasis. In addition, clinical studies have suggested that lung cancer

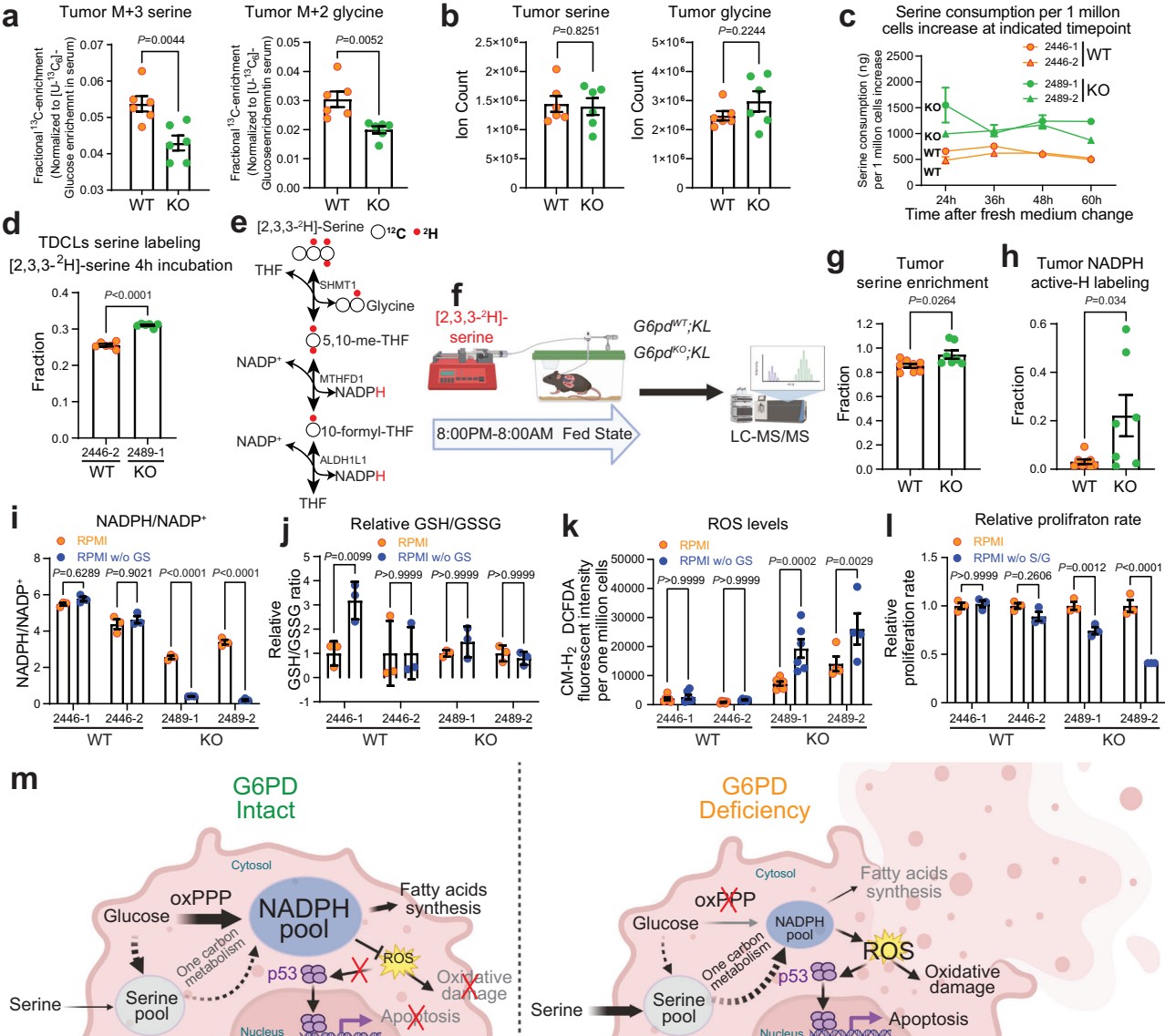

**Fig. 7 | G6PD ablation reprograms KL tumor serine metabolism. a** Normalized [13]C labeling fraction from glucose to serine and glycine of KL lung tumors. **b** Pool size of serine and glycine of KL lung tumors. **c** Serine consumption of *G6pd^WT;KL* and *G6pd^KO;KL* TDCLs in nutrient rich conditions. **d** [2]H labeling fraction of serine in *G6pd^WT;KL* and *G6pd^KO;KL* TDCLs after 4 hours [2,3,3-[2]H]-serine labeling in nutrient rich conditions. *n* = 6 replicates for each clone. **e** Scheme of hydrogen contribution from [2,3,3-[2]H]-serine to NADPH (Created with BioRender.com released under a Creative Commons Attribution-NonCommercial-NoDerivs 4.0 International license). **f** Scheme of in vivo [2,3,3-[2]H]-serine tracing (Created with BioRender.com released under a Creative Commons Attribution-NonCommercial-NoDerivs 4.0 International license). **g** [2]H labeling fraction of serine in *G6pd^WT;KL* (*n* = 8 mice) and *G6pd^KO;KL* (*n* = 7 mice) lung tumors at 12 weeks post-tumor induction. **h** NADPH active-H labeling from [2,3,3-[2]H]-serine in *G6pd^WT;KL* (*n* = 8 mice) and *G6pd^KO;KL* (*n* = 7 mice) lung tumors at 12 weeks post-tumor induction. **i** NADPH/NADP[+] ratio of *G6pd^WT;KL* and *G6pd^KO;KL* TDCLs cultured with RPMI medium with or without serine

and glycine for 24 hours. RPMI denotes cells cultured in complete RPMI medium, RPMI w/o GS denotes cells cultured in complete RPMI medium without serine and glycine. *n* = 3 replicates for each clone under different conditions. **j** Relative GSH/GSSG ratio of same samples from (**i**). **k** ROS levels of *G6pd^WT;KL* and *G6pd^KO;KL* TDCLs cultured with RPMI medium with or without serine and glycine for 48 hours. *n* = 6 replicates for each clone under different conditions, except 2489-2 with *n* = 4 replicates. **l** Relative proliferation rate of *G6pd^WT;KL* and *G6pd^KO;KL* TDCLs cultured with RPMI medium with or without serine and glycine for 48 hours. *n* = 3 replicates for each clone under different conditions. **m** Model of G6PD-mediated KL lung tumor growth (Created with BioRender.com released under a Creative Commons Attribution-NonCommercial-NoDerivs 4.0 International license). Data are presented as mean ± SEM, significance was calculated by two-tailed unpaired *t*-test (**a**, **b**, **d**, **g**, **h**), or two-way ANOVA followed by Bonferroni's multiple comparisons test (**i**, **j**, **k**, **l**). Source data are provided as a Source Data file.

patients with KL mutations are resistant to most cancer therapies, indicating increased aggressiveness compared to patients with KP mutations[22]. This increased aggressiveness of KL lung tumors has also been observed in preclinical mouse models[39]. Therefore, the enhanced aggressiveness in KL lung tumors could be attributed to increased proliferation, leading to a higher demand for NADPH and a greater dependence on G6PD-mediated oxPPP compared to KP lung tumors.

The p53 protein binds to G6PD and prevents the formation of the active dimer. p53 loss releases G6PD-inhibitory activity, potentially increasing PPP glucose flux in tumor cells[13]. However, we found that loss of G6PD-mediated oxPPP has no impact on KP lung tumorigenesis. To overcome G6PD loss, KP lung tumors may employ a strategy to boost NADPH production through alternative pathways like ME1, IDH1, or folate metabolism. This compensatory NADPH generation could also be accompanied by an alternative source of ribose-phosphate,

likely through the non-oxPPP. Additionally, KP lung tumors could obtain lipids and/or nucleosides from the surrounding microenvironment or bloodstream, thereby reducing their dependence on G6PD-derived products. Comprehensive mechanistic studies are needed to fully understand this resilience.

The role of G6PD in cancer development depends on its metabolic function in producing NADPH to reduce ROS and to support reductive biosynthesis[1,2]. We observed that G6PD deficiency has a substantial impact on NADPH levels in KL lung tumors. NADPH serves as a source of hydrogen and is crucial for fatty acid synthesis, which is essential for the growth and viability of NSCLC cells. LKB1 phosphorylates and activates AMPK, while AMPK can also be activated by CaMKKβ[19,31]. AMPK inhibits the activity of ACC to suppress de novo fatty acid synthesis and promote fatty acid oxidation[38]. An allosteric inhibitor of the ACC enzymes ACC1 and ACC2 markedly suppressed KL lung tumor growth[40]. These suggest the potential anti-tumorigenic role of AMPK. We observed that *G6pd^{WT};KL* lung tumors display lower AMPK activity compared to *G6pd^{WT};K* tumors, whereas *G6pd^{KO};KL* lung tumors exhibit heightened AMPK activity. The phosphorylation status of ACC supports this difference. Enhanced AMPK activity observed in *G6pd^{KO};KL* lung tumors may be attributed to CaMKKβ, and requires further investigation. Indeed, *G6pd^{KO};KL* lung tumors show reduced de novo lipogenesis, probably due to reduced hydrogen source from NADPH. Moreover, HFD supplementation rescued KL lung tumor growth caused by G6PD ablation, indicating that less fat availability due to reduced de novo fatty acid synthesis may contribute to the slower growth of *G6pd^{KO};KL* lung tumors. Fatty acyl groups composition in KL lung tumors was altered by G6PD loss at fasted state with a decrease in long-chain fatty acyl groups (C14, C16) and an accumulation of very long-chain fatty acyl groups (≥ C18) in *G6pd^{KO}; KL* lung tumors. The amount of long-chain and very long-chain fatty acids is intricately linked to various cellular processes, including de novo lipogenesis, dietary intake and elongation. Long-chain fatty acids have dual source--dietary intake and de novo synthesis, while very long-chain fatty acids come from both dietary sources and elongation[41]. The reduction in long-chain fatty acyl groups can be attributed to the reduction in de novo synthesis due to a decrease in NADPH generation caused by G6PD loss, as evidenced by in vivo $D_2O$ tracing and in *G6pd^{KO};KL* TDCLs through in vitro [U-$^{13}C_6$]-glucose labeling. Following the reduction in de novo synthesis, the very long-chain fatty acids from dietary sources accumulate, this phenomenon is in line with findings in other contexts where inhibition of endogenous de novo lipogenesis led to the accumulation of dietary very long-chain fatty acids[42]. Moreover, certain polyunsaturated very long-chain fatty acids are recognized for their antioxidant properties[43,44], and *G6pd^{KO};KL* lung tumors may favor the accumulation of polyunsaturated very long-chain fatty acids as a compensatory mechanism to counteract G6PD loss-induced oxidative stress. Various pathological conditions, including childhood adrenoleukodystrophy[45], Zellweger syndrome[46], and colorectal cancer[47], have been reported to exhibit the accumulation of very long-chain fatty acids. Further investigation is needed to understand how this composition change is associated with the slow tumor growth observed in the absence of G6PD. Moreover, despite the reduction in the de novo lipogenesis due to G6PD loss, the absorption of fatty acids from dietary sources in fed state might play an important role in maintaining the fatty acid levels in *G6pd^{KO};KL* tumors for tumor growth.

Cancer cells exhibit aberrant redox homeostasis. While ROS are pro-tumorigenic, high ROS levels are cytotoxic[30] and can be induced by oncogenic activity[48]. p53 influences the cellular redox balance by regulating several genes with antioxidant or pro-oxidant properties, which depends on various factors, including p53 protein levels[49]. Moderately elevated ROS levels inhibit p53, while higher levels promote its expression[49]. The loss of G6PD significantly increases oxidative stress in KL lung tumors, potentially leading to the activation of p53 and the upregulation of its downstream targets to impede tumor growth. Our findings reveal that the reduction of KL lung tumors by G6PD ablation is rescued by the absence of p53. Despite this, increased oxidative stress persists in *G6pd^{KO};KPL* lung tumors, indicating impaired redox homeostasis in KPL lung tumors due to G6PD deficiency. These observations suggest that oxidative stress alone, without p53, is not sufficient to impede KL lung tumor growth. Thus, the slow growth of G6PD-knockout KL lung tumors is attributed to p53 activation inhibiting tumor progression.

Cancer cells frequently alter metabolism to adapt to challenges. Inhibition of one cytosolic NADPH-producing metabolic pathway may lead to upregulation of others to compensate. Indeed, in vitro cell culture studies show that cancer cells can tolerate the loss of any two of the four canonical cytosolic NADPH production routes[50]. G6PD ablation in KL lung tumors did not completely inhibit the tumor growth, suggesting the presence of metabolic reprogramming or compensation during tumor progression. Our in vivo isotope tracing and flux analysis revealed that G6PD deficiency in KL lung tumors does not affect glucose carbon flux to tumor pyruvate, lactate, and TCA cycle intermediates. However, G6PD loss in KL lung tumors reduces glucose carbon flux to serine. Additionally, serine uptake is increased to maintain the serine pool size level in G6PD-deficient KL lung tumors for cytosolic NADPH production. We found that in in vitro cell culture, increased serine uptake is used to maintain redox homeostasis for cell proliferation. Therefore, serine-mediated one-carbon metabolism compensates for G6PD loss in KL cancer cell survival, although this does not preclude the potential compensatory cytosolic NADPH production through ME1 or IDH1. Indeed, in KEAP1 mutant KP lung tumor cells, G6PD loss triggered TCA intermediate depletion because of upregulation of the alternative NADPH-producing enzymes ME1/2 and IDH1/2[7]. However, in KP lung tumors, the depletion of G6PD resulted in an increase in the levels of TCA cycle intermediates. This emphasizes that oncogenic events play a crucial role in determining the dependence and associated mechanisms of distinct subtypes of *KRAS*-driven lung cancer on G6PD. This also indicates that the compensation for G6PD loss may involve mechanisms beyond the upregulation of ME1/2 and IDH1/2 alone in KP lung tumors. In the case of KL lung tumors, G6PD loss alters serine metabolism by decreasing serine biosynthesis and increasing serine uptake in KL tumors. The redox status has a significant impact on enzyme activity in various metabolic pathways, including those associated with serine metabolism. In the context of serine biosynthesis, 3-phosphoglycerate dehydrogenase (PHGDH) acts as a key enzyme, facilitating the conversion of 3PG to phosphohydroxypyruvate. The enzymatic activity of PHGDH is intricately connected to the NAD$^+$/NADH ratio[51,52]. G6PD deficiency observed in KL lung tumors has a significant impact on NADPH availability, disrupting redox equilibrium. This disruption could potentially affect NAD$^+$/NADH ratio and impair serine biosynthesis, resulting in the accumulation of 3PG. Simultaneously, during tumor progression, G6PD-deficient cells increase serine uptake to maintain serine-driven one-carbon metabolism as an alternative NADPH source. Hence, our study also proposes an innovative therapeutic approach for treating KL lung cancer by combining G6PD inhibitors with a serine/glycine depletion diet. However, in addition to its role in generating cytosolic NADPH, serine-mediated one-carbon metabolism is vital for nucleotide metabolism. Further mechanistic investigations are required to validate the effectiveness of this combination in in vivo cancer treatment.

G6PD has been proposed as a potential therapeutic target for cancer therapy in recent years due to its overexpression in various cancers[53]. G6PD inhibitors have been sought to achieve this goal[2,53,54]. According to our analysis of the cBioPortal datasets, high *G6PD* expression appears to be associated with poorer survival outcomes in lung cancer patients with co-mutations of *KRAS* and *LKB1*. Moreover, increased *MTHFD1* mRNA expression levels are linked to unfavorable survival outcomes in lung cancer patients with wild-type *KRAS*, except for those with co-mutations of *KRAS* and *LKB1*. This underscores the

significance of one-carbon metabolism in lung cancer. Additionally, while the mRNA expression of *ME1* is not statistically significantly linked to prognosis in patients with co-mutations of *KRAS* and *LKB1*, it appears to hold biological significance. Analyzing tumor mRNA expression is common in cancer research for prognostic insights. However, relying solely on this for prognosis may not always provide a comprehensive assessment due to factors like post-transcriptional modifications, tumor heterogeneity, microenvironmental influences, the dynamic nature of cancer, and treatment response. It's crucial to interpret mRNA data cautiously and integrate it with other information for a more thorough understanding of cancer prognosis. Therefore, the combination of cBioPortal data analysis with findings from our preclinical mouse study suggests that patients harboring co-mutations of *KRAS* and *LKB1* may benefit from G6PD inhibitor therapy.

Cancer cells exhibit greater sensitivity to the cytotoxic effects of oxidative stress when compared to normal cells. The pro-oxidant properties of high-dose Vit C, achieved through the generation of ROS including $H_2O_2$, make it a promising adjuvant in cancer treatment and has been explored in many pre-clinical and clinical studies[55–57]. Our observations reveal that *G6pd^{KO};KL* tumors are responsive to high-dose Vit C, resulting in tumor reduction. This suggests that when treating KL lung tumors with a G6PD inhibitor, incorporating high-dose Vit C as an adjuvant may be beneficial. Furthermore, exploring the potential therapeutic strategy of combining a G6PD inhibitor with agents that induce oxidative stress holds promise for treating this specific subtype of *KRAS*-mutant NSCLC.

While the discoveries from GEMMs are indeed exciting, it is essential to acknowledge a key distinction. In our GEMMs, LKB1, p53, and G6PD are completely depleted in *KRAS*-driven lung tumors at the initiation of tumor formation. In contrast, cancer patients gradually accumulate mutations in LKB1 and p53 over time, presumably upregulating G6PD expression. Furthermore, p53 mutation in patients may result in gain-of-function alterations, a complexity not fully reflected in current GEMMs. Although our GEMM findings highlight the role of G6PD in promoting KL, not KP, tumorigenesis, a deeper investigation is warranted to comprehend the implications of this discovery for the growth and treatment of KL or KP tumors in lung cancer patients.

## Methods

### Mice

All animal experiments were approved by the Institutional Animal Care and Use Committee (IACUC) of Rutgers University (Protocol number: PROTO999900099). Both male and female mice were used in this study. Wild type male C57BL6/J mice (Stock number: 008463) were obtained from the Jackson Laboratory. *G6pd^{+/+};Kras^{LSL-G12D};Lkb1^{flox/flox}* mice, *G6pd^{+/+};Kras^{LSL-G12D};P53^{flox/flox}* mice were generated in our previous study[58,59]. *G6pd^{flox/flox}* mice were generated by Rutgers Cancer Institute Genome Editing core facility. Mice were housed under a 12-hour light/dark cycle with 6 AM light on and 6 PM light off. The temperature was maintained between 21 and 24 °C and the humidity was between 30 and 70%. The sequences of primers for mouse genotyping are listed in Supplementary Table 2. Mice with same genotype or bearing same TDCLs were randomly assigned to different treatment groups. Sample sizes were chosen based on the power calculation. The investigators were blinded to the group allocation during experiments and when assessing outcomes.

For genetically engineered mouse models generation, *G6pd^{flox/flox};Kras^{LSL-G12D};Lkb1^{flox/flox}* mice were generated by cross-breeding *G6pd^{flox/flox}* mice with *G6pd^{+/+};Kras^{LSL-G12D};Lkb1^{flox/flox}* mice, *G6pd^{flox/flox};Kras^{LSL-G12D};P53^{flox/flox}* mice were generated by cross-breeding *G6pd^{flox/flox}* mice with *G6pd^{+/+};Kras^{LSL-G12D};P53^{flox/flox}* mice, *G6pd^{+/+};Kras^{LSL-G12D};P53^{flox/flox};Lkb1^{flox/flox}* mice were generated by cross-breeding *G6pd^{+/+};Kras^{LSL-G12D};Lkb1^{flox/flox}* mice with *G6pd^{+/+};Kras^{LSL-G12D};P53^{flox/flox}* mice, and *G6pd^{flox/flox};Kras^{LSL-G12D};P53^{flox/flox};Lkb1^{flox/flox}* mice were generated by cross-breeding *G6pd^{flox/flox}* mice with *G6pd^{+/+};Kras^{LSL-G12D};P53^{flox/flox};*

*Lkb1^{flox/flox}* mice. At 6-8 weeks of age, *G6pd^{+/+};Kras^{G12D};Lkb1^{-/-}* (*G6pd^{WT};KL*) lung cancer in *G6pd^{+/+};Kras^{LSL-G12D};Lkb1^{flox/flox}* mice, *G6pd^{-/-};Kras^{G12D};Lkb1^{-/-}* (*G6pd^{KO};KL*) lung tumor in *G6pd^{flox/flox};Kras^{LSL-G12D};Lkb1^{flox/flox}* mice, *G6pd^{+/+};Kras^{G12D};P53^{-/-}* (*G6pd^{WT};KP*) lung tumor in *G6pd^{+/+};Kras^{LSL-G12D};P53^{flox/flox}* mice, *G6pd^{-/-};Kras^{G12D};P53^{-/-}* (*G6pd^{KO};KP*) lung tumor in *G6pd^{flox/flox};Kras^{LSL-G12D};P53^{flox/flox}* mice, *G6pd^{+/+};Kras^{G12D};P53^{-/-};Lkb1^{-/-}* (*G6pd^{WT};KPL*) lung tumor in *G6pd^{+/+};Kras^{LSL-G12D};P53^{flox/flox};Lkb1^{flox/flox}* mice, and *G6pd^{-/-};Kras^{G12D};P53^{-/-};Lkb1^{-/-}* (*G6pd^{KO};KPL*) lung tumor in *G6pd^{flox/flox};Kras^{LSL-G12D};P53^{flox/flox};Lkb1^{flox/flox}* mice were induced by intranasally infection with Lenti-Cre (University of Iowa Viral Vector Core, Iowa-28) at $5 \times 10^6$ plaque-forming units (pfu) per mouse, following the methodology employed in our previous investigation[58].

Mice were fed with a regular chow diet (LabDiet, Cata#5058). For high-fat diet treatment, on the same day that *G6pd^{WT};KL* and *G6pd^{KO};KL* lung tumors were induced by intranasal infection with Lenti-Cre, half of mice were fed with the high-fat diet (Bio-Serv Mouse Diet, Cata#F3282) and the other half were fed with the control diet (normal diet) (Bio-Serv Mouse Diet, Cata#S4207). Following a 7-week and 11-week treatment period, the mice were euthanized, and lung tissues were collected for H&E staining, tumor number/burden quantification and IHC. In addition, after 7 weeks treatment of HFD, mice were euthanized, serum and lung tumors were collected for lipidomics analysis.

For TDCLs generation, *G6pd^{WT};KL* or *G6pd^{KO};KL* TDCLs were made from *G6pd^{WT};KL* or *G6pd^{KO};KL* lung tumors at 12 weeks post-tumor induction, respectively. TDCLs were cultured in complete RPMI medium (RPMI medium (Gibco, Cata#11875-093) supplemented with 10% fetal bovine serum (FBS), 1% Penicillin-Streptomycin, and 0.075% sodium bicarbonate) at 37 °C with 5% $CO_2$. Regular testing using the Universal mycoplasma detection kit (ATCC, Cata#30-1012k) confirmed the absence of mycoplasma contamination in the cell lines.

For allograft tumor induction and high-dose Vit C treatment, *G6pd^{WT};KL* or *G6pd^{KO};KL* TDCLs were subcutaneously injected into the right and left flank of male C57BL/6 mice with $1 \times 10^6$ cells/injection at 6–8 weeks of age. Then the mice bearing allograft tumors were administered Vit C at a dosage of 4 g/kg intraperitoneally (*i.p.*) daily for 2 weeks. Tumor size was measured using a caliper every other day during the 2-week treatment period. The tumor sizes were not exceeded the maximal tumor size (2000 mm³) permitted by the Institutional Animal Care and Use Committee of Rutgers University. After 2 weeks of treatment, the mice were euthanized, and tumors were collected and weighed for further analysis.

### Histology and IHC

Mice were euthanized via cervical dislocation at the designated time points following Lenti-Cre infection. Lung tissues were collected and placed in formaldehyde (Fisher Scientific, Cata#SF93-4) for a period of 12–24 hours. Afterward, the tissues were transferred to 70% ethanol solution and stored at 4 °C. Paraffin-embedded tissue sections were prepared using the methodology described in a previous study for histology and IHC[60]. For histology, the tissue sections were first deparaffinized using xylene and then rehydrated through a graded series of ethanol and water. Subsequently, the sections were stained with hematoxylin (Sigma, Cata#GHS216) and eosin (Sigma, Cata#1170811000), commonly referred to as H&E staining. Following the staining procedure, the sections were dehydrated and mounted onto slides using Cytoseal 60 mounting medium (Thermo Scientific, Cata#23-244256) for further microscopic examination. For IHC, the tissue sections were deparaffinized and rehydrated following the protocol for H&E staining. The sections were then heated at 95 °C in citrate buffer (Diagnostic Biosystems, Cata#K035) for 20 minutes. Subsequently, the sections were incubated with 3% hydrogen peroxide (Walgreens, Cata#715333) for 10 minutes to block endogenous peroxidase activity, followed by blocking in 10% goat serum (Fisher Scientific, Cata#16210064) for 1 hour at room temperature. The sections

were then incubated overnight at 4 °C with the anti-G6PD (Abcam, Cata#AB993, Clone#Polyclonal, Lot#GR274589-46, 1:2000 dilution, https://www.abcam.com/products/primary-antibodies/glucose-6-phosphate-dehydrogenase-antibody-ab993.html), anti-Ki67 (Abcam, Cata#ab15580, Clone#Polyclonal, Lot#GR3375556-1, 1:2000 dilution, https://www.abcam.com/products/primary-antibodies/ki67-antibody-ab15580.html), anti-pS6 (Cell Signaling, Cata#4858 S, Clone#D57.2.2E, Lot#21, 1:500 dilution, https://www.cellsignal.com/products/primary-antibodies/phospho-s6-ribosomal-protein-ser235-236-d57-2-2e-xp-rabbit-mab/4858), anti-P-p42/44 MAPK (pERK) (Cell Signaling, Cata#9101 S, Clone#NA, Lot#26, 1:500 dilution, https://www.cellsignal.com/products/primary-antibodies/phospho-p44-42-mapk-erk1-2-thr202-tyr204-antibody/9101), anti-cleaved caspase3 (Cell Signaling, Cata#9661 S, Clone#NA, Lot#47, 1:150 dilution, https://www.cellsignal.com/products/primary-antibodies/cleaved-caspase-3-asp175-antibody/9661), anti-p53 (Leica, Cata#NCL-L-p53-CM5p, Clone#POLYCLONAL, Lot#6065476, 1:2000 dilution, https://shop.leicabiosystems.com/us/ihc-ish/ihc-primary-antibodies/pid-p53-protein-cm5), anti-p21 (Santa Cruz Biotech, Cata#sc-6246, Clone#F-5, Lot#I1020, 1:1000 dilution, https://www.scbt.com/p/p21-antibody-f-5), anti-8-oxo-dG (R&D systems, Cata#4354-MC-050, Clone#15A3, Lot#P323432, 1:1000 dilution, https://www.rndsystems.com/products/8-oxo-dg-antibody-15a3_4354-mc-050), anti-γ-H2AX (Cell Signaling, Cata#9718, Clone#20E3, Lot#21, 1:1000 dilution, https://www.cellsignal.com/products/primary-antibodies/phospho-histone-h2a-x-ser139-20e3-rabbit-mab/9718), anti-NQO1 (Invitrogen, Cata#PA5-21290, Clone#AB_11153144, Lot#YL4152869, 1:1000 dilution, https://www.thermofisher.com/antibody/product/NQO1-Antibody-Polyclonal/PA5-21290), anti-NRF2 (Invitrogen, Cata#PA5-27882, Clone#AB_2545358, Lot#YF3956921A, 1:1000 dilution, https://www.thermofisher.com/antibody/product/Nrf2-Antibody-Polyclonal/PA5-27882), anti-pACC (S79) (Cell Signaling, Cata#3661, Clone#NA, Lot#10, 1:1000 dilution, https://www.cellsignal.com/products/primary-antibodies/phospho-acetyl-coa-carboxylase-ser79-antibody/3661), or anti-pAMPK (Cell Signaling, Cata#50081, Clone#D4D6D, Lot#6, 1:1000 dilution, https://www.cellsignal.com/products/primary-antibodies/phospho-ampka-thr172-d4d6d-rabbit-mab/50081) antibodies. The following day, the sections were incubated with biotin-conjugated secondary antibody for 30 minutes (Vector, Cata#BA-1000), horseradish peroxidase streptavidin for 10 minutes (Vector Laboratories, Cata#SA-5704-100) and developed by DAB (Agilent/Dako, Cata#K346811-2,) followed by hematoxylin staining. Sections were then dehydrated, mounted in Cytoseal 60 mounting medium for further analysis.

For the quantification of IHC for Ki67, pS6, pERK, cleaved caspase3, p53, p21, 8-oxo-dG, γ-H2AX, NQO1, NRF2, pACC, and pAMPK, more than 10 representative images from each group were obtained using a Nikon Eclipse 80i microscope and scored using the ImageJ (Version 1.52a) software.

### Tumor number/burden quantification

H&E-stained lung specimens were imaged using an Olympus VS120 whole-slide scanner (Olympus Corporation of the Americas) at 20 × magnification at the Rutgers Cancer Institute Biomedical Informatics shared resource. Image analysis was conducted using a custom-developed protocol on the Visiopharm image analysis platform (Visiopharm A/S). The protocol facilitated the identification of tissue area and the computation of tumor burden based on semiautomatically detected tumors. Low-resolution image maps, extracted from the whole-slide images, were utilized to generate tumor masks and whole-tissue masks. These masks were generated for each slide, enabling the segmentation of tumor burden ratios.

### D₂O, [U-¹³C₆]-glucose and [2,2,3-²H]-serine infusion

Before the infusion experiments, venous catheters were surgically implanted into the jugular veins of tumor-bearing mice, with a 3 to

4 days interval. The infusions were conducted on conscious, freely moving mice. For the infusion of $D_2O$ (Cambridge Isotope, Cata#DLM-4-50) and [2,2,3-²H]-serine (Cambridge Isotope, Cata#DLM-582-0.5), mice were fed continuously throughout the infusion period (8:00 PM - 8:00 AM). For the infusion of [U-¹³C₆]-glucose (Cambridge Isotope, Cata#CLM-1396-1), food was removed from the mice at approximately 9:00 AM, and infusion was commenced between 2:30 PM-5:00 PM. Mice were infused with $D_2O$ saline (0.9% NaCl) at a rate of 0.1 mL/g/minute, or [2,2,3-²H]-serine (200 mmol/L) at a rate of 0.2 mL/g/minute for 12 hours overnight, or [U-¹³C₆]-glucose (200 mmol/L) at a rate of 0.1 mL/g/minute for 2.5 hours before being euthanized for rapid lung tumors collection. Blood samples for serum analysis were collected from the mice's cheeks into 1.5 mL Eppendorf Tubes (Flex-Tubes, Cata#20901-551). Lung tumors were swiftly dissected and frozen using a liquid-nitrogen cold clamp to halt metabolic activity and then stored at −80 °C until further metabolites extraction.

### cBioPortal data processing

The overall survival analysis comparing the low and high expression levels of *G6PD*, *IDH1*, *ME1*, *MTHFD1*, and *NFE2L2* (*NRF2*) in lung cancer patients was conducted using the cBioPortal datasets[61] (https://www.cbioportal.org/, accessed on December 09, 2023). Data from 28 studies (as listed in Supplementary Table 1) available in the cBioPortal datasets were utilized for the present analysis.

For the overall survival analysis, the "gene specific" option was chosen, adding gene names including *G6PD*, *IDH1*, *ME1*, *MTHFD1*, and *NRF2*. The mRNA data type selected was "mRNA expression z-scores relative to all samples (log RNA Seq V2 RSEM)". A chart was then generated to compare the two groups based on the median expression of the indicated gene's mRNA. Subsequently, the overall survival was compared between the mRNA low expression group and the mRNA high expression group of the indicated gene.

For the analysis of mRNA expression levels of *G6PD*, *IDH1*, *ME1*, and *MTHFD1*, data were obtained from a 586 samples study on lung cancer (Lung Adenocarcinoma, TCGA, Firehose Legacy) available in the cBioPortal datasets (https://www.cbioportal.org/, accessed on December 09, 2023). Sample information for those with *KRAS/TP53* co-mutations and *KRAS/LKB1* co-mutations was extracted from the study, and the mRNA expression levels of the indicated genes were compared between these two groups. The mRNA expression levels were represented as mRNA expression z-scores relative to all samples (log RNA Seq V2 RSEM).

### mRNA-seq and GSEA analysis

*G6pd^{WT};KL* and *G6pd^{KO};KL* lung tumors were induced by intranasal infection with Lenti-Cre. At 12 weeks post-tumor induction, mice were euthanized by cervical dislocation. The lung tumors were rapidly dissected and snap-frozen in liquid nitrogen. Efforts have been made to collect the predominant portion of tumor tissues from each mouse lung. Subsequently, the frozen samples were pulverized to a powder using a Cryomill (Retsch). High-quality total RNA was extracted from the above samples, and mRNA enrichment were performed using RNeasy Min Kit (QIAGEN, Cata#74104). cDNA library was prepared and sequenced at Novogene.

For GSEA analysis, the gene set for "Oxidative stress" was downloaded from GeneCards (https://www.genecards.org/, accessed on April 09, 2023), and the gene sets for "GOBP positive regulation of intrinsic apoptotic signaling pathway by p53 class mediator", "GOBP lipid biosynthetic process" and "GOBP fatty acids biosynthetic process" were downloaded from mSigDB website (https://www.gsea-msigdb.org/, accessed on April 09, 2023). A dataset containing mRNA expression profiles of all genes for *G6pd^{WT};KL* and *G6pd^{KO};KL* lung tumors was prepared. The GSEA software (Version 4.3.2), using the classic setting recommended for mRNA-seq data in the GSEA manual, was employed to perform the GSEA analysis.

## Western blot

Western blot was performed as previously described[62]. Briefly, TDCLs protein samples were separated by SDS-PAGE and transferred onto PVDF membranes. The membranes were then blocked with 5% non-fat dry milk in TBST (Tris-buffered saline with 0.1% Tween 20) for 1 hour at room temperature, followed by incubation with primary antibody anti-G6PD (Abcam, Cata#AB993, Clone#Polyclonal, Lot#GR274589-46, 1:1000 dilution, https://www.abcam.com/products/primary-antibodies/glucose-6-phosphate-dehydrogenase-antibody-ab993.html) overnight at 4 °C. After washing, membranes were incubated with appropriate HRP-conjugated secondary antibody anti-β-actin (Sigma, Cata#A1978, Clone#AC-15, Lot#109M4849V, 1:100,000 dilution) for 1 hour at room temperature. Detection was carried out using ChemiDox Touch Imaging System (BIO-RAD). Uncropped scan of the Western blot for G6PD and β-actin is provided in the Source Data file.

## Cell proliferation assay

For IncuCyte measurement, $G6pd^{WT};KL$ or $G6pd^{KO};KL$ TDCLs were seeded at $4 \times 10^4$ cells per well in 12-well plates in complete RPMI medium. The IncuCyte live-cell imaging system automatically quantified cell surface area coverage to determine the percentage of confluence in one well of 12-well plate every 2 hours over 4 days, and the slope of the time-course changes in the percentage of confluence was utilized to reflect the proliferation rate.

For manual cell counting, cells were treated with $H_2O_2$ (Sigma-Aldrich, Cata#88597-100ML-F) at concentrations of 0, 20, 40, and 80 µmol/L for 24 hours. Subsequently, the cells were trypsinized off the culture plates and counted using a Vi-cell XR cell viability analyzer (Beckman coulter). The relative proliferation rate for cells treated with different concentrations of $H_2O_2$ was calculated by normalizing the cell number to the corresponding cells without $H_2O_2$ treatment.

## MTS assay

$G6PD^{WT};KL$ TDCLs were seeded at $2 \times 10^4$ cells per well in 96-well plates and $G6PD^{KO};KL$ TDCLs were seeded at $5 \times 10^4$ cells per well in 96-well plates. And 2 mg/mL MTS reagent (VWR, Cata#PAG1112) and 0.92 mg/mL PMS reagent were added to RPMI medium (0.2 mL of MTS reagent for per mL of RPMI and 0.01 mL of PMS reagent per mL of RPMI) to make MTS/PMS solution freshly before each assay. At the day of assay, each well was aspirated and 200 µL of MTS/PMS solution was added with minimal light exposure. An incubation period of 1 hour was performed before the first measurement. OD measurements were obtained at an excitement wavelength of 490 nm and were performed daily up to three days. The number of replicates in each group was specified in the figure legends.

For the experiments that utilized G6PDi-1 (Cayman, Cata#31484), the day following KL TDCLs seeding on 96-well plates, TDCLs were treated with vehicle control or G6PDi-1 at concentrations of 20 and 40 µmol/L, and MTS assay was performed at the indicated time points. The number of replicates in each group was specified in the figure legends. The relative proliferation rate for cells treated with different concentrations of G6PDi-1 was calculated by normalizing the cell number to the corresponding cells without G6PDi-1 treatment.

## Apoptosis/necrosis assay

$G6pd^{WT};KL$ and $G6pd^{KO};KL$ TDCLs were seeded in 96-well plates at $3 \times 10^4$ cells per well. Blank control wells contained culture mediums without cells. Complete RPMI medium kept at 37 °C was used to dilute detecting reagents from the Promega RealTime-Glo™ Annexin V Apoptosis and Necrosis Assay kit (Promega, Cata#JA1011) 1000-fold and added to each seeded well (100 µL) during measurement. Measurements were obtained at 22 and 46 hours after seeding. Luminescence measurements were obtained simultaneously to fluorescence which was optically measured at an excitement wavelength of 485 nm and collected at an emission wavelength of 530 nm for apoptosis. Fluorescence emissions were measured multiple times for each well and the mean values were used for data analysis for necrosis.

## De novo fatty acid synthesis analysis in vitro

$G6pd^{WT};KL$ and $G6pd^{KO};KL$ TDCLs were cultured in 6-cm dishes in RPMI medium without glucose (Gibco, Cata#11879-020) supplemented with 10% fetal FBS, 1% Penicillin-Streptomycin, 0.075% sodium bicarbonate, and 2 g/L [U-$^{13}C_6$]-glucose for 24 hours and assessed in triplicate. Afterward, saponified fatty acids were extracted and subjected to LC-MS analysis for further analysis and calculation of $^{13}$C labeling fraction for fatty acids.

## Serine consumption assay

$G6pd^{WT};KL$ or $G6pd^{KO};KL$ TDCLs were seeded at $0.5 \times 10^5$ or $1 \times 10^5$ cells per well in 24-well plates in complete RPMI medium, respectively. The following day, fresh complete RPMI medium was replaced, and medium was collected at 0, 24, 36, 48, and 60 hours. Each timepoint set up duplicate wells for both $G6pd^{WT};KL$ and $G6pd^{KO};KL$ TDCLs. The pool size levels of serine in medium were measured using LC-MS. A Vi-cell XR cell viability analyzer was used to measure cell number at each time point. Based on the following formula: the reduction in serine amount in the well (serine amount at the 0-hour timepoint minus the serine amount at the indicated timepoint) divided by the increase in cell number in the same well (cell number at the indicated timepoint minus the cell number at the 0-hour timepoint), the serine consumption (µg) per one million cells increase at the indicated timepoint can be calculated.

## Serine and glycine depletion assay

$G6pd^{WT};KL$ and $G6pd^{KO};KL$ TDCLs were cultured in customized complete RPMI medium (RPMI medium without glucose, serine, and glycine (Teknova, Cata#R9660), supplemented with 2 g/L glucose (Sigma, Cata#G8270-1KG), 10 mg/L glycine (Sigma, Cata#50046-50 G) and 30 mg/L serine (Sigma, Cata#S4311-25G)) with 10% fetal FBS, 1% Penicillin-Streptomycin, 0.075% sodium bicarbonate, at 37°C with 5% $CO_2$. After 2 days, $G6pd^{WT};KL$ TDCLs were trypsinized and seeded at $0.5 \times 10^5$ cells per well, while $G6pd^{KO};KL$ TDCLs were trypsinized and seeded at $1 \times 10^5$ cells per well in 24-well plates. For Serine and glycine depletion assay, the TDCLs were cultured in the serine/glycine free RPMI medium (RPMI medium without glucose, serine and glycine, supplemented with 2 g/L glucose) with 10% fetal FBS, 1% Penicillin-Streptomycin, 0.075% sodium bicarbonate, and the complete RPMI medium as control. After 2 days, TDLCs were trypsinized off plates and then counted using a Vi-cell XR cell viability analyzer.

## ROS levels measurement

The CM-$H_2$DCFDA assay (Invitrogen, Cata#C6827) was performed to measure cellular ROS levels. $G6pd^{WT};KL$ or $G6pd^{KO};KL$ TDCLs were seeded at $0.5 \times 10^5$ or $1 \times 10^5$ cells per well in 24-well plates in complete RPMI medium, respectively. After 2 days, the cells were washed twice with HBSS (Corning, Cata#21-022-CV). Then, 0.5 mL of CM-$H_2$DCFDA solution with a concentration of 5 µmol/L in HBSS was added to each well, and the cells were incubated at 37 °C for 45 minutes in the dark. Following incubation, the medium was changed to RPMI medium (Gibco, Cata#11875-093) for 30 minutes to allow for recovery. Subsequently, the medium was replaced with HBSS, and the fluorescence intensity was measured using a microplate reader (Tecan). The excitation wavelength was set to 493 nm, and the emission wavelength was set to 520 nm. After measuring the fluorescence intensity, TDLCs were trypsinized off the plates and counted using a Vi-cell XR cell viability analyzer. The ROS levels were calculated using the following formula: the fluorescence intensity (the fluorescence intensity of each well stained with CM-$H_2$DCFDA minus the fluorescence intensity without staining) was divided by the cell number (x $10^6$) in the same well.

For the ROS levels measurement under the $H_2O_2$ treatment condition, cells were treated with $H_2O_2$ at concentrations of 0, 20 μmol/L for 24 hours. Subsequently, the cells were stained with CM-$H_2$DCFDA following the aforementioned method.

To measure ROS levels under serine and glycine depletion conditions, cells were treated according to the "serine and glycine depletion assay" method. Subsequently, the cells were stained with CM-$H_2$DCFDA following the aforementioned method.

### Serine uptake measurement in vitro

$G6pd^{WT};KL$ and $G6pd^{KO};KL$ TDCLs were seeded in 6-cm dishes with regular complete RPMI medium for serine uptake measurement. The following day, the medium was replaced with the RPMI medium without glucose, serine, and glycine (Teknova, Cata#R9660), supplemented with 10% FBS, 1% penicillin-streptomycin, 0.075% sodium bicarbonate, 2 g/L glucose, 10 mg/L glycine, and 30 mg/L [2,2,3-$^2$H]-serine. The cells were incubated for 4 hours, and the experiment was performed in triplicate. Subsequently, water-soluble metabolites were extracted and subjected to LC-MS analysis to further analyze and calculate $^2$H labeling for serine.

### Sample preparation of water-soluble metabolites for LC-MS analysis

For the extraction of water-soluble metabolites from lung tumors, following the methodology described in a previous study[62]. Approximately 20–30 mg of tumor samples were precisely weighed and placed into a pre-cooled tube. The samples were then pulverized using the Cryomill. Pre-cooled extraction buffer consisting of methanol: acetonitrile: $H_2O$ (40:40:20, V/V) with 0.5% formic acid (Sigma-Aldrich, Cata#F0507-100ML) was added to the resulting powder (40 μL of solvent per mg of tumors). The samples were then vortexed for 15 seconds and incubated on ice for 10 minutes. Subsequently, 15% $NH_4HCO_3$ solution (5% V/V of the extraction buffer) was used to neutralize the samples. Then all samples were vortexed again for 10 seconds and centrifuged at 4 °C, 13,000 × $g$ for 20 minutes. The resulting supernatant was transferred to LC-MS vials for subsequent analysis.

For the extraction of water-soluble metabolites from serum, following the methodology described in a previous study[62], pre-cooled methanol was added to the serum samples in a 1.5 mL Eppendorf Tube (Tube A). The mixture was vortexed for 10 seconds and left at −20 °C for 20 minutes. Afterward, the tube was centrifuged at 4 °C for 10 minutes. The top supernatant was carefully transferred to another 1.5 mL Eppendorf Tube (Tube B) as the first extract. Then, 200 μL of an extraction buffer composed of methanol: acetonitrile: $H_2O$ (40:40:20, V/V) was added to Tube A, which still contained the pellet at the bottom. The tube was vortexed for 10 seconds and placed on ice for 10 minutes. Subsequently, the tube was centrifuged at 4 °C, 13,000 × $g$ for 10 minutes. The top supernatant from this step was combined with the first extract in Tube B, resulting in a final extract volume of 250 μL. The extract was loaded onto the Phree Phospholipid Removal Tabbed 1 mL tube (Phenomenex, Cata#8B-S133-TAK) and centrifuged at 4 °C, 13,000 × $g$ for 10 minutes. The extract was transferred to LC-MS vials for subsequent analysis.

For the extraction of water-soluble metabolites from cultured cell samples, following the methodology described in a previous study[58], $G6pd^{WT};KL$ and $G6pd^{KO};KL$ TDCLs cultured in triplicate in 6-cm dishes were washed twice with PBS. Subsequently, the cells were incubated with 1 mL of pre-cooled extraction buffer containing methanol: acetonitrile: $H_2O$ in a ratio of 40:40:20, along with 0.5% formic acid solution, on ice for 5 minutes. The extraction process was followed by neutralization with 50 μL of 15% ammonium bicarbonate. The cells were then scraped from the plates and transferred to 1.5 mL tubes. Afterward, the tubes were centrifuged at 4 °C, 13,000 × $g$ for 10 minutes. The resulting supernatant was transferred to LC-MS vials for subsequent analysis.

### Sample preparation of saponified fatty acids for LC-MS analysis

To extract saponified fatty acids from lung tumors and serum, samples were collected under two states, as indicated in the figure legend: fed state and fasted state. In fed state, the food was available in cages, and mice were euthanized with samples collected at 8:00 AM. In fasted state, food was removed from the mice at approximately 9:00 AM, and mice were euthanized with samples collected at 3:00 PM. The extraction methodology described in a previous study was followed[35]. Pre-cooled methanol was added to the resulting powder or serum (12 μL of methanol per mg of lung tumors or μL of serum). The samples were vortexed for 10 seconds, followed by adding −20 °C MTBE (40 μL of MTBE per mg of tumors or μL of serum). After another 10 seconds vortexing step, the samples were incubated on a shaker at 4 °C for 6 minutes. Next, $H_2O$ (10 μL of $H_2O$ per mg of tumors or μL of serum) was added, and the samples were centrifuged at 4 °C, 13,000 × $g$ for 2 minutes. Following centrifugation, 160 μL of the top MTBE layer was transferred to a new 1.5 mL Eppendorf Tube. The liquid was then dried by nitrogen. Subsequently, the sample was resuspended in 1 mL of saponification solvent (0.3 mol/L KOH in 90:10 methanol/$H_2O$), and the entire volume was transferred to 4 mL glass vials. The vials were placed in a water bath at 80 °C for 1 hour. After incubation, the vials were cooled on ice for 3 minutes, and 100 μL of formic acid was added. Then, 300 μL of hexanes was added, and the samples were vortexed for 10 seconds, resulting in two layers. The top layer was transferred to a new 1.5 mL Eppendorf Tube, and this step was repeated to obtain a final volume of 600 μL. The liquid was then dried in the 1.5 mL Eppendorf Tube under air. To resuspend the extracted sample, 150 μL of resuspension solvent (50:50 isopropanol/methanol) was added. The samples were centrifuged at 4 °C, 13,000 × $g$ for 10 minutes, and the resulting supernatant was transferred to LC-MS vials for further analysis.

For the extraction of saponified fatty acids from cultured cell samples, following the methodology described in a previous study[58], $G6pd^{WT};KL$ and $G6pd^{KO};KL$ TDCLs cultured in triplicate in 6-cm dishes were washed twice with PBS, followed by the addition of 1 mL of −20 °C 90% methanol containing 0.3 mmol/L KOH. The resulting liquid, along with the cell debris, was scraped into 4 mL glass tubes. The samples were then heated at 80 °C for 1 hour to saponify the fatty acids. After saponification, the samples were acidified with 100 μL of formic acid, followed by 1 minute of vortexing. The samples were extracted twice with 1 mL of hexane, and the organic phase was collected. The extracts were dried under air and dissolved in a mixture of methanol, chloroform, and isopropanol in a 1:1:1 ratio. All samples were vortexed for 20 seconds and then centrifuged at 4 °C, 13,000 × $g$ for 10 minutes. The resulting supernatant was transferred to LC-MS vials for subsequent analysis.

### LC-MS methods

For the LC-MS analysis of water-soluble metabolites, following the methodology described in a previous study[63], the experimental conditions were optimized using an HPLC-ESI-MS system consisting of a Thermo Scientific Vanquish HPLC coupled with a Thermo Q Exactive Plus MS. The HPLC system was equipped with a Waters XBridge BEH Amide column (2.1 mm × 150 mm, 2.5 μm particle size, 130 Å pore size) and a Waters XBridge BEH XP VanGuard cartridge (2.1 mm x 5 mm, 2.5 μm particle size, 130 Å pore size) guard column. The column temperature was maintained at 25 °C. The mobile phase A consisted of a mixture of $H_2O$: acetonitrile (95:5, V/V) with 20 mmol/L $NH_3AC$ and 20 mmol/L $NH_3OH$ at pH 9. The mobile phase B consisted of a mixture of acetonitrile: $H_2O$ (80:20, V/V) with 20 mmol/L $NH_3AC$ and 20 mmol/L $NH_3OH$ at pH 9. The composition of mobile phase B varied over time as follows: 0 minutes, 100%; 3 minutes, 100%; 3.2 minutes, 90%; 6.2 minutes, 90%; 6.5 minutes, 80%; 10.5 minutes, 80%; 10.7 minutes, 70%; 13.5 minutes, 70%; 13.7 minutes, 45%; 16 minutes, 45%; 16.5 minutes, 100%. The flow rate was set to 300 μL/minute, and the injection

volume was 5 µL. The column temperature was maintained at 25 °C throughout the analysis. MS scans were acquired in negative ionization mode, with a resolution of 70,000 at m/z 200. The automatic gain control (AGC) target was set to $3 \times 10^6$, and the mass-to-charge ratio (m/z) scan range was set from 72 to 1000 [35].

For the LC-MS analysis of NADPH and NADP$^+$, the gradient consisted of the following steps: 0 minutes, 85% B; 2 minutes, 85% B; 3 minutes, 60% B; 9 minutes, 60% B; 9.5 minutes, 35% B; 13 minutes, 5% B; 15.5 minutes, 5% B; 16 minutes, 85% B. The run was stopped at 20 minutes, and the injection volume was 15 µL. As described previously, full scans were alternated with targeted scans in the m/z range of 640-765, with a resolution of 35,000 at m/z 200, and with AGC target of $5 \times 10^5$.

For the LC-MS analysis of fatty acids samples, following the methodology described in a previous study[58], a Vanquish Horizon UHPLC system (Thermo Fisher Scientific, Waltham, MA) with a Poroshell 120 EC-C$_{18}$ column (150 mm × 2.1 mm, 2.7 µm particle size, Agilent Infinity Lab, Santa Clara, CA) was employed using a gradient of solvent A (90%:10% H$_2$O: methanol with 34.2 mmol/L acetic acid, 1 mmol/L ammonium acetate, pH 9.4), and solvent B (75%:25% IPA: methanol with 34.2 mmol/L acetic acid, 1 mmol/L ammonium acetate, pH 9.4). The gradient program was as follows: 0 minutes, 25% B; 2 minutes, 25% B; 5.5 minutes, 65% B; 12.5 minutes, 100% B; 19.5 minutes, 100% B; 20 minutes, 25% B; 30 minutes, 25% B. The flow rate was set to 200 µL/minute, and the column temperature was maintained at 55 °C. MS/MS data were acquired using a Thermo Q Exactive PLUS mass spectrometer with heated electrospray ionization source. The spray voltage was set to −2.7 KV in negative mode. The sheath gas, auxiliary gas, and sweep gas flow rates of 40, 10, and 2 (arbitrary unit), respectively. The capillary temperature was set to 300 °C, and the auxiliary gas heater was set to 360 °C. The S-lens RF level was 45. In negative ionization mode, the m/z scan range was set from 200 to 2,000. The AGC target was set to $1 \times 10^6$ and the maximum injection time was 200 milliseconds. The resolution was set to 140,000. Data-dependent MS/MS scans were acquired from pooled samples in negative ionization mode with a loop count of 3, an AGC target of $1 \times 10^6$, and the maximum IT of 50 milliseconds. The mass resolution was set to 17,500, and the normalized collision energy was stepped at 20, 30, and 40. Dynamic exclusion was set to 10 seconds. The MS/MS data were processed using MS-DIAL[64,65], and the fatty acids species annotations were performed by matching the built-in MS/MS database. The annotated fatty acids species were quantified from MS1 runs for better accuracy[66] using EI-MAVEN[67,68].

## NADPH active-H labeling calculation

NADPH and NADP$^+$ features were extracted in EI-MAVEN software. The $^2$H isotope natural abundance and impurity of labeled substrate were corrected using AccuCor written in R[69]. NADPH active-H labeling ***p*** from [2,3,3-$^2$H]-serine was determined based on labeling of NADPH-NADP$^+$ pair and calculated using the previously described Eq. (1)[35], the matrix on the left side of Eq. (1) contains the experimentally measured mass $^2$H isotope distribution for NADP$^+$, while the right side contains the experimentally measured mass $^2$H isotope distribution for NADPH.

$$\begin{pmatrix} M+0 & 0 \\ M+1 & M+0 \\ M+2 & M+1 \\ \cdot & \cdot \\ \cdot & \cdot \\ \cdot & \cdot \\ M+i & M+(i-1) \\ 0 & M+i \end{pmatrix} \times \begin{pmatrix} 1-p \\ p \end{pmatrix} = \begin{pmatrix} M+0 \\ M+1 \\ M+2 \\ \cdot \\ \cdot \\ \cdot \\ M+i \\ M+(i+1) \end{pmatrix} \quad (1)$$

## Water-soluble metabolites and fatty acids labeling fraction calculation

Water-soluble metabolites and fatty acids data were obtained using the EI-MAVEN software package[68] with each labeled isotope fraction. The isotope natural abundance and tracer isotopic impurity were corrected using AccuCore written in R[69]. Additionally, the labeling fraction and enrichment was calculated automatically by the AccuCor[69]. Fractional $^{13}$C-enrichment of indicated water-soluble metabolite in the tumor was calculated by dividing the labeling fraction of the indicated water-soluble metabolite from tumor by [U-$^{13}$C$_6$]-glucose enrichment in the serum from same mouse.

## Statistics and reproducibility

GraphPad Prism 9.1.0 (GraphPad Software Inc., La Jolla, CA) was employed for data analysis. The normal distribution of variables was estimated using Shapiro-Wilk (W) test, while Friedman's test was utilized to assess the homogeneity of variance. The data underwent analysis using appropriate statistical tests, including the two-tailed unpaired *t*-test, two-tailed unpaired *t*-test with Welch's correction, Mann-Whitney test, one-way and two-way ANOVA followed by *t*-test, one-way ANOVA followed by *t*-test with Welch's correction, and one-way ANOVA followed by Bonferroni's multiple comparisons test, as specified in the figure legends. The log-rank test determined the significance of Kaplan-Meier analyses for survival. Data were presented as the mean ± SEM, and $P < 0.05$ was considered statistically significant. GSEA statistical test was performed by GSEA software (Version 4.3.2), gene set with |NES| > 1, NOM *p*-value < 0.05, FDR *q*-value < 0.25 was considered as significant. All experiments were repeated independently three times with similar results.

## Reporting summary

Further information on research design is available in the Nature Portfolio Reporting Summary linked to this article.

## Data availability

The raw data of mRNA-seq have been deposited in the Gene Expression Omnibus (GEO) database under accession number GSE253613. Additionally, full list of RNA-seq result is provided in Supplementary Data 1. The metabolomics and lipidomics raw data have been deposited in Metabolomics Workbench under project ID PR001843. Source data are provided with this paper.

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

## Acknowledgements

We are grateful to Eric Chiles and Yujue Wang in the Xiaoyang Su laboratory at Rutgers Cancer Institute for their assistance with LC-MS performance, and Jianming Wang in the Wenwei Hu laboratory at Rutgers Cancer Institute for providing p53 and p21 antibodies for IHC. This work was supported by National Institute of Health (NIH) grants R01CA237347 and R21CA263136, American Cancer Society grant 134036-RSG-19-165-01-TBG, GO2 Foundation for Lung Cancer, and Ludwig Princeton Branch of the Ludwig Institute for Cancer Research to J.Y.G.; NIH grant R01CA163591 and Ludwig Princeton Branch of the Ludwig Institute for Cancer Research to E.W.; NJCCR postdoc fellowship COCR23PDF004 to W.W.; NIH grant P30 CA072720 to Rutgers Cancer Institute (Metabolomics Shared Resource, Biomedical Informatics shared resource, and Biospecimen Repository Service shared resource, at Rutgers Cancer Institute).

## Author contributions

J.Y.G. was the lead principal investigator who conceived and supervised the project. J.Y.G. and T.L. designed the experiments, performed the data analysis, and interpreted the data. J.M.G. and J.D.R. provided *G6pd*^*flox/flox* mouse strain, T.L., S.A., J.L., H.K., W.W., S.W., E.C.L., M.S., and V.B. performed most experiments. S.W. and M.S. performed IHC quantification. S.A., S.W., X.L., and J.L. maintained mouse colonies and mouse genotyping. Z.H. performed mouse surgery. X.S. provided technical support for metabolomics and lipidomics measurements and analyses. J.D.R, and E.W. provided intellectual input in project development. J.Y.G. and T.L. wrote the manuscript that was reviewed and edited by all authors.

## Competing interests

E.W. is a stockholder in a founder of Vescor Therapeutics. J.D.R. is an advisor and stockholder in Colorado Research Partners, L.E.A.F. Pharmaceuticals, Bantam Pharmaceuticals, Rafael Pharmaceuticals; a paid consultant of Third Rock Ventures; a founder, director and stockholder of Farber Partners, Serien Therapeutics and Sofro Pharmaceuticals; a founder and stockholder in Empress Therapeutics; and a director of the Princeton University–PKU Shenzhen collaboration. The Rabinowitz lab at Princeton University and the Princeton University-PKU Shenzhen collaboration have discovered and generated intellectual property regarding G6PD inhibitors. Other authors have no conflict of interest to declare.
