## [Peer Review File · Nature Communications]

G6PD Maintains Redox Homeostasis and Biosynthesis in LKB1-Deficient KRAS-Driven Lung CancerREVIEWER COMMENTS

Reviewer #1 (Remarks to the Author):

In the manuscript entitled "G6PD Maintains Redox Homeostasis and Biosynthesis in LKB1-Deficient KRAS-Driven Lung Cancer" Lan et al investigated the distinct dependency of different subtypes of KRAS-driven NSCLC on the pentose phosphate pathway (PPP) enzyme G6PD. To this aim, they utilized genetically engineered mouse models lacking G6PD in the context of *Kras*G12D/+; *p53*^{-/-} (KP), *Kras*G12D/+; *Lkb1*^{-/-} (KL) or *Kras*G12D/+; *p53*^{-/-}; *Lkb1*^{-/-} (KPL), as well as tumour-derived cell lines (TDCLs). They demonstrated that G6PD is indispensable for KL but not for KP lung tumorigenesis as indicated by an extension of the mouse lifespan and a substantial reduction in the KL lung tumour number, burden, and proliferation. The authors suggest that G6PD loss in KL lung cancer is associated with a disruption of cellular redox homeostasis via suppression of NADPH generation resulting in oxidative stress, p53-dependent apoptosis, and growth arrest. Additionally, the decrease in NADPH affects de novo lipogenesis in KL lung tumours, which in turn inhibits tumour growth. Finally, the authors propose an increase in serine uptake as an alternative cytosolic NADPH-producing metabolic pathway in order to maintain redox homeostasis in G6PD-deficient KL tumours.

Although potentially interesting, this study appears to lack a clear rationale and scope, and logic in the selection and presentation of the experiments. At times, it appears to be an exercise of mouse genetics without an underlying question. Moreover, the work harbours major technical and conceptual limitations that question the conclusions and translational potential of the work. Finally, the role of G6PD in lung cancer driven by oncogenic KRAS has been previously shown (by the same authors), diminishing the overall novelty and impact of the work.

Major critiques:

1. The mechanistic part of the paper addresses the causes of the vulnerability of the KL tumours upon G6PD deficiency and branches in three distinct directions: redox stress, decreased lipogenesis, and increased serine uptake. However, these parts remain mechanistically disconnected, limiting the overall relevance of these findings and depth of investigation. For example, as supported by the data in figure 4i, G6PD deficiency significantly reduces allograft tumor growth but the clinical relevance of the shown vitamin C treatment is questionable. Do the authors suggest a combination of vitamin C as an adjuvant agent together with G6PD inhibitors for the patients harbouring KRAS and LKB1 co-MUT? If so, this has to be clearly stated in the discussion.

The same argument applies to the essentiality of serine uptake in this model (figure 7). It is unclear why the authors focused on this pathway and the significance these results have in finding novel patient therapies. The authors should clarify this point and discuss it better.

Along the same line, the lipogenesis part seems disconnected from the main message of the paper and superficial. To strengthen the hypothesis that HFD contributes to lung tumour growth in G6PD-deficient animals, the authors should perform a lipidomic analysis in serum and tumour from the mice fed with HFD side by side with the ND to check for differences in the abundance, saturation, and elongation of the fatty acids. The authors should more thoroughly assess the connection between changes in NADPH and lipogenesis, for instance, by overexpressing cytosolic TPNOX (Cracan et al 2017) to test whether reduced NADPH levels are indeed the underlying cause for the observed reduction in lipogenesis. Also, why is the synthesis of long chain fatty acids affected by the loss of G6PD?

Finally, the significance and rationale of the experiments with the KPL mouse model are unclear. To link this part better to the rest of the paper and to strengthen the main hypothesis, the authors should also assess the NADPH/NADP⁺ and GSH/GSSG ratio in the KPL mice to check for oxidative stress.

2. In general, it is unclear why the authors chose to focus on the redox stress in the first place. To make this hypothesis stronger and justify the focus the authors should show all their metabolomics data as a whole rather than only picking specifically on NADPH and GSH levels (figure 2 j,k and figure 4a, b). In this way it would be possible to appreciate the general effects of G6PD deficiency

on metabolism (Is the PPP affected? Is nucleotide biosynthesis affected?) and how it differentially affects metabolism in the different mouse models. Importantly, a side-by-side comparison of the two mouse models (KP and KL) is warranted to understand better why G6PD depletion is particularly important for redox homeostasis in the KL model. It is indeed possible that the KP and KL model have a different baseline of redox stress.

3. There is a significant inconsistency in the timing of sacrifice of the animals without a clear justification. This is an important concern because the redox phenotype may have different dynamics and kinetics in their various models, leading to possible misinterpretations when choosing a single time point of harvest. For example, the IHC analysis in figure 5b of p53, p21 and CC3 need to be performed at 6 weeks post tumour induction and not only at 12 weeks. Similarly, in the KPL mouse model, all the analyses were performed at 6 weeks after tumour induction but it is unclear whether at a later time point difference in the tumor growth would arise like seen in the KL model (no difference at 7 weeks but at 12 weeks).

Minor critiques:

1. For the overall survival analysis in Figure 1, it is indicated that 33 studies obtained from the cBioPortal datasets were used to draw a correlation between expression levels of cytosolic NADPH-generating enzymes and survival of lung cancer patients. Yet, while the study focuses on KRAS-driven NSCLC, the list includes studies from SCLC, thoracic cancer and thoracic PDX. These studies need to be excluded from the analysis as they might skew the contribution of the other enzymes in the total survival of the patients. Additionally, both G6PD and MTHFD1 considerably impact overall survival in patients with KRAS/LKB1 co-MUT. The authors have to clarify the purpose of choosing G6PD for their study.

2. Loss of G6PD has to be additionally confirmed by orthogonal assay (western blot or qPCR) performed on the tumours since the IHC analysis that can be unspecific.

3. There is no clear explanation on focusing on pERK and pS6 in figure 3i. The IHC staining looks quite unspecific. This part seems disconnected from the rest of the paper.

4. Statistics should be included in all graphs and the number of replicates should be stated for all shown data. All figure legends should include a more detailed description of the experimental procedure, the n of all experiments, and the dependence of replicates.

5. Figure 3 and 4a, b could be combined to have a more consistent presentation of the data. It is advisable to show figure 3 before figure 2 or move figure 2 to the supplementary section.

6. The lenti-Cre nasal inhalation has been performed based on a previously applied methodology as clearly stated in the method section. However, the mice were infected with 5×10^6 plaque-forming units (pfu) per mouse in contrast to 2×10^6 that has been used before. The endpoint of the experiments was also quite different from the previous one. These differences need to be clearly stated.

7. The H&E-stained lung images of all experiments are very small and it is difficult to evaluate and judge the cell count and tumor size. For this reason, increasing their size for better visibility would be beneficial.

8. The Incucyte experiment should be better presented in figure 4i. What is depicted in the y axis? The graph for WT and KO should be combined in one graph and replicates should be presented as dots, including the SD. For better evaluation of the mechanism a cell death assay could be additionally performed together with the proliferation assay.

9. The authors should clearly explain the name of the TDCLs in the method section or the figure legend (Fig 4 f-n). What is 1-9, 2-2 or 4-5? Or even simplify the names (clone 1, 2, etc). The same needs to be done for the mouse numbers in figure 6d.

10. In Fig. 5b a better representative picture should be chosen for the p53 staining. There is no

clear positive staining that corresponds to the respective quantification.

11. The authors must explain why serum/tumours from different mice have been used for the lipidomics experiment in figure 6d. Why only show KP fed, whereas differences are pinpointed in KL fasted state? The authors should also include fatty acid species up to C34.4 for the fasted state in figure 6d, as shown in the supplementary part for the fed state.

12. In Figure 7b,c the individual fractions should be presented for the metabolomic analysis, including all isotopologues in one graph for clarity.

13. To corroborate the findings and the drawn conclusion, NADPH/NADP⁺ and GSH/GSSG ratio should be assessed in KL TDCLs upon serine/glycine depletion.

14. All omics data should be provided as spreadsheets as a supplement.

Reviewer #2 (Remarks to the Author):

The manuscript represents important work by identifying metabolic vulnerabilities of lung cancer in a specific genetic context. The data are mostly derived from transgenic mouse models, whose use is backed by an analysis of human lung cancer patients. Together, with mechanistical insights into metabolic rewiring of tumours using sophisticated in vivo and in vitro methods, these results could lead to genetically-informed personalized treatment options targeting the pentose phosphate pathway.

This leaves only a few questions:

Results in the KL model: Knock out of LKB1 should lead to reduced AMPK signalling and with that to increased fatty acid biosynthesis (often mediated by regulation of ACC1 phosphorylation through AMPK). How can this be reconciled with the observation of reduced lipogenesis and fatty acid synthesis in the KL model? Is AMPK signalling (ACC phosphorylation) impacted in KL tumours (comparison KRAS only, KL, KL;G6pdKO)?

Can the authors describe how they calculate tumour burden? Especially, because there is a disparity between burden in the KL and the KPL mice: In Fig 3g it is ~8% after 7 weeks and in Fig 5i (KPL) ~0.18% after 6 weeks. It is unlikely that additional knock-out of p53 improves tumour burden to this extent.

Along these lines, it should be discussed if differences of KL and KP tumour aggressiveness (tumour burden in KL vs KP is 20% vs 5% after 12 weeks) is causative for the differences in the two models. I.e., is higher proliferation leading to higher NADPH demand and increased dependency on oxPPP and G6PD?

It should be discussed, how LKB1, TP53, and G6PD mutations in patients can be compared to complete knock out in the used models.

Lipdomics data:

- Fig 6c, d (and supplementary heatmaps): What is S1, S2, T1, T2? Why are there different group sizes?
- Heatmaps in general: can the authors provide statistical measures to support their claim that there is no change under fed only in fasted state?
- With changes only in fasted KL tumours, the proper comparison would be lipidomics in fasted KP tumours (as neither KL nor KP tumours show changes in lipidomic profiles according to the authors' interpretation).
- If changes are only found in the fasted state, why are tracing experiments done in fed state (that actually show changes in Fig 6f and h)?

Mouse group sizes should be given in the figure legends!

To increase evidence for p53 activation in the KL model (as claimed in the abstract) the authors should show regulation (from RNA-seq data or with qPCR) of canonical p53 target genes in KL;G6pdKO vs WT tumours. For instance, apoptosis regulators (Puma, Bax, Bak) or the ones contributing to the GSEA enrichment in Fig 5a.

As Nrf2 is a central transcription factor regulating oxidative stress response and PPP genes, it would be interesting to include it in the cBioportal analysis.

If the authors could provide evidence that G6pd inhibition results in reduced tumour growth in G6pdWT;KL model or in the tumor-derived cell lines, this would additionally support the translational value of the manuscript.

The results should be discussed in the light of an earlier paper showing p53 inhibition of PPP through direct interaction with G6pd (Jiang et al, 2011, PMID: 21336310).

Fig 3b: KO tumours seem to be G6pd-positive on the tumour margins. How can this be explained?

Fig 3i: Why is downstream RAS and mTORC1 signalling reduced?

Fig 6m and 6o: Is the difference between HFD WT vs HFD KO significant?

Fig 7: Can the authors discuss how G6pd can influence serine metabolism despite higher 3PG levels?

Reviewer #3 (Remarks to the Author):

In the manuscript by Taijin Lan et al. titled "G6PD Maintains Redox Homeostasis and Biosynthesis in LKB1-Deficient KRAS-Driven Lung Cancer", the authors studied the reliance of different genetic subtypes of non-small cell lung cancer on G6PD activity. Previous studies from this group revealed that KRAS mutant/p53 loss (KP) tumors were unaffected by G6PD disruption (Ghergurovich JM, 2020). Now, the authors show that, in contrast to KP tumors, KRAS mutation/LKB1 loss (KL) tumors are heavily dependent on G6PD. The authors identify several alterations in KL tumors elicited by G6PD disruption, which include a decrease in NADPH levels, induction of oxidative stress, and activation of p53. These results provided important mechanistic insight into the differences between KP and KL. They also showed that G6PD deletion promotes an alteration in lipid abundance, and lipid supplementation through a high-fat diet can rescue growth defects in KL tumors. Further, they showed that G6PD deletion specifically rewires serine metabolism, increasing the flux of extracellular serine to the one-carbon metabolism, generating NADPH from one-carbon units donated by serine. This study is interesting; however, several technical and conceptual concerns should be addressed.

Major concerns

1. In Figure 1, the authors state, "These results suggest that G6PD and MTHFD1 expression impact survival in a subset of lung cancer patients (KRAS/LKB1 co-MUT and not KRAS/TP53 co-MUT lung cancers)." However, the data provided does not support this statement. The data suggests that G6PD and MTHFD1 expression are correlated with survival. These findings are potentially better suited as a Supplementary Figure instead of a main Figure.

2. A major question is why G6PD loss does not alter NADPH levels in KP lung tumors. This is significant because the authors suggest that in KL tumors, G6PD loss causes a drop in NADPH levels, oxidative stress, p53 induction, and slow tumor growth. Based on this rationale, G6PD loss should decrease NADPH levels independent of p53 status. This point should be addressed. Further, it would be informative to know whether loss of G6PD in KPL tumors impacts NADPH and NADP+ levels.

3. Several technical details need to be provided in the Figures and Figure Legends. For example, the authors show a GSEA plot referring to an "Oxidative stress" signature, but no other data is provided referring to how this signature was generated. Is this a "Hallmark" gene set or a "GO Biological Pathway"? Further, measuring the levels of specific NRF2 target genes, such as Nqo1 and

Hmox1, would be informative as a readout of oxidative stress. Later, in Figure 4i, the authors show a graph with an unlabeled y-axis. They mention that this is a "proliferation rate," but additional details should be included in the graph and Figure legend. Also, the author should include in the Figure legends the number of animals used in each experiment.

4. Some technical approaches could be clearer. They mention for "mRNA-seq" that "The lung tumors were rapidly dissected and snap-frozen in liquid nitrogen." In Figures 2 and 3, the authors report upwards to hundreds of tumors per mouse in the lung tissue. It is unclear how these tumors were individually isolated, not to include non-tumor lung tissue.

5. The authors state, "Thus, the slow growth of G6PD-knockout KL tumors is due to oxidative stress-inducing p53 activation and p53 activation inhibiting tumor progression." But this isn't shown. The authors demonstrate that the slow growth of G6PD-knockout KL tumors is due to p53, but they have not rescued the oxidative-stress phenotypes they observe in G6PD-knockout KL tumors; thus, do not know if it is oxidative stress that is inducing p53 activation. The authors could test this using antioxidant supplementation. Alternatively, the authors could change the writing to state the slow growth of G6PD-knockout KL tumors is due to p53 activation inhibiting tumor progression".

6. Conceptually, the manuscript is hard to follow. The Introduction section suggests that the authors will address how different genetic drivers can impact NSCLC reliance on G6PD, and the authors investigate this in Figures 2-5. However, in Figures 6 and 7, they introduce a new line of investigation into how NADPH modulates lipid synthesis and induces a reliance on serine abundance. These findings appear disjointed from the remaining Figures and reduce the clarity of the overall study.

7. Some aspects of the study are not fully discussed. They mention, "Compared with G6pdWT; KL lung tumors, G6pdKO; KL tumors had significantly lower levels of long-chain fatty acyl groups, whereas very long-chain fatty acyl groups accumulated in G6pdKO; KL lung tumors (Fig. 6d)." These findings are not discussed. How do authors interpret the accumulation of very long-chain FA over long-chain FA? Is this observed in any disease? The authors should speculate on the functional consequences of these phenotypes. Further, did the high-fat diet rescue this lipidomic imbalance?

8. It is unclear why only four mice are shown in Figures 5E, F, H, and I if Fig 5K presents data from more than 25 mice of the same genotype.

9. As Vitamin C is paradoxically an antioxidant or a pro-oxidant depending on the dose, demonstrating that Vitamin C is causing oxidative stress/damage in your model (e.g., 8-oxo-dG, 4-HNE, or even mRNA expression of NRF2 targets) would strengthen the data.

Minor concerns

1. It is unclear why the authors only use IHC and not qPCR to confirm G6PD deletion in tumors.

2. At Line 126, a reference is needed for the statement: "Tumors exhibit an enormous demand for NADPH due to uncontrolled proliferation."

3. Lines 236-237 appear to have truncated text. Please verify.

4. It would be informative to present levels of oxPPP metabolites in Fig 7B-C-D.

5. It would be informative to present NADPH levels of TDCL in Ser/Gly-free media.

6. At the end of Figure 2's legend, there is a mention of D2O infusion, which seems to be a mistake.

Reviewer #4 (Remarks to the Author):

This study by Lan et. al investigates impact of G6pd loss in NSCLC harboring co-mutations in KRAS and LKB1 (KL). The authors demonstrate that G6PD is important for KL tumorigenesis as well as cellular NADPH production. Further, the authors identify serine-glycine one-carbon (SGOC) metabolism as a key NADPH generating source under G6PD suppression in KL tumors. Loss of G6PD in KL tumors drives upregulation of SGOC metabolism, which drives increased NADPH production for antioxidant defenses. The finding that G6PD is selectively required for KL tumorigenesis and G6PD loss reprograms SGOC metabolism is interesting.

1. In Fig.2, the authors claim that G6PD is not required for NADPH and redox control in KP tumors. Given the critical role of G6PD in cytosolic NADPH production, there might be some compensatory mechanisms to maintain NADPH and GSH pools in KP tumors. It would be important to understand contribution of IDH1 and ME1 in cellular NADPH production in KP-G6pd WT and KO tumors to confirm 1) G6PD is not the major NADPH generating machinery in KP tumors and 2) test whether G6PD loss reprograms contribution of other cytosolic NADPH sources. 1-2H-glucose (G6PD), 2,3,3,4,4-2H-glutamine (IDH1), and 2,3,3-2H-aspartate (ME1) could be used to label cells.

2. In relation to the previous point, the authors should check NADPH production from IDH, ME1 as well as SGOC in KL-G6pd WT and KO TDCL. IDH1 and ME1 contribution for NADPH production would be hard to determine from U-13C glucose.

3. In Fig.1, although ME1 expression is not 'statistically' significantly associated with prognosis, it does seem to have biological meaning; graph looks almost identical to that of G6PD. The authors might want to mention about it.

4. Based on KM graph and mRNA expression data in Fig1c,d,e, mRNA expression is not always a good readout for prognosis. It would be important to discuss in the manuscript (e.g., what would be the authors' thought?).

Re: NCOMMS-23-47286-T

We appreciate the peer reviewers for their positive assessment of our manuscript and for their constructive suggestions. We have carefully considered each comment and made revisions accordingly to improve the overall quality and clarity of our work. We have addressed the comments point-by-point below.

Reviewer #1 (Remarks to the Author):

In the manuscript entitled “G6PD Maintains Redox Homeostasis and Biosynthesis in LKB1-Deficient KRAS-Driven Lung Cancer” Lan et al investigated the distinct dependency of different subtypes of KRAS-driven NSCLC on the pentose phosphate pathway (PPP) enzyme G6PD. To this aim, they utilized genetically engineered mouse models lacking G6PD in the context of *Kras*G12D/+; *p53*^{-/-} (KP), *Kras*G12D/+; *Lkb1*^{-/-} (KL) or *Kras*G12D/+; *p53*^{-/-};*Lkb1*^{-/-} (KPL), as well as tumour-derived cell lines (TDCLs). They demonstrated that G6PD is indispensable for KL but not for KP lung tumorigenesis as indicated by an extension of the mouse lifespan and a substantial reduction in the KL lung tumour number, burden, and proliferation. The authors suggest that G6PD loss in KL lung cancer is associated with a disruption of cellular redox homeostasis via suppression of NADPH generation resulting in oxidative stress, *p53*-dependent apoptosis, and growth arrest. Additionally, the decrease in NADPH affects *de novo* lipogenesis in KL lung tumours, which in turn inhibits tumour growth. Finally, the authors propose an increase in serine uptake as an alternative cytosolic NADPH-producing metabolic pathway in order to maintain redox homeostasis in G6PD-deficient KL tumours.

Although potentially interesting, this study appears to lack a clear rationale and scope, and logic in the selection and presentation of the experiments. At times, it appears to be an exercise of mouse genetics without an underlying question. Moreover, the work harbours major technical and conceptual limitations that question the conclusions and translational potential of the work. Finally, the role of G6PD in lung cancer driven by oncogenic KRAS has been previously shown (by the same authors), diminishing the overall novelty and impact of the work.

Thank you for your valuable feedback. We've made substantial revisions, offering a clearer and more succinct rationale for our study to underscore its significance. In our prior publication (PMID: 32661137), we found that G6PD is dispensable for KP lung tumorigenesis, which is further confirmed using different mouse model in this study. Given the distinct nature of KL and KP as two subtypes of KRAS-driven lung cancer, known to exhibit varying responses to standard cancer treatments, our current study is novel in revealing the essential role of G6PD in KL lung tumorigenesis. This underscores the need for personalized therapies tailored to different subgroups of KRAS-driven lung cancer. Our findings carry translational potential, particularly in the context of using G6PD inhibitors for cancer treatment. To delve deeper into this aspect, we have expanded the discussion section to thoroughly explore how our results contribute to the development of personalized therapeutic approaches in the treatment of KRAS-driven lung cancer. We believe that these enhancements strengthen the manuscript and more effectively convey the implications of our research.

Major critiques:

1. The mechanistic part of the paper addresses the causes of the vulnerability of the KL tumours upon G6PD deficiency and branches in three distinct directions: redox stress, decreased lipogenesis, and increased serine uptake. However, these parts remain mechanistically disconnected, limiting the overall relevance of these findings and depth of investigation. For example, as supported by the data in figure 4i, G6PD deficiency significantly reduces allograft tumor growth but the clinical relevance of the shown

vitamin C treatment is questionable. Do the authors suggest a combination of vitamin C as an adjuvant agent together with G6PD inhibitors for the patients harboring KRAS and LKB1 co-MUT? If so, this has to be clearly stated in the discussion.

We appreciate your input. G6PD-dependent cytosolic NADPH generation plays a crucial role in managing redox stress and lipogenesis. The loss of G6PD in KL tumors leads to a reprogramming of serine metabolism. The modifications in these three branches: redox stress, decreased lipogenesis, and increased serine uptake, potentially contribute to the initial reduction in the growth of G6PD-deficient KL lung tumors and their subsequent growth in later stages. This has been further elaborated in the results and integrated into our discussion. Additionally, we have included a summary model (**New Fig. 6q, Page 13, Line 24-26; Page 14, Line 1-7**) to elucidate this.

Vitamin C (Vit C) as an adjuvant has been explored in both preclinical and clinical studies. Our findings indicate that a combination with G6PD inhibitors and high-dose Vit C could be a therapeutic approach for treating KL lung cancer. As suggested, we have stated this in the discussion (**Page 19, Line 1-8**):

“Cancer cells exhibit greater sensitivity to the cytotoxic effects of oxidative stress when compared to normal cells. The pro-oxidant properties of high-dose Vit C, achieved through the generation of ROS including H₂O₂, make it a promised adjuvant in cancer treatment and has been explored in many pre-clinical and clinical studies⁵⁵⁻⁵⁷. Our observations reveal that *G6PD^{KO};KL* tumors are responsive to high-dose Vit C, resulting in tumor reduction. This suggests that when treating KL lung tumors with a G6PD inhibitor, incorporating high-dose Vit C as an adjuvant may be beneficial. Furthermore, exploring the potential therapeutic strategy of combining a G6PD inhibitor with agents that induce oxidative stress holds promise for treating this specific subtype of KRAS-mutant NSCLC.”

The same argument applies to the essentiality of serine uptake in this model (figure 7). It is unclear why the authors focused on this pathway and the significance these results have in finding novel patient therapies. The authors should clarify this point and discuss it better.

Thank you for your comments. Tumor cells continually undergo metabolic reprogramming *in vivo* to preserve cellular homeostasis and progression. Cytosolic NADPH production could occur through different pathways, including serine-mediated one-carbon metabolism. Despite the inhibitory effect of G6PD loss on KL tumor progression, compensatory mechanisms exist that can support the growth of KL lung tumors. Our *in vivo* tracing data suggest that reprogrammed serine metabolism may be one of these compensatory mechanisms. Thus, G6PD loss reshapes serine metabolism, influencing KL tumor progression and proposing a novel combination treatment for KL lung cancer by combining G6PD inhibitors with the blockade of serine mediated one-carbon metabolism. However, besides generating cytosolic NADPH, one-carbon metabolism plays a crucial role in maintaining nucleotide metabolism. Further mechanistic studies are required to validate the effectiveness of this combination in cancer treatment *in vivo*. More detailed discussion was added in the revised manuscript (**Page 17, Line 4-26; Page 18, Line 1-9**).

Along the same line, the lipogenesis part seems disconnected from the main message of the paper and superficial. To strengthen the hypothesis that HFD contributes to lung tumour growth in G6PD-deficient animals, the authors should perform a lipidomic analysis in serum and tumour from the mice fed with HFD side by side with the ND to check for differences in the abundance, saturation, and elongation of the fatty acids. The authors should more thoroughly assess the connection between changes in NADPH and lipogenesis, for instance, by overexpressing cytosolic TPNOX (Cracan et al 2017) to test whether reduced NADPH levels are indeed the underlying cause for the observed reduction in lipogenesis. Also, why is the synthesis of long chain fatty acids affected by the loss of G6PD?

Thanks for your suggestion. We performed lipidomics of lung tumors and serum from tumor-bearing mice at fasted state at 7-week post tumor induction. HFD significantly increased the levels of fatty acids in serum of KL tumor bearing mice, but had no impact on fatty acyl composition and levels of *G6pd*^{WT};KL lung tumors. Due to the minimal tumor burden of *G6pd*^{KO};KL lung tumors at 7 weeks post tumor induction in normal diet (ND), we were unable to collect *G6pd*^{KO};KL lung tumors for lipidomics. Despite this, we were able to collect *G6pd*^{KO};KL lung tumors in HFD. Therefore, we compared fatty acyl group composition between *G6pd*^{WT};KL and *G6pd*^{KO};KL lung tumors under HFD conditions. The level of C16:0 is comparable between *G6pd*^{KO};KL lung tumors and *G6pd*^{WT};KL lung tumors under HFD (**Supplemental Fig. 7e, f**). However, the levels of many very long-chain fatty acyl groups in *G6pd*^{KO};KL lung tumors were lower than those in *G6pd*^{WT};KL lung tumors under HFD conditions (**Supplemental Fig. 7e**). This suggests that HFD partially rescue the alterations in fatty acyl groups pool size levels caused by G6PD loss. New data were added in **Supplemental Fig. 7a-f, Page 11, Line 3-14**.

It is indeed a valuable suggestion to more thoroughly assess the connection between changes in NADPH and lipogenesis. We observed that G6PD-deficient KL tumors show lower NADPH than WT tumors (**Fig. 2a**). Moreover, G6PD loss leads to a significant reduction in fatty acid synthesis in *G6pd*^{KO};KL tumors, as evidenced by *in vivo* D₂O tracing (**Fig. 5e**) and in *G6pd*^{KO};KL TDCLs through *in vitro* [U-¹³C₆]-glucose labeling (**Supplemental Fig. 5a-b**). These findings are further supported by GSEA of tumor RNA-seq data and pACC IHC (**Fig. 5a-c**), indicating reduced fatty acid and lipid biosynthesis in G6PD-deficient KL lung tumors. Importantly, it is established that cytosolic NADPH exclusively serves as the hydrogen source for *de novo* lipogenesis. Taken together, our data demonstrate that G6PD-mediated cytosolic NADPH in KL lung tumors plays a pivotal role in supporting *de novo* lipogenesis. We have elucidated this connection more explicitly in the revised manuscript (**Page 9, Line 26, Page 10, Line 1-11**).

We appreciate your valuable suggestion to investigate G6PD-mediated NADPH in lipogenesis through the overexpression of cytosolic TPNOX. However, we think this is beyond the scope of the current study.

Finally, the significance and rationale of the experiments with the KPL mouse model are unclear. To link this part better to the rest of the paper and to strengthen the main hypothesis, the authors should also assess the NADPH/NADP⁺ and GSH/GSSG ratio in the KPL mice to check for oxidative stress.

Thank you for your suggestions.

DNA damage and oxidative stress activate p53, leading to cell cycle arrest and apoptosis. We observed increased oxidative stress in KL tumors by G6PD loss. We therefore hypothesize that p53 activation in KL tumors may contribute to slow tumor growth. Rather than checking the NADPH/NADP⁺ and GSH/GSSG ratios to assess oxidative stress in KPL lung tumors, a process requiring fresh tumor tissues to extract polar metabolites, a process of minimum of 6-8 months (inclusive of mice breeding, tumor induction, and metabolomics), we opted to perform IHC of NRF2 and NQO1, markers indicative of oxidative stress, using available KPL tumor paraffin sections. We observed increased oxidative stress in *G6pd*^{KO};KPL lung tumors compared to *G6pd*^{WT};KPL lung tumors (**New Fig. 4I, Page 9, Line 6-9**). This suggests that oxidative stress alone, without p53, is not sufficient to slow KL lung tumor growth. It underscores that the slower growth of *G6pd*^{KO};KL tumors may be attributed to p53 activation.

2. In general, it is unclear why the authors chose to focus on the redox stress in the first place. To make this hypothesis stronger and justify the focus the authors should show all their metabolomics data as a whole rather than only picking specifically on NADPH and GSH levels (figure 2 j,k and figure 4a, b). In this way it would be possible to appreciate the general effects of G6PD deficiency on metabolism (Is the PPP affected? Is nucleotide biosynthesis affected?) and how it differentially affects metabolism in the different mouse models. Importantly, a side-by-side comparison of the two mouse models (KP and KL) is warranted to understand better why G6PD depletion is particularly important for redox homeostasis in

the KL model. It is indeed possible that the KP and KL model have a different baseline of redox stress.

Maintaining cellular redox homeostasis, particularly through cytosolic NADPH, is crucial for proper tumor growth. This study aims to elucidate the functional importance of metabolic enzymes involved in cytosolic NADPH homeostasis *in vivo*. Therefore, the study begins by exploring the NADPH/NADP⁺ and GSH/GSSG ratios and the impact on redox homeostasis. We provided further clarification on this matter in the revised version.

As suggested, a side-by-side comparison of the two mouse models (KP and KL) of general metabolomics in KP and KL lung tumors with or without G6PD was provided (**New Supplemental Fig. 9 for KP** and **New supplemental Fig. 8 for KL**). Interestingly, we found that in contrast to KL lung tumors, the levels of TCA cycle metabolites are higher in *G6pd*^{KO};*KP* lung tumors compared to *G6pd*^{WT};*KP* lung tumors, and the levels of other core metabolites were similar between *G6pd*^{WT};*KP* and *G6pd*^{KO};*KP* lung tumors (**New Supplemental Fig. 9, Page 12, Line 6-10**). Previous research has shown that G6PD loss in *KEAP1*^{-/-};*KP* lung cancer cells leads to TCA intermediate depletion (PMID: 34788087). Here we found that G6PD loss does not affect TCA cycle metabolism in KL lung tumors (**Fig. 6c, New Supplemental Fig. 8b, c**). This emphasizes that oncogenic events play a crucial role in determining the dependence and associated mechanisms of distinct subtypes of KRAS-driven lung cancer on G6PD. This was discussed in revise manuscript (**Page 17, Line 8-22**):

“Our *in vivo* isotope tracing and flux analysis revealed that G6PD deficiency in KL lung tumors does not affect glucose carbon flux to tumor pyruvate, lactate, and TCA cycle intermediates. However, G6PD loss in KL lung tumors reduces glucose carbon flux to serine. Additionally, serine uptake is increased to maintain the serine pool size level in G6PD-deficient KL lung tumors for cytosolic NADPH production. We found that in *in vitro* cell culture, increased serine uptake is used to maintain redox homeostasis for cell proliferation. Therefore, serine-mediated one-carbon metabolism compensates for G6PD loss in KL cancer cell survival, although this does not preclude the potential compensatory cytosolic NADPH production through ME1 or IDH1. Indeed, in *KEAP1* mutant *KP* lung tumor cells, G6PD loss triggered TCA intermediate depletion because of up-regulation of the alternative NADPH-producing enzymes ME1/2 and IDH1/2⁷. However, in *KP* lung tumors, the depletion of G6PD resulted in an increase in the levels of TCA cycle intermediates. This emphasizes that oncogenic events play a crucial role in determining the dependence and associated mechanisms of distinct subtypes of KRAS-driven lung cancer on G6PD. This also indicates that the compensation for G6PD loss may involve mechanisms beyond the upregulation of ME1/2 and IDH1/2 alone in *KP* lung tumors.”

It's really a good suggestion that **“It is indeed possible that the KP and KL model have a different baseline of redox stress”**. We have further explored the potential reasons for the divergent dependence on G6PD between KP and KL lung tumors in discuss section (**Page 14, Line 9-24**):

“The differing dependency on G6PD in KL and KP lung tumorigenesis can be attributed to the following factors. LKB1 serves as a central modifier of cellular response to different metabolic stress. Loss of LKB1-AMPK signaling results in heightened sensitivity to energy depletion and to disturbances in redox homeostasis³⁸. It is possible that KL lung tumors, which lack proper AMPK activity, exhibit a greater metabolic vulnerability and less plasticity in response to G6PD loss when compared to KP lung tumors that retain intact LKB1 function. In contrast, KP lung tumors can swiftly adapt to G6PD loss due to their functional LKB1/-AMPK signaling, ensuring tumor survival. Specifically, the absence of G6PD has minimal impact on the NADPH/NADP⁺ and GSH/GSSG ratios in KP lung tumors, whereas these ratios are significantly altered in G6PD-deficient KL lung tumors. This suggests that KL lung tumors may inherently possess higher basal redox stress than KP lung tumors, rendering them more sensitive to disturbances in redox homeostasis. In addition, clinical studies have suggested that lung cancer patients with KL mutations are resistant to most cancer therapies, indicating increased aggressiveness compared to patients with KP mutations²². This increased aggressiveness of KL lung tumors has also been observed in preclinical mouse models³⁹. Therefore, the enhanced aggressiveness in KL lung tumors

could be attributed to increased proliferation, leading to a higher demand for NADPH and a greater dependence on G6PD-mediated oxPPP compared to KP lung tumors.”

3. There is a significant inconsistency in the timing of sacrifice of the animals without a clear justification. This is an important concern because the redox phenotype may have different dynamics and kinetics in their various models, leading to possible misinterpretations when choosing a single time point of harvest. For example, the IHC analysis in figure 5b of p53, p21 and CC3 need to be performed at 6 weeks post tumour induction and not only at 12 weeks. Similarly, in the KPL mouse model, all the analyses were performed at 6 weeks after tumor induction but it is unclear whether at a later time point difference in the tumor growth would arise like seen in the KL model (no difference at 7 weeks but at 12 weeks).

This is a very good point that redox phenotype may have different dynamics and kinetics in the various models. While we perform time course study of KL tumor model, we noticed that at 7 weeks post-tumor induction, mice bearing *G6pd*^{KO};*KL* lung tumors exhibit an extremely low tumor burden, rendering immunohistochemistry (IHC) analysis at 6 weeks post-tumor induction less meaningful. Therefore, we only provided IHC of KL tumors at 12 weeks time point. In contrast, KPL tumors are considerably more aggressive than KL tumors, with mice beginning to die at 8.5 weeks post-tumor induction. Therefore, KPL mice were sacrificed at 6 weeks post-tumor induction for histology analysis. Additionally, the tumor burden of *G6pd*^{WT};*KL* at 12 weeks (**Fig. 1n**) is comparable to that of *G6pd*^{WT};*KPL* at 6 weeks post-tumor induction (**Fig. 4i**). In addition to tumor burden analysis, survival curves also provide strong evidence that G6PD deletion has no effect on KPL tumor growth (**Fig. 4k**). Given these considerations, we don't think it's necessary to include a later time point for the KPL model.

Minor critiques:

1. For the overall survival analysis in Figure 1, it is indicated that 33 studies obtained from the cBioPortal datasets were used to draw a correlation between expression levels of cytosolic NADPH-generating enzymes and survival of lung cancer patients. Yet, while the study focuses on KRAS-driven NSCLC, the list includes studies from SCLC, thoracic cancer and thoracic PDX. These studies need to be excluded from the analysis as they might skew the contribution of the other enzymes in the total survival of the patients. Additionally, both G6PD and MTHFD1 considerably impact overall survival in patients with KRAS/LKB1 co-MUT. The authors have to clarify the purpose of choosing G6PD for their study.

In this revision, we have excluded SCLC, thoracic cancer, and thoracic PDX cBioPortal datasets from our analysis, reaffirming the same conclusion as in our previous findings (**New supplemental Fig. 1**).

Higher MTHFD1 expression is also associated with poor survival of lung cancer patients with wild type KRAS, except for those with KRAS/LKB1 co-mutations (**New Supplemental Fig. 1a**). In contrast, the impact of G6PD on overall survival is specific to patients with KRAS/LKB1 co-mutations. Therefore, G6PD was selected as the focus of this study, and this has been explicitly outlined in the revised text (**Page 5, Line 15-17**): “Regarding MTHFD1, besides its connection with survival outcomes in lung cancer patients with WT KRAS (Fig. 1a), its high expression is also associated to poorer survival in patients with KRAS/LKB1 co-mutations (Supplemental Fig. 1c).”

2. Loss of G6PD has to be additionally confirmed by orthogonal assay (western blot or qPCR) performed on the tumours since the IHC analysis that can be unspecific.

The G6PD antibody used for IHC has been validated in our previous publication (PMID: 32661137). Due to complicated tumor microenvironment, obtaining pure tumor samples for Western blot or qPCR is unfeasible. Therefore, IHC is expected to yield more robust results *in vivo* than Western blot or qPCR. In addition, we have provided G6PD mRNA expression from KL lung tumor mRNA-seq data to show the

reduced *G6pd* mRNA expression in *G6pd*^{KO};*KL* lung tumors compared to *G6pd*^{WT};*KL* lung tumors (**New supplemental Fig. 2b**). The remaining *G6pd* mRNA expression could be from other cells in tumor microenvironment, including stromal cells, infiltrated immune cells, and adjacent normal lung tissues. This can be reflected by *Lkb1* mRNA expression in both *G6pd*^{WT};*KL* and *G6pd*^{KO};*KL* lung tumors. Furthermore, the Cre-Lox system represents a well-established model for investigating gene knockout in KRAS-driven non-small cell lung cancer (NSCLC). For mouse lung tumor derived cell lines, we have provided Western blot to confirm G6PD deletion (**New Fig. 3b**).

3. There is no clear explanation on focusing on pERK and pS6 in figure 3i. The IHC staining looks quite unspecific. This part seems disconnected from the rest of the paper.

pERK and pS6 serve as additional markers for tumor growth, as explained further in this revision (**Page 6, Line 19-22**). Both pErk and pS6 antibodies have been validated by our previous publications (PMID: 23824538, 24875857). The IHC staining we showed are specific for pERK and pS6. We also included a lower magnification of pErk IHC and pS6 IHC in **New Supplementary Fig. 2c** to confirm the specificity of the antibodies.

4. Statistics should be included in all graphs and the number of replicates should be stated for all shown data. All figure legends should include a more detailed description of the experimental procedure, the n of all experiments, and the dependence of replicates.

Figure legends have been revised as suggested.

5. Figure 3 and 4a, b could be combined to have a more consistent presentation of the data. It is advisable to show figure 3 before figure 2 or move figure 2 to the supplementary section.

As advised by Reviewer 3, original Fig. 1 has been relocated to the **New Supplemental Fig. 1**. To directly compare the different responses of KP and KL lung tumor on G6PD ablation, we combined original Fig. 2 a-i and Fig. 3 as a **New Fig. 1**, moved original Fig. 2j, k to create as a **New Supplemental Fig. 3**, moved original Fig. 4a-d to create as a **New Fig. 2**, and created original Fig. 4e-n as a **New Fig. 3**. This reorganization makes the manuscript more cohesive.

6. The lenti-Cre nasal inhalation has been performed based on a previously applied methodology as clearly stated in the method section. However, the mice were infected with 5x10⁶ plaque-forming units (pfu) per mouse in contrast to 2x10⁶ that has been used before. The endpoint of the experiments was also quite different from the previous one. These differences need to be clearly stated.

In our experience, the titers of various batches of Lenti-virus may exhibit slight variations. Typically, titers ranging between 2x10⁶ and 10x10⁶ have proven effective for inducing lung tumors. In this study, we chose 5x10⁶.

7. The H&E-stained lung images of all experiments are very small and it is difficult to evaluate and judge the cell count and tumor size. For this reason, increasing their size for better visibility would be beneficial.

Representative larger-sized H&E images have been provided, as recommended, in **New Fig. 1e, I, New Fig. 4g, and New Fig. 5m**.

8. The Incucyte experiment should be better presented in figure 4i. What is depicted in the y axis? The graph for WT and KO should be combined in one graph and replicates should be presented as dots, including the SD. For better evaluation of the mechanism a cell death assay could be additionally performed together with the proliferation assay.

The y-axis label has been added to the Incucyte data. Combined Incucyte data were included to show statistics (**New Fig. 3f**).

In addition, we performed MTS assay of *G6pd*^{WT};KL and *G6pd*^{KO};KL TDCLs in nutrient rich conditions. *G6pd*^{KO};KL TDCLs show less proliferation compared to *G6pd*^{WT};KL cells (**New Supplemental Fig. 4a**), consistent with Incucyte data. As suggested, cell death assay (Apoptosis/Necrosis assay) was performed and no significant cell death was observed in both *G6pd*^{WT};KL and *G6pd*^{KO};KL TDCLs in nutrient rich conditions (**New Supplemental Fig. 4b, c, Page 7, Line 25-26**), indicating decreased proliferation of *G6pd*^{KO};KL TDCLs was not linked to apoptosis and necrosis.

9. The authors should clearly explain the name of the TDCLs in the method section or the figure legend (Fig 4 f-n). What is 1-9, 2-2 or 4-5? Or even simplify the names (clone 1, 2, etc). The same needs to be done for the mouse numbers in figure 6d.

Thank you for your valuable feedback. The clone number for TDCLs has been marked in the new **Fig. 3b** when these cells were first characterized by western blot. Mouse number was simplified as suggested.

10. In Fig. 5b a better representative picture should be chosen for the p53 staining. There is no clear positive staining that corresponds to the respective quantification.

A better representative p53 staining was provided (**New Fig. 4b**).

11. The authors must explain why serum/tumours from different mice have been used for the lipidomics experiment in figure 6d. Why only show KP fed, whereas differences are pinpointed in KL fasted state? The authors should also include fatty acid species up to C34.4 for the fasted state in figure 6d, as shown in the supplementary part for the fed state.

The exclusion of one serum from the dataset was due to the contamination of the serum sample with a significant amount of red cell lysis during blood collection, rendering it unsuitable for lipidomics analysis. As a result, we have included only three mouse serum samples. Since tumor was not affected, we kept it in the tumor dataset. Those differences do not affect our conclusion.

Due to the inherent variability in LC-MS, the detection of metabolites may exhibit slight differences across runs. In the fasted state, fatty acid species up to C34.2 were included in the analysis (**New Fig. 5i**). However, due to the weak peak signaling for C34.3 and C34.4, these fatty acid species were not included in the analysis. This doesn't affect our conclusion.

Since G6PD has no impact on KP lung tumor growth and NADPH production, we therefore did not further explore fatty acid composition of KP tumors in fasted state in our first submission. In this revised manuscript, lipidomics of KP lung tumor and serum from tumor bearing mice at fasted state was provided as suggested. In contrast to KL lung tumors, G6PD loss had no significant effect on the compositions of fatty acids in KP lung tumors in fasted state (**New Supplemental Fig. 6a-d, Page 10, Line 20-21**).

12. In Figure 7b,c the individual fractions should be presented for the metabolomic analysis, including all isotopologues in one graph for clarity.

The individual fractions were provided for each metabolite as suggested (**New Supplemental Fig. 8a, b**).

13. To corroborate the findings and the drawn conclusion, NADPH/NADP⁺ and GSH/GSSG ratio should

be assessed in KL TDCLs upon serine/glycine depletion.

This information is now included in the revised manuscript (**New Fig. 6m, n**). Following 24 hours of serine/glycine depletion, there was a significant reduction in intracellular serine and glycine levels in both *G6pd^{WT};KL* and in *G6pd^{KO};KL* TDCLs (**New Supplemental Fig. 10a**). Moreover, the relative reduction of intracellular serine and glycine in *G6pd^{KO};KL* TDCLs was significantly greater than in *G6pd^{WT};KL* TDCLs (**New Supplemental Fig. 10b, c**). This suggests that *G6pd^{KO};KL* TDCLs rely more on serine/glycine uptake to maintain serine/glycine levels compared to *G6pd^{WT};KL* TDCLs. Additionally, serine/glycine depletion significantly decreased NADPH pool size level and NADPH/NADP⁺ ratio in *G6pd^{KO};KL* TDCLs compared to *G6pd^{WT};KL* TDCLs (**New Fig. 6m, New Supplemental Fig. 10d, Page 13, Line 11-13**). *G6pd^{WT};KL* TDCLs exhibited a trend of increased GSH/GSSG ratio, indicating an adaptive response to acute serine/glycine depletion to maintain redox balance. However, the GSH/GSSG ratio in *G6pd^{KO};KL* TDCLs remained unchanged, suggesting reduced adaptability in the absence of G6PD (**New Fig. 6n, Page 13, Line 13-16**). As a result, ROS level was significantly higher in *G6pd^{KO};KL* TDCLs compared to *G6pd^{WT};KL* TDCLs under serine/glycine depletion.

14. All omics data should be provided as spreadsheets as a supplement.

The RNA-seq data were provided as **Supplementary Data 1**, and data for metabolomics and lipidomics were provided in the **Source data** file.

Reviewer #2 (Remarks to the Author):

The manuscript represents important work by identifying metabolic vulnerabilities of lung cancer in a specific genetic context. The data are mostly derived from transgenic mouse models, whose use is backed by an analysis of human lung cancer patients. Together, with mechanistical insights into metabolic rewiring of tumours using sophisticated in vivo and in vitro methods, these results could lead to genetically-informed personalized treatment options targeting the pentose phosphate pathway.

We appreciate your positive comments on our manuscript.

This leaves only a few questions:

Results in the KL model: Knock out of LKB1 should lead to reduced AMPK signalling and with that to increased fatty acid biosynthesis (often mediated by regulation of ACC1 phosphorylation through AMPK). How can this be reconciled with the observation of reduced lipogenesis and fatty acid synthesis in the KL model? Is AMPK signaling (ACC phosphorylation) impacted in KL tumours (comparison KRAS only, KL, KL;G6pdKO)?

LKB1 phosphorylates and activates AMPK, while AMPK can also be activated by CaMKK β . As recommended, we conducted IHC analyses of pAMPK and its substrate pACC on *G6pd^{WT};Kras^{G12D/+}* (*G6pd^{WT};K*), *G6pd^{WT};KL*, and *G6pd^{KO};KL* lung tumors. As expected, pAMPK and pACC in *G6pd^{WT};KL* lung tumors was significantly lower than that in *G6pd^{WT};K* lung tumors due to LKB1 loss. This also indicates higher *de novo* fatty acid synthesis and more aggressiveness in *G6pd^{WT};KL* lung tumors compared to *G6pd^{WT};K* lung tumors. Additionally, *G6pd^{KO};KL* lung tumors exhibited higher pAMPK and pACC than *G6pd^{WT};KL* lung tumors, suggesting reduced *de novo* fatty acid synthesis. Increased pAMPK in *G6pd^{KO};KL* lung tumors could be activated by CaMKK β . New data were included in **New Fig. 5c, Page 9, Line 20-24**, and discussed in **Page 15, Line 13-21**: “LKB1 phosphorylates and activates AMPK, while AMPK can also be activated by CaMKK β ^{19,31}. AMPK inhibits the activity of ACC to suppress *de novo* fatty acid synthesis and promote fatty acid oxidation³⁸. An allosteric inhibitor of the ACC enzymes ACC1 and ACC2 markedly suppressed KL lung tumor growth⁴⁰. These suggest the potential anti-tumorigenic

role of AMPK. We observed that *G6pd*^{WT};*KL* lung tumors display lower AMPK activity compared to *G6pd*^{WT};*K* tumors, whereas *G6pd*^{KO};*KL* lung tumors exhibit heightened AMPK activity. The phosphorylation status of ACC supports this difference. Enhanced AMPK activity observed in *G6pd*^{KO};*KL* lung tumors may be attributed to CaMKK β , and requires further investigation. Indeed, *G6pd*^{KO};*KL* lung tumors show reduced *de novo* lipogenesis, probably due to reduced hydrogen source from NADPH.”

Can the authors describe how they calculate tumour burden? Especially, because there is a disparity between burden in the KL and the KPL mice: In Fig 3g it is ~8% after 7 weeks and in Fig 5i (KPL) ~0.18% after 6 weeks. It is unlikely that additional knock-out of p53 improves tumour burden to this extent.

The tumor burden analysis was detailed in the methods section (**Page 20, Line 18-25**). We identified an error in the y-axis labeling in original Fig 5i (**New Fig. 4i**); the number was actually a fraction that should be multiplied by 100 to represent a percentage. This has been rectified. Thank you for bringing it to our attention.

Along these lines, it should be discussed if differences of KL and KP tumour aggressiveness (tumour burden in KL vs KP is 20% vs 5% after 12 weeks) is causative for the differences in the two models. I.e., is higher proliferation leading to higher NADPH demand and increased dependency on oxPPP and G6PD?

Based on our utilization of GEMMs to study KRAS-driven NSCLC, we noted that KL lung tumors demonstrate increased aggressiveness in comparison to KP lung tumors. This increased aggressiveness in KL tumors could be attributed to enhanced proliferation, leading to elevated NADPH demand and a greater dependence on G6PD-mediated oxPPP compared to KP tumors. We appreciate your suggestion and have incorporated this insight into the discussion (**Page 14, Line 9-24**):

“The differing dependency on G6PD in KL and KP lung tumorigenesis can be attributed to the following factors. LKB1 serves as a central modifier of cellular response to different metabolic stress. Loss of LKB1-AMPK signaling results in heightened sensitivity to energy depletion and to disturbances in redox homeostasis³⁸. It is possible that KL lung tumors, which lack proper AMPK activity, exhibit a greater metabolic vulnerability and less plasticity in response to G6PD loss when compared to KP lung tumors that retain intact LKB1 function. In contrast, KP lung tumors can swiftly adapt to G6PD loss due to their functional LKB1-AMPK signaling, ensuring tumor survival. Specifically, the absence of G6PD has minimal impact on the NADPH/NADP⁺ and GSH/GSSG ratios in KP lung tumors, whereas these ratios are significantly altered in G6PD-deficient KL lung tumors. This suggests that KL lung tumors may inherently possess higher basal redox stress than KP lung tumors, rendering them more sensitive to disturbances in redox homeostasis. In addition, clinical studies have suggested that lung cancer patients with KL mutations are resistant to most cancer therapies, indicating increased aggressiveness compared to patients with KP mutations²². This increased aggressiveness of KL lung tumors has also been observed in preclinical mouse models³⁹. Therefore, the enhanced aggressiveness in KL lung tumors could be attributed to increased proliferation, leading to a higher demand for NADPH and a greater dependence on G6PD-mediated oxPPP compared to KP lung tumors.”

It should be discussed, how LKB1, TP53, and G6PD mutations in patients can be compared to complete knock out in the used models.

As suggested, the following was included in the discussion (**Page 19, Line 10-17**):

“While the discoveries from GEMMs are indeed exciting, it's essential to acknowledge a key distinction. In our GEMMs, LKB1, TP53, and G6PD are completely depleted in KRAS-driven lung tumors at the initiation of tumor formation. In contrast, cancer patients gradually accumulate mutations in LKB1 and TP53 over time, presumably upregulating G6PD expression. Furthermore, TP53 mutations in patients may result in gain-of-function alterations, a complexity not fully reflected in current GEMMs. Although our

GEMM findings highlight the role of G6PD in promoting KL, not KP, tumorigenesis, a deeper investigation is warranted to comprehend the implications of this discovery for the growth and treatment of KL or KP tumors in lung cancer patients.”

Lipidomics data:

- Fig 6c, d (and supplementary heatmaps): What is S1, S2, T1, T2? Why are there different group sizes?

Those are different mouse ID. As suggested by reviewer 1, we have simplified the mouse labeling. Please note, we excluded one serum from the dataset due to the contamination of the serum sample with a significant amount of red cell lysis during blood collection, rendering it unsuitable for lipidomics analysis. As a result, we have included only three mouse serum samples. Since tumor was not affected, we keep it in the tumor dataset (**New Fig. 5i**). Those differences do not affect our conclusion.

- Heatmaps in general: can the authors provide statistical measures to support their claim that there is no change under fed only in fasted state?

Statistics are presented in an Excel file provided as a supplementary table in **Source data** file. Significance levels are denoted by asterisks, with * indicating $P < 0.05$; ** indicating $P < 0.01$; *** indicating $P < 0.001$; **** indicating $P < 0.0001$, placed beside the relevant metabolites or fatty acids (FAs).

- With changes only in fasted KL tumours, the proper comparison would be lipidomics in fasted KP tumours (as neither KL nor KP tumours show changes in lipidomic profiles according to the authors' interpretation).

Since G6PD has no impact on KP lung tumor growth and NADPH production, we therefore did not further explore fatty acid level/composition of KP tumor in fasted state in our first submission. In this revised manuscript, lipidomics of KP lung tumor and serum from tumor bearing mice at fasted state was provided as suggested. In contrast to KL lung tumors, G6PD loss had no significant effect on the compositions of fatty acyl groups in KP lung tumors in fasted state (**New supplemental Fig. 6, Page 10, Line 20-21**).

- If changes are only found in the fasted state, why are tracing experiments done in fed state (that actually show changes in Fig 6f and h)?

De novo lipogenesis (DNL) exclusively occurs in the fed state, not during fasting. Therefore, DNL was measured at night 8pm-8am. Despite the reduction in DNL due to G6PD loss, the absorption of fatty acids from dietary sources might play a role in maintaining the consistent fatty acid levels in *G6pd*^{KO};KL tumors in fed state. This was incorporated into the discussion section (**Page 15, Line 22-26, Page 16, Line1-17**):

“Moreover, HFD supplementation rescued KL lung tumor growth caused by G6PD ablation, indicating that less fat availability due to reduced *de novo* fatty acid synthesis may contribute to the slower growth of *G6pd*^{KO};KL lung tumors. Fatty acyl groups composition in KL lung tumors was altered by G6PD loss at fasted state with a decrease in long-chain fatty acyl groups (C14, C16) and an accumulation of very long-chain fatty acyl groups ($\geq C18$) in *G6pd*^{KO}; KL lung tumors. The amount of long-chain and very long-chain fatty acids is intricately linked to various cellular processes, including *de novo* lipogenesis, dietary intake and elongation. Long-chain fatty acids have dual source—dietary intake and *de novo* synthesis, while very long-chain fatty acids come from both dietary sources and elongation⁴¹. The reduction in long-chain fatty acyl groups can be attributed to the reduction in *de novo* synthesis due to a decrease in NADPH generation caused by G6PD loss, as evidenced by *in vivo* D₂O tracing and in *G6pd*^{KO};KL TDCLs through *in vitro* [U-¹³C₆]-glucose labeling. Following the reduction in *de novo* synthesis, the very long-chain fatty acids from dietary sources accumulate, this phenomenon is in line with findings in other contexts where

inhibition of endogenous *de novo* lipogenesis led to the accumulation of dietary very long-chain fatty acids⁴². Moreover, certain polyunsaturated very long-chain fatty acids are recognized for their antioxidant properties^{43,44}, and *G6pd*^{KO};*KL* lung tumors may favor the accumulation of polyunsaturated very long-chain fatty acids as a compensatory mechanism to counteract G6PD loss-induced oxidative stress. Various pathological conditions, including childhood adrenoleukodystrophy⁴⁵, Zellweger syndrome⁴⁶, and colorectal cancer⁴⁷, have been reported to exhibit the accumulation of very long-chain fatty acids. Further investigation is needed to understand how this composition change is associated with the slow tumor growth observed in the absence of G6PD. Moreover, despite the reduction in the *de novo* lipogenesis due to G6PD loss, the absorption of fatty acids from dietary sources in fed state might play an important role in maintaining the fatty acid levels in *G6pd*^{KO};*KL* tumors for tumor growth.”

Mouse group sizes should be given in the figure legends!

Thanks for your suggestion. This was added in the figure legends.

To increase evidence for p53 activation in the KL model (as claimed in the abstract) the authors should show regulation (from RNA-seq data or with qPCR) of canonical p53 target genes in KL;*G6pd*KO vs WT tumours. For instance, apoptosis regulators (Puma, Bax, Bak) or the ones contributing to the GSEA enrichment in Fig 5a.

Thanks for your suggestion. We have incorporated a heatmap illustrating the relative expression of genes contributing to the GSEA enrichment of "GOBP positive regulation of intrinsic apoptotic signaling pathway by p53 class mediator" (**New Fig. 4a, bottom panel**). Our analysis indicates a general trend of increased expression in *G6pd*^{KO};*KL* lung tumors compared to *G6pd*^{WT};*KL* lung tumors for most genes, but there is no statistically significant difference on expression of those genes between *G6pd*^{KO};*KL* and *G6pd*^{KO};*KL* lung tumors.

As Nrf2 is a central transcription factor regulating oxidative stress response and PPP genes, it would be interesting to include it in the cBioportal analysis.

NRF2 was included in the cBioportal analysis as suggested. We observed that high expression level of NRF2 was associated with poorer survival in patients with KRAS/LKB1 co-mutations (**New Supplemental Fig. 1c**), but not in patients with KRAS/TP53 co-mutations (**New Supplemental Fig. 1d, Page 5, Line 19-23**).

If the authors could provide evidence that G6pd inhibition results in reduced tumour growth in *G6pd*WT;*KL* model or in the tumor-derived cell lines, this would additionally support the translational value of the manuscript.

This is an excellent suggestion. However, there's no specific G6PD inhibitor available for *in vivo* study. Therefore, we performed *in vitro* study using G6PDi-1 (PMID: 32393898, Cayman #31484) and confirmed that *G6pd*^{WT};*KL* TDCLs are sensitive to G6PDi-1. Please note, one of *G6pd*^{KO};*KL* TDCL clone show weak response to G6PDi-1. This could be due to high sensitivity to drug toxicity. New data were added in **New Fig. 3g, Page 7, Line 26 and Page 8, Line 1**.

The results should be discussed in the light of an earlier paper showing (Jiang et al, 2011, PMID: 21336310).

Thanks for your suggestion. This paper was discussed in the revised manuscript in **Page 14, Line 26; Page 15, 1-8**:

“The p53 protein binds to G6PD and prevents the formation of the active dimer. p53 loss releases G6PD-inhibitory activity, potentially increasing PPP glucose flux in tumor cells¹³. However, we found that loss of G6PD-mediated oxPPP has no impact on KP lung tumorigenesis. To overcome this loss of G6PD, KP lung tumors may employ a strategy to boost NADPH production through alternative pathways like ME1, IDH1, or folate metabolism. This compensatory NADPH generation could also be accompanied by an alternative source of ribose-phosphate, likely through the non-oxPPP. Additionally, KP lung tumors could obtain lipids and/or nucleosides from the surrounding microenvironment or bloodstream, thereby reducing their dependence on G6PD-derived products. Comprehensive mechanistic studies are needed to fully understand this resilience.”

Fig 3b: KO tumours seem to be G6pd-positive on the tumour margins. How can this be explained? Fig 3i: Why is downstream RAS and mTORC1 signaling reduced? Fig 6m and 6o: Is the difference between HFD WT vs HFD KO significant? Fig 7: Can the authors discuss how G6pd can influence serine metabolism despite higher 3PG levels?

Due to complicated tumor microenvironment, numerous cells with intact G6PD, including lymphocytes, stromal cells and normal adjacent lung tissues, will be present in or around the tumor, exhibiting G6PD positivity through IHC staining. In addition, bulk KL lung tumor mRNA-seq data show the reduced *G6pd* mRNA expression in *G6PD*^{KO};*KL* tumors than *G6PD*^{WT};*KL* tumors (**New supplemental Fig. 2b**). The remaining G6PD mRNA expression could come from other cells in tumor microenvironment, including stromal cells, infiltrated immune cells, and adjacent normal lung tissues. This can be reflected by *Lkb1* mRNA expression in both *G6pd*^{WT};*KL* and *G6pd*^{KO};*KL* lung tumors (**New supplemental Fig. 2b**). For the mouse lung tumor derived cell lines (TDCLs), G6PD deletion was validated by Western blot (**New Fig. 3b**).

We utilized downstream RAS and mTORC1 signaling markers (pErk and pS6) along with Ki67 to signify slow tumor growth caused by G6PD loss, as further elaborated in the text (**Page 6, Line 19-22**).

Tumor number was significantly lower in mice bearing *G6pd*^{KO};*KL* tumors than mice bearing *G6pd*^{WT};*KL* when mice were fed with HFD (***) $P < 0.001$ was added in **New Fig. 5n**). However, there was no significant difference of tumor burden between mice bearing *G6pd*^{WT};*KL* and *G6pd*^{KO};*KL* lung tumors. This suggests that HFD mainly rescues *G6pd*^{KO};*KL* tumor by promoting tumor growth, not tumor initiation. Thank you for bringing this to our attention.

As suggested, we further discussed how G6PD can influence serine metabolism (**Page 17, Line 22-26; Page 18, Line 1-9**): “In the case of KL lung tumors, G6PD loss alters serine metabolism by decreasing serine biosynthesis and increasing serine uptake in KL lung tumors. The redox status has a significant impact on enzyme activity in various metabolic pathways, including those associated with serine metabolism. In the context of serine biosynthesis, 3-phosphoglycerate dehydrogenase (PHGDH) acts as a key enzyme, facilitating the conversion of 3PG to phosphohydroxypyruvate. The enzymatic activity of PHGDH is intricately connected to the NAD⁺/NADH ratio^{47,48}. G6PD deficiency observed in KL tumors has a significant impact on NADPH availability, disrupting redox equilibrium. This disruption could potentially affect NAD⁺/NADH ratio and impair serine biosynthesis, resulting in the accumulation of 3PG. Simultaneously, during tumor progression, G6PD-deficient cells increase serine uptake to maintain serine-driven one-carbon metabolism as an alternative NADPH source. Hence, our study also proposes an innovative therapeutic approach for treating KL lung cancer by combining G6PD inhibitors with a serine/glycine depletion diet. However, in addition to its role in generating cytosolic NADPH, serine-mediated one-carbon metabolism is vital for nucleotide metabolism. Further mechanistic investigations are required to validate the effectiveness of this combination in in vivo cancer treatment.”

Reviewer #3 (Remarks to the Author):

In the manuscript by Taijin Lan et al. titled “G6PD Maintains Redox Homeostasis and Biosynthesis in LKB1-Deficient KRAS-Driven Lung Cancer”, the authors studied the reliance of different genetic subtypes of non-small cell lung cancer on G6PD activity. Previous studies from this group revealed that KRAS mutant/p53 loss (KP) tumors were unaffected by G6PD disruption (Ghergurovich JM, 2020). Now, the authors show that, in contrast to KP tumors, KRAS mutation/LKB1 loss (KL) tumors are heavily dependent on G6PD. The authors identify several alterations in KL tumors elicited by G6PD disruption, which include a decrease in NADPH levels, induction of oxidative stress, and activation of p53. These results provided important mechanistic insight into the differences between KP and KL. They also showed that G6PD deletion promotes an alteration in lipid abundance, and lipid supplementation through a high-fat diet can rescue growth defects in KL tumors. Further, they showed that G6PD deletion specifically rewires serine metabolism, increasing the flux of extracellular serine to the one-carbon metabolism, generating NADPH from one-carbon units donated by serine. This study is interesting; however, several technical and conceptual concerns should be addressed.

Thanks for your valuable feedback. We have incorporated your suggestions into the revised manuscript.

Major concern

1. In Figure 1, the authors state, “These results suggest that G6PD and MTHFD1 expression impact survival in a subset of lung cancer patients (KRAS/LKB1 co-MUT and not KRAS/TP53 co-MUT lung cancers).” However, the data provided does not support this statement. The data suggests that G6PD and MTHFD1 expression are correlated with survival. These findings are potentially better suited as a Supplementary Figure instead of a main Figure.

Thanks for your suggestion. We have moved this data as a **Supplemental Figure 1**. In addition, in response to the suggestion by Reviewer 2, we have included NRF2 in the cBioportal analysis. The corresponding text has been modified to reflect these changes (**Page 5, Line 15-17**).

“Regarding MTHFD1, besides its connection with survival outcomes in lung cancer patients with WT KRAS (Fig. 1a), its high expression is also associated to poorer survival in patients with KRAS/LKB1 co-mutations (Supplemental Fig. 1c).”

2. A major question is why G6PD loss does not alter NADPH levels in KP lung tumors. This is significant because the authors suggest that in KL tumors, G6PD loss causes a drop in NADPH levels, oxidative stress, p53 induction, and slow tumor growth. Based on this rationale, G6PD loss should decrease NADPH levels independent of p53 status. This point should be addressed. Further, it would be informative to know whether loss of G6PD in KPL tumors impacts NADPH and NADP⁺ levels.

We have further discussed the possible reasons that KP lung tumors overcome the loss of G6PD to facilitate growth. One potential strategy involves enhancing the flow through alternative pathways for NADPH production, such as ME1, IDH1, or folate metabolism. This compensatory NADPH generation could also be accompanied by an alternative source of ribose-phosphate, likely through the non-oxidative pentose phosphate pathway. Additionally, KP lung tumors could obtain lipids and/or nucleosides from the surrounding microenvironment or bloodstream, thereby reducing the demand for products from the G6PD reaction. Unraveling the underlying reasons for this resilience requires further in-depth mechanistic studies. This has been incorporated into discuss section (**Page 15, Line 2-8**): “To overcome G6PD loss, KP lung tumors may employ a strategy to boost NADPH production through alternative pathways like ME1, IDH1, or folate metabolism. This compensatory NADPH generation could also be accompanied by an alternative source of ribose-phosphate, likely through the non-oxPPP. Additionally, KP lung tumors could obtain lipids and/or nucleosides from the surrounding microenvironment or bloodstream, thereby

reducing their dependence on G6PD-derived products. Comprehensive mechanistic studies are needed to fully understand this resilience.” We are currently exploring the compensatory mechanism of NADPH generation in KP lung tumors. This will be included in our future publication.

Rather than assessing the NADPH/NADP⁺ and GSH/GSSG ratios in KPL lung tumor, a process requiring fresh tumor tissues to extract polar metabolites, a process of minimum of 6-8 months (inclusive of mice breeding, tumor induction, and metabolomics), we opted to perform IHC of NRF2 and NQO1, markers indicative of oxidative stress, using available KPL tumor paraffin sections. We observed increased oxidative stress in *G6PD*^{KO};KPL lung tumors compared to *G6PD*^{WT};KPL lung tumors (**New Fig. 4I, Page 9, Line 6-7**). This suggests that the loss of G6PD triggers redox imbalance and oxidative stress, and this occurrence is independent of p53. This also suggests that oxidative stress alone, without p53, is not sufficient to slow KL lung tumor growth. It underscores that the slower growth of *G6PD*^{KO};KL tumors may be attributed to p53 activation, possibly triggered by oxidative stress. This has also been incorporated into discuss section (**Page 16, Line 23-26; Page 17, Line 1-2**): “The loss of G6PD significantly increases oxidative stress in KL lung tumors, potentially leading to the activation of p53 and the upregulation of its downstream targets to impede tumor growth. Our findings indicate that the reduction of KL lung tumors by G6PD ablation is rescued by the absence of p53. Despite this, increased oxidative stress persists in *G6pd*KO;KPL lung tumors. These findings suggest that oxidative stress alone, without p53, is not sufficient to impede KL lung tumor growth. Thus, the slow growth of G6PD-knockout KL lung tumors is attributed to p53 activation inhibiting tumor progression.”

3. Several technical details need to be provided in the Figures and Figure Legends. For example, the authors show a GSEA plot referring to an “Oxidative stress” signature, but no other data is provided referring to how this signature was generated. Is this a “Hallmark” gene set or a “GO Biological Pathway”? Further, measuring the levels of specific NRF2 target genes, such as *Nqo1* and *Hmox1*, would be informative as a readout of oxidative stress. Later, in Figure 4i, the authors show a graph with an unlabeled y-axis. They mention that this is a “proliferation rate,” but additional details should be included in the graph and Figure legend. Also, the author should include in the Figure legends the number of animals used in each experiment.

Thanks for your suggestions. We have updated figure legend with more details. The y-axis label has been added to the Incucyte data (**New Fig. 3f**).

Gene sets information has added in figure legends and method section (**Page 22, Line 4-8**):

“the gene set for “Oxidative stress” was downloaded from GeneCards (<https://www.genecards.org/>, accessed on April 09, 2023), and gene sets for “GOBP positive regulation of intrinsic apoptotic signaling pathway by p53 class mediator”, “GOBP lipid biosynthetic process” and “GOBP fatty acids biosynthetic process” were downloaded from MSigDB website (<https://www.gsea-msigdb.org/>, accessed on April 09, 2023).”

To further confirm that G6PD loss in KL lung tumors impairs redox homeostasis, we performed IHC of NRF2 and NRF2 target NQO1. We found that G6PD deficiency significantly increased NRF2 and NQO1 expression in both KL and KPL lung tumors. This new data were added (**New Fig. 2d, Page 7, Line 11-12; New Fig. 4I, Page 9, Line 6-7**).

4. Some technical approaches could be clearer. They mention for “mRNA-seq” that “The lung tumors were rapidly dissected and snap-frozen in liquid nitrogen.” In Figures 2 and 3, the authors report upwards to hundreds of tumors per mouse in the lung tissue. It is unclear how these tumors were individually isolated, not to include non-tumor lung tissue.

The tumors are manually dissected from the mouse lung, inevitably including normal lung tissues. We

have made our best efforts to collect the majority of tumor tissues for mRNA-seq. Method was modified as “Efforts have been made to collect the predominant portion of tumor tissues from each mouse lung” (Page 21, Line 24-25).

5. The authors state, “Thus, the slow growth of G6PD-knockout KL tumors is due to oxidative stress-inducing p53 activation and p53 activation inhibiting tumor progression.” But this isn’t shown. The authors demonstrate that the slow growth of G6PD-knockout KL tumors is due to p53, but they have not rescued the oxidative-stress phenotypes they observe in G6PD-knockout KL tumors; thus, do not know if it is oxidative stress that is inducing p53 activation. The authors could test this using antioxidant supplementation. Alternatively, the authors could change the writing to state the slow growth of G6PD-knockout KL tumors is due to p53 activation inhibiting tumor progression”.

Thanks for your suggestion. We have modified the text as “the slow growth of G6PD-knockout KL tumors is due to p53 activation inhibiting tumor progression” (Page 9, Line 8-9).

6. Conceptually, the manuscript is hard to follow. The Introduction section suggests that the authors will address how different genetic drivers can impact NSCLC reliance on G6PD, and the authors investigate this in Figures 2-5. However, in Figures 6 and 7, they introduce a new line of investigation into how NADPH modulates lipid synthesis and induces a reliance on serine abundance. These findings appear disjointed from the remaining Figures and reduce the clarity of the overall study.

We have improved the manuscript to establish a more coherent structure. We also created a model figure (New Fig. 6q, Page 14, Line 2-7) to depict G6PD-mediate KL lung tumorigenesis. “Specifically, in KL lung tumors, G6PD-mediated oxPPP sustains the NADPH pool, crucial for maintaining redox balance and supporting lipid metabolism, and prevents p53 activation-induced cell death. Loss of G6PD in KL lung tumors triggers a shift in serine metabolism, increasing serine uptake to maintain one-carbon metabolism-driven NADPH production as an alternative. This, in turn, maintains redox homeostasis, facilitating the eventual progression of G6PD-deficient KL lung tumors (Fig. 6q)”

7. Some aspects of the study are not fully discussed. They mention, “Compared with G6pdWT; KL lung tumors, G6pdKO; KL tumors had significantly lower levels of long-chain fatty acyl groups, whereas very long-chain fatty acyl groups accumulated in G6pdKO; KL lung tumors (Fig. 6d).” These findings are not discussed. How do authors interpret the accumulation of very long-chain FA over long-chain FA? Is this observed in any disease? The authors should speculate on the functional consequences of these phenotypes. Further, did the high-fat diet rescue this lipidomic imbalance?

Thanks for your suggestion. We have fully discussed this in Page 15, Line 24-26, Page 16, Line 1-17:

“Fatty acyl groups composition in KL lung tumors was altered by G6PD loss at fasted state with a decrease in long-chain fatty acyl groups (C14, C16) and an accumulation of very long-chain fatty acyl groups (\geq C18) in *G6pd*^{KO}; KL lung tumors. The amount of long-chain and very long-chain fatty acids is intricately linked to various cellular processes, including *de novo* lipogenesis, dietary intake and elongation. Long-chain fatty acids have dual source—dietary intake and *de novo* synthesis, while very long-chain fatty acids come from both dietary sources and elongation⁴¹. The reduction in long-chain fatty acyl groups can be attributed to the reduction in *de novo* synthesis due to a decrease in NADPH generation caused by G6PD loss, as evidenced by *in vivo* D₂O tracing and in *G6pd*^{KO};KL TDCLs through *in vitro* [U-¹³C₆]-glucose labeling. Following the reduction in *de novo* synthesis, the very long-chain fatty acids from dietary sources accumulate, this phenomenon is in line with findings in other contexts where inhibition of endogenous *de novo* lipogenesis led to the accumulation of dietary very long-chain fatty acids⁴². Moreover, certain polyunsaturated very long-chain fatty acids are recognized for their antioxidant properties^{43,44}, and *G6pd*^{KO};KL lung tumors may favor the accumulation of polyunsaturated very long-

chain fatty acids as a compensatory mechanism to counteract G6PD loss-induced oxidative stress. Various pathological conditions, including childhood adrenoleukodystrophy ⁴⁵, Zellweger syndrome ⁴⁶, and colorectal cancer ⁴⁷, have been reported to exhibit the accumulation of very long-chain fatty acids. Further investigation is needed to understand how this composition change is associated with the slow tumor growth observed in the absence of G6PD. Moreover, despite the reduction in the *de novo* lipogenesis due to G6PD loss, the absorption of fatty acids from dietary sources in fed state might play an important role in maintaining the fatty acid levels in *G6pd*^{KO};*KL* tumors for tumor growth. ”

We performed lipidomics of lung tumors and serum from tumor-bearing mice at fasted state at 7-week post tumor induction. HFD significantly increased the levels of fatty acids in serum of *KL* tumor bearing mice, but had no impact on fatty acyl composition and levels of *G6pd*^{WT};*KL* lung tumors. Due to the minimal tumor burden of *G6pd*^{KO};*KL* lung tumors at 7 weeks post tumor induction in normal diet (ND), we were unable to collect *G6pd*^{KO};*KL* lung tumors for lipidomics. Despite this, we were able to collect *G6pd*^{KO};*KL* lung tumors in HFD. Therefore, we compared fatty acyl group levels between *G6pd*^{WT};*KL* and *G6pd*^{KO};*KL* lung tumors under HFD conditions. The level of C16:0 is comparable between *G6pd*^{KO};*KL* lung tumors and *G6pd*^{WT};*KL* lung tumors under HFD (**Supplemental Fig. 7e, f**). However, the levels of many very long-chain fatty acyl groups in *G6pd*^{KO};*KL* lung tumors were lower than those in *G6pd*^{WT};*KL* lung tumors under HFD conditions (**Supplemental Fig. 7e**). This suggests that HFD partially rescue the alterations in fatty acyl groups pool size levels caused by G6PD loss. New data were added in **Supplemental Fig. 7a-f, Page 11, Line 3-14**.

8. It is unclear why only four mice are shown in Figures 5E, F, H, and I if Fig 5K presents data from more than 25 mice of the same genotype.

We randomly sacrificed 4 mice from each group at 6 weeks post-tumor induction for tumor burden analysis, as depicted in **New Fig. 4i**. Since no significant differences in tumor burden were observed, the remaining mice were retained for mouse survival analysis (**New Fig. 4k**), contributing to a more robust conclusion.

9. As Vitamin C is paradoxically an antioxidant or a pro-oxidant depending on the dose, demonstrating that Vitamin C is causing oxidative stress/damage in your model (e.g., 8-oxo-dG, 4-HNE, or even mRNA expression of NRF2 targets) would strengthen the data.

Thanks for your suggestion. We performed IHC of NRF2, NQO1 and 8-oxo-dG, markers for oxidative stress, in Vitamin C treated allograft tumors. IHC analysis of NRF2, NQO1 and 8-oxo-dG confirmed increased oxidative stress in *KL* allografts with high-dose Vit C treatment, with *G6pd*^{KO};*KL* allograft tumors exhibiting higher oxidative stress than *G6pd*^{WT};*KL* allografts, further intensified by high-dose Vit C (**New Fig. 3m, Page 8, Line 12-14**).

Minor concerns

1. It is unclear why the authors only use IHC and not qPCR to confirm G6PD deletion in tumors.

The G6PD antibody used for IHC has been validated in our previous publication (PMID: 32661137). Due to complicated tumor microenvironment, obtaining pure tumor samples for Western blot or qPCR is unfeasible. Therefore, IHC is expected to yield more robust results *in vivo* than Western blot or qPCR. In addition, we have provided *G6pd* mRNA expression from *KL* lung tumor RNA-seq data to show the reduced G6PD mRNA expression in *G6pd*^{KO};*KL* lung tumors than *G6pd*^{WT};*KL* lung tumors (**New Supplemental Fig. 2b**). The remaining *G6pd* mRNA expression could be from other cells in tumor microenvironment, including stromal cells, infiltrated immune cells, and adjacent normal lung

tissues. This is supported by *Lkb1* mRNA expression in both *G6pd*^{WT};*KL* and *G6pd*^{KO};*KL* lung tumors (**New Supplemental Fig. 2b**). Furthermore, the Cre-Lox system represents a well-established model for investigating gene knockout in KRAS-driven non-small cell lung cancer (NSCLC). For mouse lung tumor derived cell lines, we have provided Western blot to confirm G6PD deletion (**New Fig. 3b**).

2. At Line 126, a reference is needed for the statement: "Tumors exhibit an enormous demand for NADPH due to uncontrolled proliferation."

Reference was provided.

3. Lines 236-237 appear to have truncated text. Please verify.

Thank you for bringing this to our attention. This was corrected in the revised manuscript.

4. It would be informative to present levels of oxPPP metabolites in Fig 7B-C-D.

Thank you for the suggestion. We revisited the LC-MS raw metabolomics data and confirmed that, apart from glucose-6-phosphate (G6P), the signal of other oxPPP intermediates, such as 6-phosphogluconolactone, 6-phosphogluconate, and ribulose-5-phosphate (Ru5P), are extremely low, making reliable signal peaks challenging to obtain via LC-MS. Only the non-oxPPP intermediate ribose-5-phosphate was detected, showing no significant difference between *G6pd*^{WT};*KL* and *G6pd*^{KO};*KL* lung tumors (**New Supplemental Fig. 8d**).

5. It would be informative to present NADPH levels of TDCL in Ser/Gly-free media.

Thanks for suggestion. The levels of NADPH, NADP⁺ and NADPH/NADP⁺ ratio were provided in **New Fig. 6m, Supplemental Fig. 10d, Page 13, Line 11-13**. "Serine/glycine depletion significantly decreased the NADPH level and NADPH/NADP⁺ ratio in *G6pd*^{KO};*KL* TDCLs, while no such effect was observed in *G6pd*^{WT};*KL* TDCLs."

6. At the end of Figure 2's legend, there is a mention of D2O infusion, which seems to be a mistake.

Thank you for bringing this to our attention. We have rectified this mistake.

Reviewer #4 (Remarks to the Author):

This study by Lan et. al investigates impact of *G6pd* loss in NSCLC harboring co-mutations in KRAS and LKB1 (KL). The authors demonstrate that G6PD is important for KL tumorigenesis as well as cellular NADPH production. Further, the authors identify serine-glycine one-carbon (SGOC) metabolism as a key NADPH generating source under G6PD suppression in KL tumors. Loss of G6PD in KL tumors drives upregulation of SGOC metabolism, which drives increased NADPH production for antioxidant defenses. The finding that G6PD is selectively required for KL tumorigenesis and G6PD loss reprograms SGOC metabolism is interesting.

1. In Fig.2, the authors claim that G6PD is not required for NADPH and redox control in KP tumors. Given the critical role of G6PD in cytosolic NADPH production, there might be some compensatory mechanisms to maintain NADPH and GSH pools in KP tumors. It would be important to understand contribution of IDH1 and ME1 in cellular NADPH production in KP-G6pd WT and KO tumors to confirm 1) G6PD is not the major NADPH generating machinery in KP tumors and 2) test whether G6PD loss reprograms contribution of other cytosolic NADPH sources. 1-2H-glucose (G6PD), 2,3,3,4,4-2H-glutamine (IDH1),

and 2,3,3-2H-aspartate (ME1) could be used to label cells.

We fully agree the existence of compensatory mechanisms that maintain the pools of NADPH and glutathione (GSH) in KP lung tumors. Currently, we are actively investigating such compensatory pathways. We have generated *Me1^{Flox/Flox};KP* and *G6pd^{Flox/Flox};Me1^{Flox/Flox};KP* GEMMs and found that loss of ME1 alone (Fig. 1a-e) or loss of ME1 and G6PD together (Fig. 1f, g) had no impact on KP lung tumor growth and mouse survival. This suggest that ME1 and G6PD do not appear to mutually compensate. Additionally, we explored the role of IDH1 in KP lung tumorigenesis via multiplex CRISPR/Cas9 system. Lentivirus expressing control sgTom-Cre, or expressing *sgldh1-1_Sgldh1-2-Cre* were intranasally delivered to *Kras^{LSL_G12D/+};p53^{flox/flox};Cas9-GFP^{flox/flox}* (KPC) mice to induce lung tumor and knock out *Idh1* simultaneously (Fig. 1h, i). We found that IDH1 ablation alone had no effect on KP lung tumorigenesis (Fig. 1j-l). Thus, the regulation of cytosolic NADPH homeostasis in KP lung tumors appears more intricate than anticipated, underscoring the necessity for additional investigation.

As recommended by the reviewer, labeling cells with 1-2H-glucose (G6PD), 2,3,3,4,4-2H-glutamine (IDH1), and 2,3,3-2H-aspartate (ME1) is a possibility. However, it's important to note that *in vitro* metabolism may differ from *in vivo* metabolism. We are currently collaborating with the Rabinowitz lab to develop *in vivo* labeling methods. Hence, we prefer to focus on exploring compensatory mechanisms for maintaining cytosolic NADPH homeostasis in KP lung tumors *in vivo*, a scope that extends beyond the focus of this manuscript.

As a point of reference, we have provided tumor burden and mouse survival analyses upon ME1 or IDH1 deletion or co-deletion of ME1 and G6PD in KP lung tumors. These findings will be included in a future publication concentrating on cytosolic NADPH modulation in KP lung tumors.

Fig. 1 The role of ME1 and IDH1 in KP lung tumorigenesis

2. In relation to the previous point, the authors should check NADPH production from IDH, ME1 as well as SGOC in KL-G6pd WT and KO TDCL. IDH1 and ME1 contribution for NADPH production would be hard to determine from U-13C glucose.

Thanks for your suggestion. We agree that [¹³C₆]-Glucose is not suitable for assessing the contribution of IDH1 and ME1 to NADPH production. However, there is currently no suitable tracer for *in vivo* tracking of NADPH generation specifically mediated by ME1 or IDH1. Indeed, we have generated *Me1^{Flox/Flox};KL* GEMM and found that ME1 loss does not affect KL lung tumorigenesis. It is plausible that G6PD loss may induce compensatory NADPH production through ME1 or IDH1. Currently, we are generating *G6pd^{Flox/Flox};Me1^{Flox/Flox};KL* GEMM to assess this possibility. This is beyond the scope of this manuscript.

We have identified that serine-mediated one-carbon metabolism compensates for G6PD loss in KL cell survival. Therefore, this manuscript now focuses on the reprogramming of serine metabolism mediated by G6PD loss in KL lung tumor. We have expanded the discussion to include the potential compensatory roles played by ME1 or IDH1 (**Page 17, Line 8-16**): “Our *in vivo* isotope tracing and flux analysis revealed that G6PD deficiency in KL lung tumors does not affect glucose carbon flux to tumor pyruvate, lactate, and TCA cycle intermediates. However, G6PD loss in KL lung tumors reduces glucose carbon flux to serine. Additionally, serine uptake is increased to maintain the serine pool size level in G6PD-deficient KL lung tumors for cytosolic NADPH production. We found that in *in vitro* cell culture, increased serine uptake is used to maintain redox homeostasis for cell proliferation. Therefore, serine-mediated one-carbon metabolism compensates for G6PD loss in KL cancer cell survival, although this does not preclude the potential compensatory cytosolic NADPH production through ME1 or IDH1.”

Additionally, we had new data to show that serine/glycine depletion led to a decrease of NADPH level and NADPH/NADP+ ratio in *G6pd*^{KO};KL TDCLs, while no such effect was observed in *G6pd*^{WT};KL TDCLs, suggesting that serine-mediated one carbon mechanism compensates G6PD loss for NADPH production (**New Fig. 6m, Supplemental Fig. 10d, Page 13, Line 11-13**).

3. In Fig.1, although ME1 expression is not ‘statistically’ significantly associated with prognosis, it does seem to have biological meaning; graph looks almost identical to that of G6PD. The authors might want to mention about it.

We have mentioned it according to your suggestion in discussion section: “while the mRNA expression of ME1 is not statistically significantly linked to prognosis in patients with co-mutations of KRAS and LKB1, it appears to hold biological significance” (**Page 18, Line 17-19**).

4. Based on KM graph and mRNA expression data in Fig1c,d,e, mRNA expression is not always a good readout for prognosis. It would be important to discuss in the manuscript (e.g., what would be the authors’ thought?).

Thanks for your suggestion. We agree that mRNA expression is not always a good readout for prognosis. In response to the suggestion from reviewer 3, we have moved original Figure 1 as a supplemental data (**New Supplemental Fig. 1**). We also incorporated this in discussion section (**Page 18, Line 19-25**):

“Analyzing tumor mRNA expression is common in cancer research for prognostic insights. However, relying solely on this for prognosis may not always provide a comprehensive assessment due to factors like post-transcriptional modifications, tumor heterogeneity, microenvironmental influences, the dynamic nature of cancer, and treatment response. It’s crucial to interpret mRNA data cautiously and integrate it with other information for a more thorough understanding of cancer prognosis. Therefore, the combination of cBioPortal data analysis with findings from our preclinical mouse study suggests that patients harboring co-mutations of KRAS and LKB1 may benefit from G6PD inhibitor therapy.”

REVIEWER COMMENTS

Reviewer #1 (Remarks to the Author):

The authors have satisfactorily addressed all our concerns, and the revised manuscript now includes a significant amount of new data that strengthens the main conclusions. The authors should be commended for this effort.

Before publication, there are a few minor critiques to address:

- 1) The authors still need to clarify the names of the TDCLs. They can simplify them further.
- 2) By comparing the lipidomic profile of KL lung tumour-bearing mice at fed state (old Supplementary 1b vs new Supplementary 5d) or KP lung tumour-bearing mice at fed state (old Supplementary 1d vs new Supplementary 6d), we realized that the authors selected specific tumour samples from the previous analysis to be included in the new analysis, while serum samples remained the same. For instance, in Sup. Fig. 5d T6, T4, T1, T3, T2 and T5 corresponds to the old 3096 T3, 3096 T1, 3094 T2, 3095 T2, 3094 T3, 3096 T2 from Sup. Fig. 1b, respectively. How did they select and name the new samples? Why serum and tumours from different mice have been used for the lipidomic analysis?
- 3) The scheme at Fig. 6q should be updated
- 4) Statistics is missing in the following panels:

Main Figures:

1d, 1f, 1g, 1h, 1k, 1n (7 weeks)
2a (NADP+), 2b (GSH)
3c (NADP+), 3g (KO, 2489 1-9), 3i (WT), 3j, 3l
4f, 4h, 4i, 4j
5l, 5o (7 weeks)
6b, 6c, 6d, 6f, 6g, 6m, 6n, 6o (WT), 6p (WT)

Supplementary Figures:

1e (MTHFD1)
2b (Lkb1)
3a, 3b
4a, 4b, 4c
5a, b (M+6, M+8)
7f (tumour C16:0)
8c
9b (succinic acid, aspartate)
10d, b

- 5) correct some typos

- line 55 (abstract): "NAPDH" change to "NADPH"
- line 78, 183, 190, 191, 199, 345, 375, 464, 466, also in Fig 2a, Fig 3c, Fig 6m : the „+“ needs to be superscript
- line 143: „are occurred“ needs to be changed to „occurr“
- line 183: the „+“ needs to be superscript; also in Fig 2a, and Fig 3c, Fig 6m
- line 190, 191: „+“ needs to be superscript; space missing before bracket
- line 209: it should be: „are more susceptible“
- line 312: an „of“ is missing before „other“
- line 349: a space is missing before the bracket

Main Figures

- 3a: remove lorem ipsum
- 3e: correct CM-H2DCFDA, y axis
- 3f: correct confluence, y axis
- 4b: graph p53 with small letters)
- 5d: correct NaCl
- 6a: correct acetyl-CoA, aspartate
- 6g: correct increase
- 6p: proliferation rate

Supplementary Figures

- 1e: remove G6PD from y axis
- 2a: Lenti-Cre, cDNA synthesis and library
- 2b: Lbk1
- 8c: aspartate

Reviewer #2 (Remarks to the Author):

This extensive revision addresses all comments very well. There are have no further remarks.

Reviewer #3 (Remarks to the Author):

In the revised manuscript by Taijin Lan et al. titled "G6PD Maintains Redox Homeostasis and Biosynthesis in LKB1-Deficient KRAS-Driven Lung Cancer", the authors have not addressed several points outlined by the reviewer.

Point 2:

"2. A major question is why G6PD loss does not alter NADPH levels in KP lung tumors. This is significant because the authors suggest that in KL tumors, G6PD loss causes a drop in NADPH levels, oxidative stress, p53 induction, and slow tumor growth. Based on this rationale, G6PD loss should decrease NADPH levels independent of p53 status. This point should be addressed. Further, it would be informative to know whether loss of G6PD in KPL tumors impacts NADPH and NADP+ levels."

Regarding measuring NADPH and NADP+ levels, the authors' reply is:

"Rather than assessing the NADPH/NADP+ and GSH/GSSG ratios in KPL lung tumor, a process requiring fresh tumor tissues to extract polar metabolites, a process of minimum of 6-8 months (inclusive of mice breeding, tumor induction, and metabolomics), we opted to perform IHC of NRF2 and NQO1, markers indicative of oxidative stress, using available KPL tumor paraffin sections. We observed increased oxidative stress in G6PDKO;KPL lung tumors compared to G6PDWT;KPL lung tumors (New Fig. 4I, Page 9, Line 6-7). This suggests that the loss of G6PD triggers redox imbalance and oxidative stress, and this occurrence is independent of p53. This also suggests that oxidative stress alone, without p53, is not sufficient to slow KL lung tumor growth. It underscores that the slower growth of G6PDKO;KL tumors may be attributed to p53 activation, possibly triggered by oxidative stress. This has also been incorporated into discuss section (Page 16, Line 23-26; Page 17, Line 1-2): "The loss of G6PD significantly increases oxidative stress in KL lung tumors, potentially leading to the activation of p53 and the upregulation of its downstream targets to impede tumor growth. Our findings indicate that the reduction of KL lung tumors by G6PD ablation is rescued by the absence of p53. Despite this, increased oxidative stress persists in G6pdKO;KPL lung tumors. These findings suggest that oxidative stress alone, without p53, is not sufficient to impede KL lung tumor growth. Thus, the slow growth of G6PD-knockout KL lung tumors is attributed to p53 activation inhibiting tumor progression."

The authors have misunderstood Point 2. NADPH and NADP+ levels should be measured in KPL tumors and compared to KPL tumors with loss of G6PD.

Point 3:

"Several technical details need to be provided in the Figures and Figure Legends....Later, in Figure 4i, the authors show a graph with an unlabeled y-axis. They mention that this is a "proliferation rate," but additional details should be included in the graph and Figure legend. "

Figure 3f is still unclear. The authors state this is "Proliferation rate", but only "Confluence %" is provided. It is unclear what the proliferation rate that they are referring to is.

Point 8:

"8. It is unclear why only four mice are shown in Figures 5E, F, H, and I if Fig 5K presents data from more than 25 mice of the same genotype."

The authors' reply is:

"We randomly sacrificed 4 mice from each group at 6 weeks post-tumor induction for tumor burden analysis, as depicted in New Fig. 4i. Since no significant differences in tumor burden were observed, the remaining mice were retained for mouse survival analysis (New Fig. 4k), contributing to a more robust conclusion."

No statistical analysis of Figures 4h and 4i is provided; thus, the authors cannot state that "no significant differences in tumor burden were observed." Further, it appears that with a larger cohort of mice, these differences could potentially become significant.

Reviewer #4 (Remarks to the Author):

The authors have addressed my queries thoroughly and comprehensively. I have no further requests and believe the manuscript is ready for editing, submission of raw data, and if these are suitable, publication.

REVIEWER

COMMENTS

We are grateful to the peer reviewers for their valuable feedback on our manuscript, which has allowed us to enhance its quality. In this second round of revisions, we have carefully incorporated further insights provided by Reviewer #1 and Reviewer #3 and also addressed the comments point-by-point below.

Reviewer #1 (Remarks to the Author):

The authors have satisfactorily addressed all our concerns, and the revised manuscript now includes a significant amount of new data that strengthens the main conclusions. The authors should be commended for this effort.

We greatly appreciate your acknowledgment of our efforts in addressing all concerns and incorporating additional data to strengthen the manuscript.

Thank you very much for your feedback throughout our revision process, as it significantly contributes to improving the quality of our manuscript.

Before publication, there are a few minor critiques to address:

1) The authors still need to clarify the names of the TDCLs. They can simplify them further.

Thank you for your suggestion. TDCLs number was simplified as suggested.

2) By comparing the lipidomic profile of KL lung tumour-bearing mice at fed state (old Supplementary 1b vs new Supplementary 5d) or KP lung tumour-bearing mice at fed state (old Supplementary 1d vs new Supplementary 6d), we realized that the authors selected specific tumour samples from the previous analysis to be included in the new analysis, while serum samples remained the same. For instance, in Sup. Fig. 5d T6, T4, T1, T3, T2 and T5 corresponds to the old 3096 T3, 3096 T1, 3094 T2, 3095 T2, 3094 T3, 3096 T2 from Sup. Fig. 1b, respectively. How did they select and name the new samples? Why serum and tumours from different mice have been used for the lipidomic analysis?

Samples were gathered from two experiments (two batches of mice) for lipidomic analysis. Serum samples were not collected for one of the experiments. During the revision process, tumor samples lacking correlated serum samples were excluded, as this did not impact our final conclusions. However, in this revised version, we believe that including all tumor samples could lead to a more robust conclusion, even without matched serum samples. Thus, we retained the original Supplementary 1d, and T7-T9 represent tumor samples without matched serum (new Supplementary 6d).

3) The scheme at Fig. 6q should be updated

Thank you very much for bringing this to our attention. We have now rectified it.

4) Statistics is missing in the following panels:

Main Figures:

1d, 1f, 1g, 1h, 1k, 1n (7 weeks)

2a (NADP+), 2b (GSH)

3c (NADP+), 3g (KO, 2489 1-9), 3i (WT), 3j, 3l

4f, 4h, 4i, 4j

5l, 5o (7 weeks)

6b, 6c, 6d, 6f, 6g, 6m, 6n, 6o (WT), 6p (WT)

Supplementary Figures:

1e (MTHFD1)

2b (Lkb1)

3a, 3b
4a, 4b, 4c
5a, b (M+6, M+8)
7f (tumour C16:0)
8c
9b (succinic acid, aspartate)
10d, b

In our previous submission, we omitted "ns" from the graphs where no significant difference was observed. In this revised version, "ns" has been included in all figures. Statistics have been updated in all figures as per the request.

5) correct some typos

- line 55 (abstract): "NAPDH" change to "NADPH"
- line 78, 183, 190, 191, 199, 345, 375, 464, 466, also in Fig 2a, Fig 3c, Fig 6m : the „+“ needs to be superscript
- line 143: „are occurred“ needs to be changed to „occur“
- line 183: the „+“ needs to be superscript; also in Fig 2a, and Fig 3c, Fig 6m
- line 190, 191: „+“ needs to be superscript; space missing before bracket
- line 209: it should be: „are more susceptible“
- line 312: an „of“ is missing before „other“
- line 349: a space is missing before the bracket

Main Figures

- 3a: remove lorem ipsum
- 3e: correct CM-H2DCFDA, y axis
- 3f: correct confluence, y axis
- 4b: graph p53 with small letters)
- 5d: correct NaCl
- 6a: correct acetyl-CoA, aspartate
- 6g: correct increase
- 6p: proliferation rate

Supplementary Figures

- 1e: remove G6PD from y axis
- 2a: Lenti-Cre, cDNA synthesis and library
- 2b: Lbk1
- 8c: aspartate

Thank you very much for bringing those typos to our attention. We have now rectified all of them.

Reviewer #2 (Remarks to the Author):

This extensive revision addresses all comments very well. There are no further remarks.

Reviewer #3 (Remarks to the Author):

In the revised manuscript by Tajjin Lan et al. titled "G6PD Maintains Redox Homeostasis and

Biosynthesis in LKB1-Deficient KRAS-Driven Lung Cancer”, the authors have not addressed several points outlined by the reviewer.

Point 2:

“2. A major question is why G6PD loss does not alter NADPH levels in KP lung tumors. This is significant because the authors suggest that in KL tumors, G6PD loss causes a drop in NADPH levels, oxidative stress, p53 induction, and slow tumor growth. Based on this rationale, G6PD loss should decrease NADPH levels independent of p53 status. This point should be addressed. Further, it would be informative to know whether loss of G6PD in KPL tumors impacts NADPH and NADP+ levels.”

Regarding measuring NADPH and NADP+ levels, the authors’ reply is:

“Rather than assessing the NADPH/NADP+ and GSH/GSSG ratios in KPL lung tumor, a process requiring fresh tumor tissues to extract polar metabolites, a process of minimum of 6-8 months (inclusive of mice breeding, tumor induction, and metabolomics), we opted to perform IHC of NRF2 and NQO1, markers indicative of oxidative stress, using available KPL tumor paraffin sections. We observed increased oxidative stress in G6PDKO;KPL lung tumors compared to G6PDWT;KPL lung tumors (New Fig. 4I, Page 9, Line 6-7). This suggests that the loss of G6PD triggers redox imbalance and oxidative stress, and this occurrence is independent of p53. This also suggests that oxidative stress alone, without p53, is not sufficient to slow KL lung tumor growth. It underscores that the slower growth of G6PDKO;KL tumors may be attributed to p53 activation, possibly triggered by oxidative stress. This has also been incorporated into discuss section (Page 16, Line 23-26; Page 17, Line 1-2): “The loss of G6PD significantly increases oxidative stress in KL lung tumors, potentially leading to the activation of p53 and the upregulation of its downstream targets to impede tumor growth. Our findings indicate that the reduction of KL lung tumors by G6PD ablation is rescued by the absence of p53. Despite this, increased oxidative stress persists in G6pdKO;KPL lung tumors. These findings suggest that oxidative stress alone, without p53, is not sufficient to impede KL lung tumor growth. Thus, the slow growth of G6PD-knockout KL lung tumors is attributed to p53 activation inhibiting tumor progression.”

The authors have misunderstood Point 2. NADPH and NADP+ levels should be measured in KPL tumors and compared to KPL tumors with loss of G6PD.

We acknowledge the suggestion to include NADPH and NADP+ levels in KPL tumors with or without G6PD, as it could enhance the strength of our manuscript. Therefore, during submitting the first revision, we started to breed more *G6PD^{flox/flox};KPL* and *G6PD^{+/+};KPL* mice. On 02/23/2024, we intranasally infected the mice with lentiviral-Cre to induce lung tumors for this experiment. At 6 weeks post-tumor induction (04/08/2024), we sacrificed the mice to collect tumors for metabolomics analysis. However, unexpectedly, we observed no growth of lung tumors in any of the mice, which has never happened in our previous experiments. The only plausible explanation is that the lentiviral-Cre titers were reduced during storage at -80°C, likely influenced by temperature fluctuations resulting from frequent opening of the freezer door. We have infected another batch of mice with the new virus on 4/12/2024. Nevertheless, it will require an additional 6 weeks for tumor growth and another 1-2 weeks for turnover of metabolomic results. Therefore, we won't be able to provide this data at this time.

While we lack specific data on NADPH and NADP+ levels, the increased oxidative stress observed in *G6pd^{KO};KPL* lung tumors compared to *G6pd^{WT};KPL* tumors suggests a compromised redox balance in KPL lung tumors due to G6PD deficiency. We add this in discussion “Our findings reveal that the reduction of KL lung tumors by G6PD ablation is rescued by the absence of p53. Despite this, increased oxidative stress persists in *G6pd^{KO};KPL* lung tumors, indicating impaired redox homeostasis in KPL lung tumors due to G6PD deficiency.” (Page 16, Line 434-436 in revised manuscript). We don't think we can conclude that “G6PD loss should decrease NADPH levels independent of p53 status”

considering the metabolic rewiring is different among KP, KL and KPL lung tumors.

To address “A major question is why G6PD loss does not alter NADPH levels in KP lung tumors”, we have discussed in Page 14-15 Line 455-414 “we found that loss of G6PD-mediated oxPPP has no impact on KP lung tumorigenesis. To overcome G6PD loss, KP lung tumors may employ a strategy to boost NADPH production through alternative pathways like ME1, IDH1, or folate metabolism. This compensatory NADPH generation could also be accompanied by an alternative source of ribose-phosphate, likely through the non-oxPPP. Additionally, KP lung tumors could obtain lipids and/or nucleosides from the surrounding microenvironment or bloodstream, thereby reducing their dependence on G6PD-derived products. Comprehensive mechanistic studies are needed to fully understand this resilience.”

Considering the results and conclusions we have reached, we believe that the absence of NADPH and NADP⁺ levels will not impact the quality of the manuscript. We hope you will accept our revision in its current version.

Point 3:

“Several technical details need to be provided in the Figures and Figure Legends....Later, in Figure 4i, the authors show a graph with an unlabeled y-axis. They mention that this is a “proliferation rate,” but additional details should be included in the graph and Figure legend. “

Figure 3f is still unclear. The authors state this is “Proliferation rate”, but only “Confluence %” is provided. It is unclear what the proliferation rate that they are referring to is.

“The IncuCyte live-cell imaging system automatically quantified cell surface area coverage to determine the percentage of confluence in one well of a 12-well plate every 2 hours over 4 days, and the slope of the time-course changes in the percentage of confluence was utilized to reflect the proliferation rate.” was included in the Materials and Methods section (Page 23, Line 591-594 in revised manuscript). “Proliferation rate was calculated using the percentage of confluence” was included in the figure legend (Page 39, Line 1048-1049 in revised manuscript). For y-axis in Fig. 4i (new Fig. 3f in revised manuscript), we changed it as “confluence (%)”.

Point 8:

“8. It is unclear why only four mice are shown in Figures 5E, F, H, and I if Fig 5K presents data from more than 25 mice of the same genotype.”

The authors' reply is:

“We randomly sacrificed 4 mice from each group at 6 weeks post-tumor induction for tumor burden analysis, as depicted in New Fig. 4i. Since no significant differences in tumor burden were observed, the remaining mice were retained for mouse survival analysis (New Fig. 4k), contributing to a more robust conclusion.”

No statistical analysis of Figures 4h and 4i is provided; thus, the authors cannot state that “no significant differences in tumor burden were observed.” Further, it appears that with a larger cohort of mice, these differences could potentially become significant.

In our previous submission, we omitted “ns” from the graphs where no significant difference was observed. In this revised version, “ns” has been included in all figures, including Fig. 4f, Fig. 4i and 4j. There is no significant difference observed between mice bearing WT and KO KPL lung tumors in wet lung weight (Fig. 4f), quantification of tumor number (Fig. 4h), and tumor burden (Fig. 4i). Due to the variability in tumor burden among the mice, we concur that expanding the sample size will improve the robustness of the data. However, we assert that conducting a mouse survival study would yield more meaningful results compared to analyzing tumor burden at a single time point, as supported by our own research and that of other groups

utilizing GEMMs for cancer research. In our survival study, we have utilized a larger cohort ($n \geq 26$), demonstrating no significant difference between the two groups. Therefore, based on the tumor burden analysis and survival study, we can conclude that G6PD loss does not affect KPL lung tumorigenesis.

Reviewer #4 (Remarks to the Author):

The authors have addressed my queries thoroughly and comprehensively. I have no further requests and believe the manuscript is ready for editing, submission of raw data, and if these are suitable, publication.

REVIEWERS' COMMENTS

Reviewer #1 (Remarks to the Author):

No further comments, congratulations to the authors.

Reviewer #3 (Remarks to the Author):

After reviewing the response and the manuscript, the authors have addressed the concerns and the data in the manuscript supports the conclusions. I do not believe any additional experiments are required.